# The Fifth International Workshop on Ice Nucleation phase 2 (FIN-02): Laboratory intercomparison of ice nucleation measurements

DeMott, Paul J.[1], Ottmar Möhler[2], Daniel J. Cziczo[3,4], Naruki Hiranuma[2,a], Markus D. Petters[5], Sarah S. Petters[5,b], Franco Belosi[6], Heinz G. Bingemer[7], Sarah D. Brooks[8], Carsten Budke[9], Monika Burkert-Kohn[10], Kristen N. Collier[8], Anja Danielczok[7,c], Oliver Eppers[11], Laura Felgitsch[12], Sarvesh Garimella[3,d], Hinrich Grothe[12], Paul Herenz[13], Thomas C. J. Hill[1], Kristina Höhler[2], Zamin A. Kanji[10], Alexei Kiselev[2], Thomas Koop[9], Thomas B. Kristensen[13,e], Konstantin Krüger[7,2], Gourihar Kulkarni[14], Ezra J. T. Levin[1], Benjamin J. Murray[15], Alessia Nicosia[6,f], Daniel O'Sullivan[15], Andreas Peckaus[2,g], Michael J. Polen[16], Hannah C. Price[15,h], Naama Reicher[17], Daniel A. Rothenberg[3], Yinon Rudich[17], Gianni Santachiara[6], Thea Schiebel[2], Jann Schrod[7], Teresa M. Seifried[12], Frank Stratmann[13], Ryan C. Sullivan[16], Kaitlyn J. Suski[1,i], Miklós Szakáll[11], Hans P. Taylor[5], Romy Ullrich[2], Jesus Vergara-Temprado[15,10], Robert Wagner[2], Thomas F. Whale[15], Daniel Weber[7], André Welti[13,j], Theodore W. Wilson[15,k], Martin J. Wolf[3], and Jake Zenker[8]

[1]Department of Atmospheric Science, Colorado State University, Fort Collins, CO 80523-1371, USA
[2]Karlsruhe Institute of Technology (KIT), Institute of Meteorology and Climate Research (IMK-AAF), Eggenstein-Leopoldshafen, Germany
[3] Department of Earth, Atmospheric and Planetary Sciences, Massachusetts Institute of Technology, Cambridge, MA, USA
[4]Department of Civil and Environmental Engineering, Massachusetts Institute of Technology, Cambridge, MA, USA
[5]Department of Marine, Earth and Atmospheric Sciences, North Carolina State University, Raleigh, NC, USA
[6]Institute of Atmospheric Sciences and Climate (ISAC-CNR), Bologna, Italy
[7]Institute for Atmospheric and Environmental Sciences, Goethe-University Frankfurt, 60438 Frankfurt am Main, Germany
[8]Department of Atmospheric Sciences, Texas A&M University, College Station, TX, USA
[9]Faculty of Chemistry, Bielefeld University, Bielefeld, Germany
[10]Institute for Atmospheric and Climate Science, ETH Zurich, Zurich, Switzerland
[11]Institute for Atmospheric Physics, Johannes Gutenberg University, Mainz, Germany
[12]Institute of Materials Chemistry, TU Wien, Vienna, Austria
[13]Leibniz Institute for Tropospheric Research, 04318 Leipzig, Germany
[14]Atmospheric Sciences and Global Change Division, Pacific Northwest National Laboratory, Richland, WA, USA
[15]Institute for Climate and Atmospheric Science, School of Earth and Environment, University of Leeds, Woodhouse Lane, Leeds, LS2 9JT, UK
[16]Center for Atmospheric Particle Studies, Carnegie Mellon University, Pittsburgh, PA, USA
[17]Department of Earth and Planetary Sciences, Weizmann Institute, Rehovot 76100, Israel
[a]now at: Department of Life, Earth and Environmental Sciences, West Texas A&M University, Canyon, TX, USA
[b]now at: Department of Environmental Sciences and Engineering, University of North Carolina, Chapel Hill, NC, USA
[c]now at: German Weather Service, Satellite-based Climate Monitoring, 63067 Offenbach am Main, Germany
[d]now at: ACME AtronOmatic, LLC, Portland, OR, USA
[e]now at: Division of Nuclear Physics, Lund University, Box 118, Lund SE-22100, Sweden
[f]now at: Laboratoire de Méteorologie Physique (Lamp-CNRS), Aubière, France
[g]now at German Aerospace Center (DLR), Institute of Technical Physics, 70569 Stuttgart, Germany
[h]now at Facility for Airborne Atmospheric Measurements, Cranfield, MK43 0AL, UK
[i]now at: Pacific Northwest National Laboratory, Richland, WA, USA
[j]now at: Finnish Meteorological Institute, FI-00101 Helsinki, Finland
[k]now at Owlstone Medical Ltd., 162 Cambridge Science Park, Milton Road, Cambridge, CB4 0GH, UK

*Correspondence to*: Paul J. DeMott (Paul.Demott@colostate.edu)

**Abstract.** The second phase of the Fifth International Ice Nucleation Workshop (FIN-02) involved the gathering of a large number of researchers at the Karlsruhe Institute of Technology's Aerosol Interactions and Dynamics of the Atmosphere (AIDA) facility to promote characterization and understanding of ice nucleation measurements made by the variety of methods used worldwide. Compared to the previous workshop in 2007, participation was doubled, reflecting a vibrant research area. Experimental methods involved sampling of aerosol particles by direct processing ice nucleation measuring systems from the same volume of air in separate experiments using different ice nucleating particle (INP) types, and collections of aerosol particle samples onto filters or into liquid for sharing amongst measurement techniques that post-process these samples. In this manner, any errors introduced by differences in generation methods when samples are shared across laboratories were mitigated. Furthermore, as much as possible, aerosol particle size distribution was controlled so that the size limitations of different methods were minimized. The results presented here use data from the workshop to assess the comparability of immersion freezing measurement methods activating INPs in bulk suspensions, methods that activate INPs in condensation and/or immersion freezing modes as single particles on a substrate, continuous flow diffusion chambers (CFDCs) directly sampling and processing particles well above water saturation to maximize immersion and subsequent freezing of aerosol particles, and expansion cloud chamber simulations in which liquid cloud droplets were first activated on aerosol particles prior to freezing. The AIDA expansion chamber measurements are expected to be the closest representation to INP activation in atmospheric cloud parcels in these comparisons, due to exposing particles freely to adiabatic cooling.

The different particle types used as INPs included the minerals illite NX and K-feldspar, two natural soil dusts representative of arable sandy loam (Argentina) and highly erodible sandy dryland (Tunisia) soils, respectively, and a bacterial INP (Snomax®). Considered together, the agreement among post-processed immersion freezing measurements of the numbers and fractions of particles active at different temperatures following bulk collection of particles into liquid was excellent, with possible temperature uncertainties inferred to be a key factor in determining INP uncertainties. Collection onto filters for rinsing versus directly into liquid in impingers made little difference. For methods that activated collected single particles on a substrate at a controlled humidity at or above water saturation, agreement with immersion freezing methods was good in most cases, but was biased low in a few others for reasons that have not been resolved, but could relate to water vapor competition effects. Amongst CFDC-style instruments, various factors requiring (variable) higher supersaturations to achieve equivalent immersion freezing activation dominate the uncertainty between these measurements, and for comparison with bulk immersion freezing methods. When operated above water saturation to include assessment of immersion freezing, CFDC measurements often measured at or above the upper bound of immersion freezing device measurements, but often underestimated INP concentration in comparison to an immersion freezing method that first activates all particles into liquid droplets prior to cooling (the PIMCA-PINC device), and typically slightly underestimated INP number concentrations in comparison to cloud parcel expansions in the AIDA chamber; this can be largely mitigated when it is possible to raise the relative humidity to sufficiently high values in the CFDCs, although this is not always possible operationally.

Correspondence of measurements of INPs among direct sampling and post-processing systems varied depending on the INP type. Agreement was best for Snomax® particles in the temperature regime colder than -10°C, where their

ice nucleation activity is nearly maximized and changes very little with temperature. At warmer than -10°C, Snomax[®] INP measurements (all via freezing of suspensions) demonstrated discrepancies consistent with previous reports of the instability of its protein aggregates that appear to make it less suitable as a calibration INP at these temperatures. For Argentinian soil dust particles, there was excellent agreement across all measurement methods; measures ranged within one order of magnitude for INP number concentrations, active fractions and calculated active site densities over a 25 to 30°C range and 5 to 8 orders of corresponding magnitude change in number concentrations. This was also the case for all temperatures warmer than -25°C in Tunisian dust experiments. In contrast, discrepancies in measurements of INP concentrations or active site densities exceeded two orders of magnitude across a broad temperature measurements found at warmer than -25°C in a previous study were replicated. Discrepancies also exceeded two orders of magnitude at temperatures of -20 to -25°C for K-feldspar, but these coincided with the range of temperatures where INP concentrations increase rapidly at approximately an order of magnitude per 2°C cooling for K-feldspar.

These few discrepancies did not outweigh the overall positive outcomes of the workshop activity, nor the future utility of this data set or future similar efforts for resolving remaining measurement issues. Measurements of the same materials were repeatable over the time of the workshop and demonstrated strong consistency with prior studies, as reflected by agreement of data broadly with parameterizations of different specific or general (e.g., soil dust) aerosol types. The divergent measurements of the INP activity of illite NX by direct versus post-processing methods was not repeated for other particle types, and the Snomax[®] data demonstrated that, at least for a biological INP type, there is no expected measurement bias between bulk collection-immediately processed freezing methods to as warm as -10°C. Since particle size ranges were limited for this workshop, it can be expected that for atmospheric populations of INPs, measurement discrepancies will appear due to the different capabilities of methods for sampling the full aerosol size distribution, or due to limitations on achieving sufficient water supersaturations to fully capture immersion freezing in direct processing instruments. Overall, this workshop presents an improved picture of present capabilities for measuring INPs than in past workshops, and provides direction toward addressing remaining measurement issues.

## 1 Introduction

Ice nucleating particles (INPs) are relatively rare atmospheric particles that play a large role in affecting cold cloud properties and precipitation processes. Their presence is needed to initiate ice crystal formation in the absence of conditions that would favor homogeneous freezing nucleation. They are needed as a trigger even in cases where secondary ice formation may be expected to occur. Their varied loading may influence cloud lifetime positively or negatively, as well as impact precipitation rates in mixed phase clouds (e.g., Tan et al., 2016; Fan et al., 2017). Furthermore, the efficacy of different aerosol types as INPs varies greatly, and this is not well resolved for major or minor atmospheric aerosol populations. There is a tremendous need to measure atmospheric INP populations, and to parameterize these for use in numerical models of all scales, where greatly simplified assumptions on ice phase transitions in clouds are presently used or are thought to be necessary for computational reasons. Studies of INPs occur in laboratory settings where high aerosol loadings are possible, but also in field scenarios where the low number concentrations of INPs challenge near real-time samplers and require larger bulk collections to attempt to quantify

INPs at modest supercooling. Consequently, a variety of devices exists, and the development and use of different instruments continues to expand during a period of great growth in research on mixed phase and ice cloud processes (DeMott et al., 2011). For these reasons, a series of workshops was convened over the course of a year in 2014 to 2015, continuing the historical efforts of the international ice nucleation community to compare and contrast

measurements, both to advance understanding within the community and to offer an assessment to the user communities of capabilities and present uncertainties of measurements being published independently.

The philosophy, three-phase nature, and general overview of the Fifth International Ice Nucleation Workshop, dubbed FIN will be provided in a separate publication in preparation. Briefly, and distinct from most previous workshops in its comprehensive scope, FIN sought to perform comprehensive operational comparisons of ice

nucleation instruments for sampling calibration type INPs (representative of different atmospheric classes) in a laboratory setting and for sampling ambient atmospheric aerosols in a natural setting. In addition, the component FIN-01 (first study in late 2014) sought to intercompare single particle mass spectrometer instruments that are sometimes used to assess the detailed chemical composition of INPs by sampling the residues of ice crystals nucleated in flow diffusion chambers or aerodynamically segregated from atmospheric clouds. FIN-01 tested these instruments for their

determined reference mass spectra on some of the INPs also planned for use in comparing ice nucleation instruments, it compared the different clustering algorithms used by the aerosol mass spectral community, and it repeated testing on ice crystal residues. FIN-02, the workshop phase discussed herein, was the laboratory ice nucleation instrument intercomparison. FIN-03, the field phase, was conducted at Storm Peak Laboratory in Steamboat Springs, CO. A final aspect of both FIN-01 and FIN-02 was to provide a minor period within the overall informal gatherings for scientific

study that would feature a formal intercomparison of measurements. FIN was a volunteer activity on the part of participants, who agreed to participate to their fullest extent in both the informal and formal components, but were also free to explore new developments. Referees were solicited for organizing and analyzing the results of formal comparisons in FIN-01 and FIN-02. These formal or so-called "blind" experiments were conducted to investigate the degree to which the informal results presented in papers such as this one could be independently reproduced.

This paper describes the goals and objectives, and some detailed results from the second phase of FIN, known as FIN-02, focused around comparing ice nucleation measurement systems in laboratory studies of known INPs. A related paper in preparation will describe the separate but integrated activity of comparing these same instrument systems in, the formal comparison period during FIN-02. The FIN-02 workshop was held at the Aerosol Interactions and Dynamics in the Atmosphere (AIDA) facility at Karlsruhe Institute of Technology, Eggenstein-Leopoldshafen,

Germany during March 2015. This facility was also the site for the FIN-01 workshop. FIN-02 was designed to be in the classical form of an ice nucleation workshop, from the standpoint of having a legacy in similar workshops dating back to the late 1960s, as discussed in relation to the 2007 International Workshop on Comparing Ice Nucleation Measuring Systems (ICIS-2007) (unofficially the Fourth International Workshop on Ice Nuclei) by DeMott et al. (2011). The impetus for continuation of the ice nucleation workshop concept was given in that paper. Significant

additional developments have occurred in the field of ice nucleation measurements since the time of the 4th workshop, including widespread participation from a global community of researchers and commercialization development of INP measurement systems that directly process aerosols. The FIN-02 workshop was held at the AIDA facility to take

advantage of the 4[th] workshop experience, also held there, and to once again coordinate other measurements with experiments in the AIDA cloud chamber as a mimic of ice nucleation within atmospheric adiabatic cloud parcels.

The goals and objectives of FIN-02 were to:

1) Compare ice nucleation measurement systems for conditions considered to be equivalent as much as possible, across a wide dynamic temperature range, including temperatures warmer than -15°C.

2) Gain insights into how detection of ice nucleation is influenced by the specific configuration of similar measurement systems.

3) Gain insights into the strengths and weaknesses, limits of detection, potential artifacts and other peculiarities of different INP detection systems.

4) Utilize different INP types to investigate if differences between instruments occur with these different types. This paper is intended as an overview of the informal activities of the workshop while addressing the majority of these objectives. It is not intended to answer all of the goals and objectives that are better addressed in separate studies. It is not our intent to rigorously test the capabilities of different measurement systems, but rather to point to areas of success and areas for needed development or further research. For these reasons, and to include as many measurements as possible in comparisons, we focus primarily on measurements relevant to immersion freezing nucleation, as discussed further below. This allows for integrating the most possible measurements into comparisons made for assessing one important aspect of the state of the art of ice nucleation measurements.

**2 Methods**

Guided by the objectives of FIN-02 and the variety of current systems available for measuring INPs, two broad categories of ice nucleation instruments were defined for studies. These categories are, firstly, instruments operating online or for direct processing of aerosol particles and, secondly, those utilizing collections of particles for subsequent offline or post-processing. This categorization to a large extent also separates methods that sample "dry" particles and those that utilize "wet" suspensions of particles in liquid for assessing freezing properties, with a few exceptions. Methods for sampling particles from a dry state permit assessment of the action of a variety of ice nucleation mechanisms that occur in different water relative humidity ($RH_w$) regimes: deposition nucleation primarily occurring below water saturation, and condensation and immersion freezing on approaching or exceeding water saturation, where cloud droplet activation occurs. Experiments wherein particles are suspended in water isolate the action of immersion freezing nucleation direction, and certain methods allow for isolating immersion freezing for single aerosol particles (Burkert-Kohn et al., 2017). No measurements of contact freezing were included in this study, and neither will we discuss results herein from workshop measurements that were made in the regime associated with deposition nucleation (including at temperatures below the homogeneous freezing temperature of pure water droplets ~~t~~). Instead, we will focus on inter-comparisons of particles acting via immersion freezing or proximal behaviors. By proximal behaviors, we follow the terminology of Vali et al. (2015), wherein condensation freezing is not necessarily considered as distinguishable from immersion freezing, and, hence, direct processing of particles in diffusion chambers measuring in the regime well above water saturation are considered to be able to approximate more direct measurements of immersion freezing.

The number of ice nucleation measurement systems participating in FIN-02 was slightly more than twice the number that participated in the 4th workshop in 2007, reflecting a similar increase in the number of researchers now operating in this field. There were 21 total systems represented in FIN-02, 9 directly processing and 11 post-processing instruments, plus the AIDA chamber. Names, basic descriptions, and general operating principles are provided in Tables 1 and 2, and sections 2.1 and 2.2. Detailed implementations of the basic principles in each device are given in the Supplement sub-sections. Shorthand names of instruments are defined in the manuscript at first introduction. The thermodynamic trajectories used by the primary instrument types used in FIN-02 are shown in Fig. 1 and their basic manners of operation are discussed in the two following sections. These sections are followed by a section describing the general manner of conduct of the workshop, including aerosol generation procedures. This becomes important for shaping the progression of how results are discussed in this paper.

### 2.1 Direct sampling systems

Direct sampling systems used in FIN-02 included continuous flow systems and the AIDA controlled expansion cloud chamber (see Table 1). Continuous flow ice-thermal diffusion chambers sample initially dry particles and expose these to conditions leading to ice nucleation. Amongst these were portable chambers with cylindrical (e.g., Rogers et al., 1988) and parallel plate (e.g., Stetzer et al., 2008) wall systems. The former included the Colorado State University continuous flow diffusion chamber (CFDC-CSU) and systems descendant from this design: the Texas A&M continuous flow diffusion chamber (CFDC-TAMU) and the Ice Nucleation Instrument of the Karlsruhe Institute of Technology (INKA). In all of these, a cylindrical aerosol lamina representing a minor portion of the total flow is constrained within particle-free sheath flows (top to bottom) between two cylindrical walls that are ice-coated and can be independently temperature-controlled to determine the $RH_w$ and temperature at the center of the aerosol lamina in upper "growth" regions of the chambers. Parallel plate systems insert the downward-flowing aerosol lamina between sheath flows inside two parallel rectangular ice-coated plates to similarly expose particles to controlled temperature and humidity conditions in their growth sections. Parallel plate devices of quite common design included in FIN-02 were the ETH-Zurich Portable Ice Nucleation Chamber (PINC), the Pacific Northwest National Laboratory Compact Ice Chamber (CIC-PNNL), and the Droplet Measurement Technologies SPectrometer for Ice Nucleation (SPIN) devices operated by groups from the Massachusetts Institute of Technology (SPIN-MIT) and the Leibniz Institute for Tropospheric Research (SPIN-TROPOS).

Measurements from continuous flow diffusion chambers are represented by the red lines in Fig. 1. All of the continuous flow diffusion chambers have the ability to raise the $RH_w$ above the water saturation line in order to investigate ice nucleation during or following condensation of water droplets. Once water droplets have formed, a means is required to discriminate ice particles from water droplets. The most common method used for phase discrimination in continuous flow chambers is to selectively shrink activated liquid droplets to accentuate ice crystals by their larger optical size. Some instruments use laser light depolarization for phase discrimination, but this is typically a suitable method only for higher signal to noise situations and ice active fractions that exceed several percent of particles (Nicolet et al., 2010; Garimella et al., 2016; Zenker et al., 2017). For these reasons, all such devices in FIN-02 include an "evaporation" section as a shorter column length below their growth sections, where the $RH_w$ is

lowered toward ice saturation conditions by setting the two wall temperatures to be equivalent at either the warmest wall, coldest wall, or lamina temperature in the growth sections (see Supplement). When the temperature gradient in the growth section is adjusted to generate water supersaturated conditions that activate cloud droplets within the aerosol lamina, the lower relative humidity in the evaporation section shrinks droplets back toward haze particle sizes.

This method works up to some high value of $RH_w$ in the growth section whereupon activated cloud droplets survive through to detection, often referred to as the water droplet breakthrough $RH_w$. The breakthrough value varies with temperature, geometry and flow rate for different devices. Therefore, a single $RH_w$ level in the growth region for breakthrough to occur is not noted in Fig. 1. Instead, results are stated as being associated with specific $RH_w$ values (or % supersaturation values, which equal $RH_w$-100) that are simply a value that was lower than the droplet breakthrough condition. In some cases, this was the maximum $RH_w$ achievable in the growth region prior to droplet breakthrough.

The focus on reporting of flow chamber data at highly supersaturated conditions as best representing proximal immersion freezing behaviors is motivated by recent research and publications. Presently, continuous flow diffusion chamber instruments in general do not expose particles to uniform water supersaturations with the precision achieved by cloud condensation nuclei (CCN) instruments. Rather, the transition into the immersion freezing regime above water saturation does not occur sharply in line with the supersaturation calculated for the aerosol central lamina, but ensues completely only at higher $RH_w$ as controlled by aerosol particle properties and instrument characteristics. For example, hygroscopicity and kinetic factors control water uptake, chambers have different flow rates and growth section lengths, there is a finite difference in $RH_w$ across the aerosol lamina, and many devices appear (for as yet unclear reasons) often to induce a proportion of all particles to escape the defined aerosol lamina and expose these particles to lower $RH_w$ outside of the intended central lamina (DeMott et al., 2015; Garimella et al., 2017). Hence, higher $RH_w$ is used in these instruments to bypass limitations in achieving CCN activation on the entire particle population, and to increase the condensation rate and thus water content of the formed droplets. The justification is to make the measurement conditions (most particles placed in cloud droplets larger than a few μm prior to freezing) more similar to the cloud parcel simulations in the AIDA cloud chamber (see next section) with more typical cloud supersaturations and time scales. In practice, continuous flow instruments processed dry particle samples by slowly "scanning" $RH_w$ from near ice saturation conditions to water supersaturated conditions (see DeMott et al., 2011 for discussion of these methods, and the Supplement Section S.1.2 for a few examples. Investigators were then asked to select those data they felt represented the highest (not necessarily maximum) immersion freezing activity possible to assess in their $RH_w$ scans, and reported the INP concentrations and $RH_w$ values selected.

For continuous flow diffusion chambers in FIN-02, no additional corrections besides internal losses (if known) were applied. In other words, correction factors to account for the inability to assess maximum activation in the supersaturated regime, as discussed by DeMott et al. (2015), Garimella et al. (2017), and Burkert-Kohn et al. (2017) were not applied. We discuss particle losses in lines feeding various instruments in section 2.4.

Unique among the continuous flow chambers in FIN-02 was the combination of the PIMCA (Portable Immersion Mode Cooling chamber) device in series with the PINC instrument, referred to herein as PIMCA-PINC, wherein droplets are first activated on individual dry particles at temperatures above 0°C prior to cooling during flow into the

colder temperature PINC to observe immersion freezing (See Supplement Section S.1.6). This is intended to provide the most explicit simulation of immersion freezing. Experimental trajectories for PIMCA-PINC essentially follow those of post-processing immersion freezing devices (see below), but activation is on single, immersed particles. We note that either PINC or PIMCA-PINC operations were exclusive for a given INP type on a given day.

The CIC-PNNL flow diffusion chamber instrument was also operated at times in a non-standard manner to activate droplets at high supersaturation under modest supercooling in its upper chamber region and cool them to immersion freezing in the lower chamber region during FIN-02 studies (Kulkarni et al., 2018), but only data collected in the standard manner of generating near steady-state supersaturation at a single lamina temperature were included in the comparison presented herein.

The 84 m$^3$ AIDA controlled expansion cloud chamber was used to perform experiments serving as cloud parcel comparison to other measurements. In this regard, we follow the example of the 2007 workshop, and a key recommendation from ice nucleation workshops prior to that time that an expansion cloud chamber be utilized to provide a simulation of cloud activation (DeMott et al., 2011). Schematic thermodynamic paths of the AIDA chamber experiments are shown by the yellow curves in Fig. 1. Of note in this regard is the fact that small supersaturations

occur prior to cloud formation in AIDA, but once droplets are activated on all particles, the cooling follows at water saturation until a point where evacuation can no longer sustain cooling against the surrounding warmer volume, and clouds begin to dissipate. In this regime at water saturation, a comparison to continuous flow chambers should not be made at water saturation, but only for the higher supersaturations that assure more complete droplet activation within the sample lamina of CFDCs. For comparison to immersion freezing results by other methods, we have omitted AIDA

experiments for which high ice nucleation rates were achieved at below water saturation (e.g., deposition nucleation regime), and wherein full subsequent activation of particles as CCN was not achieved due to rapid ice growth.

**2.2 Post-processing systems**

      Two types of instruments ~~that~~ post-processed particle collections. These were diffusion chamber devices that processed particles collected onto substrates and devices that recorded freezing by particles within liquid droplets or

confined liquid volumes. Thermal diffusion chamber devices that processed particles on substrates during FIN-02 were the FRIDGE (FRankfurt Ice nucleation Deposition freezinG Experiment) instrument operated to above water saturation in its standard mode (Klein et al., 2010; Schrod et al., 2016), referred to here as FRIDGE-STD (see Supplement section S.2.10), and the DFPC-ISAC (Dynamic Filter Processing Chamber - Institute of Atmospheric Sciences and Climate, National Research Council of Italy) instrument (Santachiara et al., 2010; Belosi et al., 2014;

see Supplement Section S.2.11). These two methods were developed to measure condensation freezing and deposition ice nucleation modes from below to slightly above water saturation. The thermodynamic path of measurements using these instruments is the same as for the continuous flow diffusion chambers in Fig. 1 (red lines), but typically terminate 1-2% $RH_w$ above water saturation. Both devices were designed with the intention to overcome the so-called "volume effect" on freezing (e.g., Bigg, 1990; Schrod et al., 2016 and references therein) which describes the underestimation

of INPs that can occur when processing particles on a substrate in a diffusion chamber due to vapor pressure reduction by the first particles freezing, especially when larger volumes are collected that result in larger numbers of INPs per

surface area of the substrate. The FRIDGE instrument seeks to limit this effect using a low-pressure diffusion chamber to enhance vapor deposition over particles collected onto silicon wafer substrates, while the DFPC instrument follows the methods of Langer and Rodgers (1975) to focus a flow of humid air over filter substrates, and using the best practices outlined by Bigg (1990). For both devices, attempts were made to limit particle collections to shorter times during FIN-02, in order to keep particle loading light on the substrates. An additional fundamental difference between FRIDGE and the DFPC is the use of the filter substrate in the latter case, which is placed on a paraffin layer that is heated to establish thermal contact with a cold plate prior to ice nucleation measurements. The uncertain difference between condensation freezing and immersion freezing mechanisms (Vali et al., 2015) argues for an evaluation of results obtained near water saturation in this intercomparison as representative of proximal immersion freezing, for both instruments.

Immersion freezing measurements of collected particles suspended in water are depicted in Fig. 1 by the blue arrows. These measurements fall along the water saturation line because collected particles are suspended in pure water whose final water activity is essentially 1. In some cases, the mass and surface area within liquid water volumes is varied over several orders of magnitude of weight percent, via adding purified water for dilution, in order to cover a range of activation temperatures. The various water ~~wet~~ suspension methods for immersion freezing used in FIN-02 are listed and referenced in Table 2. The specific immersion freezing methods are also described in the Supplement. The basic types of methods used involved: 1) cooling arrays of droplets of particle suspensions placed on a cold stage and within oil, as done with the Carnegie Mellon University Cold Stage (CMU-CS) and the Karlsruhe Institute of Technology Cold Stage (KIT-CS); 2) cooling of suspension aliquot volumes in array compartments, as done with the CSU Ice Spectrometer (IS) and the Bielefeld Ice Nucleation ARraY (BINARY); 3) creating and cooling emulsions of particle suspensions as done in the Vienna Optical Droplet Crystallization Analyzer (VODCA) instrument; 4) cooling of droplets containing particles that are pipetted directly onto a coated hydrophobic glass slide, as done with the University of Leeds Microliter Nucleation by Immersed Particles Instrument (μL-NIPI) and the North Carolina State University Cold Stage (NCSU-CS), and using similar droplet arrays on the FRIDGE substrates (referred to as FRIDGE-IMM in this case, for FRIDGE Immersion Freezing); 5) freezing of a droplet train within a microfluidic device WeIzmann Supercooled Droplets Observation on Microarray (WISDOM); 6) and cooling of levitated particles as in the Mainz Acoustic Levitator (M-AL).

Most groups using liquid suspension freezing methods shared common samples from collections into liquid water (see discussion of sampling protocol in section 2.3), while in many cases the IS and FRIDGE-IMM measurements involved processing particles re-suspended from filters in pure water. Among these measurements, only the μL-NIPI, hereafter simply NIPI, measurements were conducted immediately after collection at KIT, while others processed the samples at their home institutes.

### 2.3 Generation of varied INP types and general study

A variety of relevant aerosol particle types were produced for FIN-02 studies, as listed in Table 3. These types reflect key mineral compounds of atmospheric desert dust aerosols (illite NX) or their key components (K-feldspar), natural soil dust samples of varied arability collected from different regions of the world (Argentinian soil dust,

erodible Tunisian soil dust, Saharan dust), and a biological (microbial, proteinaceous) INP type (Snomax[®]). These different INPs also span a range of activation temperatures that cover most of the mixed-phase cloud regime (i.e., 0 to -36°C). Thus, they provide a stringent examination of measurement capabilities and any biases that may occur.

Aerosol generation methods largely followed those presented in Hiranuma et al. (2015). Particles were independently provided to two different chambers, these being the AIDA chamber and a 4 m$^3$ holding chamber that will be referred to here as the aerosol particle chamber (APC). A total of 27 AIDA and 29 APC experiments were carried out during FIN-02. The particle types used for all 56 experiments are summarized in Table 3. Dry soil and mineral dust particles were generated using a rotating brush disperser (PALAS, RBG1000) and were subsequently passed through a series of inertial cyclone impactor stages (with 50% cut-point diameters of about 5 and 1 μm) prior to introduction into each chamber. This was an important step in limiting the numbers of particles present at sizes above 1 μm and emphasizing sizes that could be efficiently sampled by all measurement systems, including continuous flow devices. While natural particle distributions may sometimes include INPs to much larger sizes, it was deemed important for this study to limit this factor that can lead to measurement discrepancies due to sampling limitations. Size distributions of dry particles were measured using a scanning mobility particle sizer (SMPS, TSI Inc., Model 3081 differential mobility analyzer, DMA, and Model 3010 condensation particle counter, CPC) and an aerodynamic particle sizer (APS, TSI Inc., Model 3321). Particles were assumed to be spheres, and dynamic shape factors and particle densities listed in Table 3 were used to obtain the geometric-based (volume equivalent) diameters from the SMPS and APS data (Hiranuma et al., 2014b; 2015). Total particle surface areas were calculated and tabulated as a function of time using lognormal fits to size distributions in each experiment, as shown for two exemplary soil dust experiments (one AIDA and one APC) in Fig. 2.

Aerosol generation from aqueous suspensions was used during FIN-02 to generate INPs from Snomax[®]. The injection of Snomax[®] particles into the ventilated APC and AIDA vessels was achieved by atomization of a 5 g Snomax[®] suspension in 1 L of 18.2 MΩ ultrapure water followed by a diffusion dryer. The home-built atomizer used in Wex et al. (2015) was employed for all Snomax[®] particle generation. A total of eight polydisperse Snomax[®] injections were performed during FIN-02 (Table 3). Accordingly, aerosolized Snomax[®] particles were characterized for total number concentration and size distribution during each experiment.

Due to the efforts made to limit the generation of supermicron particles, the direct sampling ice nucleation instruments typically operated without special upstream impactors that would be used during atmospheric sampling to limit aerosol particles entering at sizes that could be mistaken as grown ice crystals (i.e., many CFDCs differentiate ice and aerosols by size alone). However, in a few experiments some larger particles were present that could contribute to size channels that typically demarcate only ice crystals. Redefinition of ice channels was done in some those cases to enable use of data from these experiments. An example of such corrections is given in Supplement Section S.1.2.

The daily protocol determined for aerosol generation and measurements is an aspect of these studies that bears strongly on the organization and discussion of results in this manuscript. Especially, sampling periods were organized to optimize the opportunities and conditions for all instruments to sample the variety of aerosols. Each day over the three-week workshop period typically began with fills of one INP type into the APC, and sampling of that aerosol into liquid and onto filters over a two-hour period for later assessment by the post-processing devices, as detailed below.

This was followed by the direct samplers processing the same INP type from the APC over another two-hour period. This typically permitted direct measurement at a couple specific temperatures, with data at other temperatures being acquired on another day for the same INP type (from the APC or AIDA). Then, at midday, the AIDA chamber would be filled, typically with a different INP type than used in the morning APC experiments. Direct sampling from AIDA

by the flow chambers would occur over a period of time just prior to the start of expansion cooling experiments. As well, collections onto filters or wafers used by the DFPC and FRIDGE device (standard method) would be made only from the AIDA aerosol fill on each day, since these methods required very short sampling times in order to limit particle loading. For example, the DFPC-ISAC filters were collected for periods of 10's of seconds. Other collections into liquid or onto filters for immersion freezing post-processing would not occur from the AIDA chamber prior to

expansions. A consequence of these procedures is that we will find it convenient to present results on different bases when discussing sampling from the APC and from AIDA. While we might ideally wish to present all data on the same basis as measurements are reported for atmospheric sampling, as number concentrations, we choose to do so only for the APC experiment period that offers the opportunity for comparing the most measurement systems. We describe how that is done next. For AIDA sampling, we will display results as active fractions and ice active site density, which

then allows integrating APC results along with AIDA results for the same INP types over the course of the workshop.

An example of a timeline of APC aerosol particle properties at the start of an experimental day is shown in Fig. 3 to demonstrate a typical morning of activity that integrated sampling by post-processing and direct systems for subsequent analyses. The chamber was initially filled with a high concentration of aerosol particles to create appropriate sampling conditions for the systems for immersion freezing post-processing, which can utilize high total

particle concentration (0.4 < mass concentration < 40 mg m$^{-3}$) in order to take advantage of the ability of some of these methods to assess the lower INP concentrations active at modest supercooling. Collection of particles into liquid suspensions for shared use by a suite of immersion freezing devices was performed by impinging a flow of particles from the APC into a glass bioaerosol sampler (SKC Inc.) (Hader et al., 2014; DeMott et al., 2017), referred to here as impinger samples. Two impinger samples were collected for ~120 min with a flow rate of 12.5 L min$^{-1}$ from the APC.

Flows were checked daily. Impingers were cleaned by wiping, rinsing with ultrapure water (18 MΩ-cm), and soaking in isopropyl alcohol overnight (2-propanol, ≥99.8%, ROTH). Before assembly the impingers were rinsed using ultrapure water once more. Following the sampling of Snomax® particles, the impingers were baked overnight at 200°C instead of soaking in alcohol. This was done to eliminate the possibility of carryover by ice active due to these biological samples. During sampling the water was replenished every ~30 minutes to keep the water level near 20 mL.

Due to evaporation, the final bottled volume was typically about half of the added water. The two impinger suspensions were combined into one sample and topped off to a total of ~36 mL. The sample was divided into 4 ml aliquots, bottled in pre-cleaned DNA free cryovials and stored locally in a freezer at -20°C. The same procedure was applied to the handling of blanks. At the end of the campaign, blanks and samples were placed on dry ice and shipped overnight to participating groups. Shipment to Israel was delayed by customs, allowing the sample to thaw en-route.

After receiving the sample, each group decided on their own sample storage and handling strategies. Again, we note that the University of Leeds group performed NIPI measurements of these suspensions on-site in Karlsruhe immediately after collection (i.e., without freezing).

Filter collections from the APC were made for post-processing by the NCSU CS, FRIDGE–(for immersion freezing), CSU IS, and Leeds NIPI instruments. These filters (0.2 μm pore size polycarbonate) ran up to 100 min, aligned with the same time period as the impingers. The FRIDGE filters were collected over multiple and shorter time periods (10 min) within this same time frame. Clean protocol for preparation of filters prior to sampling is discussed within the IS instrument description in the Supplement (Section S.2.3).

In some cases, aerosol particle concentrations were sufficiently depleted that an additional APC fill was done to augment collections and suffice for the later sampling by direct sampling-systems. A smaller injection of aerosol mass and concentration was typically used during the second fills in order to optimize sampling conditions for the direct sampling instruments (i.e., they would immediately begin sampling from the APC at that point). Other sampling was suspended for the direct sampling period. Such a two-stage injection period is highlighted in Fig. 3 by two regions of blue shading. Smoothed, interpolated aerosol curves are shown in Fig. 3. Exponential fits to decay periods were found to represent particle number concentrations with $r^2$ values exceeding 0.98, as expected for the first order loss processes occurring in the APC during sampling. These loss processes were dominated by the drawing of air from the APC by samplers, replenished in all cases by clean synthetic air. Curves are shown for total particle numbers, numbers of particles larger than 0.5 μm, and total particle surface area (spherical equivalency assumed for measured particle diameters). By integrating the exponential fit functions during sampling periods (blue shading), the integrated number concentrations and surface areas were determined for the combination of sample periods for post-processing. While we focus in the following discussion on quantifying the decay of total (CPC) particle numbers in order to correct INP number concentration data during the direct sampling periods ("online" used as shorthand in equations) for equivalency with the prior post-processing collection periods ("offline" used as shorthand in equations), we noted (not shown) differences ranging from only 10-30% in fractional loss rates when instead using particle numbers in the larger size range (>500 nm) to characterize particle number decay over time in the APC. These relatively minor differences, evident in Fig. 3, are consistent with the limited physical loss mechanisms existing for particles with mode sizes as shown in Fig. 2, and with limited numbers of supermicron particles that might be subject to sedimentation.

With reference to Fig. 3 and the fit to the exponential decay in any period $i$ with start and end times $t_{0i}$ and $t_{1i}$, respectively, the period average total aerosol concentration ($\bar{n}_{CPC,offline,i}$) is given by,

$$\bar{n}_{CPC,offline,i} = \int_{t_{0i}}^{t_{1i}} a_i \exp(b_i t) = \frac{a_i}{b_i}(\exp(b_i t_{1i}) - \exp(b_i t_{0i})) \tag{1}$$

Then for $i = 1$ to x periods of offline or post-processing sampling of aerosols from the APC for interval times $\Delta t_i$,

$$\bar{n}_{CPC,offline} = \sum_{i=1}^{x} \Delta t_i n_{CPC,offline,i} / \sum_{i=1}^{x} \Delta t_i \tag{2}$$

The APC sampling period offered the best opportunity to directly compare all ice nucleation instruments aside from the AIDA chamber, and to do so in the most straightforward manner (fewest assumptions) possible, as number concentrations per volume of air. To allow such a comparison, INP concentrations measured by direct sampling systems during the later period (green shaded area in Fig. 3) at any sample time $t$ were corrected to give equivalence

to the volumetric INP concentration measured by post-processing systems for their integrated sampling periods. That is,

$$n_{INP,online,corr}(t) = n_{INP,online}(t)\,\overline{n}_{CPC,offline}/n_{CPC}(t) \tag{3}$$

Correction factors for the online period were sometimes well in excess of 1 and up to 13 in a few experiments, since sampling from the APC oftentimes continued for more than a few hours after the impinger and filter sampling period had been completed. We may note that integrated (spherical equivalent) surface areas for the post-processing sample are determined in the same manner as reflected in Eqs. 1 to 3 for results shown in Section 3.2.

Direct sampling by flow chambers from AIDA was done in the time prior to the start of cloud expansions. DFPC-ISAC and FRIDGE filter collections, and collection of particles onto wafer substrates for use in the standard (deposition/condensation freezing) FRIDGE instrument processing mode (see Supplement Section 2.10) were also performed from the AIDA chamber for limited time periods, with the previously stated goal to limit total particle number loading for the diffusion chamber measurements. Aerosol number concentrations were typically much lower in AIDA, and since the total volume of AIDA is much larger, the decay of number concentrations due to sampling by other instruments prior to expansion was much slower than in the APC. Thus, in most cases, comparison measurement from other instruments to the AIDA ice crystal activation results could be made directly, with a small correction at times to account for the slightly higher total particle (CPC) number concentrations at the time of sampling versus those during the subsequent AIDA expansion. We compare activated fractions and the deterministic active site density parameter in these experiments so that multiple AIDA sampling experiments of the same aerosol types performed on different days may be included. This also allows for comparison of selected APC results to AIDA chamber results for similar aerosol types across the entire workshop period. This type of comparison also allows evaluation of measurement consistency, and comparison to previously published parameterizations.

We use calculated geometric aerosol surface areas, under the assumption of spherical equivalent diameters, to compute and compare surface active site densities, $n_{s,geo}$ $(T)$ (m$^{-2}$). Assuming a uniform distribution of $n_{s,geo}$ $(T)$ over a given total aerosol surface area ($S_{tot}$) and its size independency, we follow Hiranuma et al. (2015) to approximate $n_{s,geo}$ $(T)$ as,

$$n_{s,geo}(T) \approx \frac{n_{INPs}(T)}{S_{tot}} \tag{4}$$

Uncertainty in $n_{s,geo}$ $(T)$ is computed in quadrature from the confidence interval data for each INP type and assuming a 25% uncertainty in $S_{tot}$. $S_{tot}$ is computed by normalizing the integrated aerosol surface area (μm$^2$ cm$^{-3}$) by total particle number concentrations. Integrated surface areas listed in Table S1 of the Supplement are determined based on lognormal fits to the aerosol distribution merged over the full particle size range from aerodynamic and aerosol mobility measurements (Fig. 2). Values of $n_{s,geo}$ $(T)$ will be listed and plotted in m$^{-2}$ herein.

Finally, we note that no corrections for particle losses in sample lines are made for comparisons shown herein. This is due to the fact that these losses may be assumed to be negligible in comparison to other uncertainties as defined

by confidence intervals for the measurements. As noted in Fig. 2, both particle number and surface area in these experiments were mainly from particles in the size range between 0.1 and 1 μm. Using the worst-case sampling scenario, which was for the PIMCA-PINC instrument sampling from the AIDA chamber (flow rate of 1.6 L min$^{-1}$ through 5 m of 0.457 cm interior diameter stainless tubing, and assumed bulk particle density of 2.6 g cm$^{-3}$),

calculations of estimated penetration efficiency through tubing versus particle size were made using equations from Baron and Willeke (2005). Calculations captured diffusional losses in tubing, inertial losses in a straight tube (i.e., incline was ignored), and impaction losses in tubing (four 90° bends assumed). This demonstrated that penetration efficiency likely exceeded 88% at all sizes below 1 μm, and even at a size of 2 μm, the proximal upper size generated in any experiments, ~60% of the particles should have reached all instruments. All investigators were given the ability

to re-evaluate data quality and potential experimental issues after the original archive was produced. The amount of data contributed to final comparisons varied widely amongst the different instruments, in some cases due to operational issues that arose during the workshop.

### 3 Results and Discussion

### 3.1 APC sampling of INPs

As discussed in Section 2, the primary basis for comparison of methods for sampling different INP types from the APC was for the measured or calculated number concentrations of INPs. Four experiments in which the largest number of measurement methods sampled from the APC are shown in Figures 4 to 7. These comparisons necessarily exclude the AIDA chamber data. Each figure assesses, 1) comparisons of direct sampling devices (larger blue colored symbols of different types are for the CFDC-CSU, SPIN-TROPOS and SPIN-MIT, CIC-PNNL, INKA and PIMCA-PINC)

versus collection and post-processing instruments (all other symbols of various types and colors); 2) comparison of different methods for immersion freezing post-processing, whether as droplet arrays on substrates or in aliquot wells (IS, BINARY, NIPI, KIT-CS, NCSU-CS, CMU-CS, VODCA, FRIDGE-IMM), in microfluidic devices (WISDOM), or in an acoustic levitator (M-AL); 3) shared (most immersion freezing arrays or devices using the common impinger samples) versus individual samples (IS and FRIDGE-IMM); and 4) different collection methods (filters for IS and

FRIDGE-IMM, except for Snomax®; impingers for others).

The comparisons obtained for sampling Argentinian soil dust particles (Fig. 4) were among the best in this study. A striking feature of these results is the general correspondence between all methods and sampling techniques in ranges of overlap, as well as the apparent meshing of results from direct sampling and post-processing of immersion freezing to capture 7 or more orders of magnitude of INP activity in the temperature regime from -5 to -35°C. Direct

overlap showing correspondence of the continuous flow methods with a minimum of four different bulk methods occurs over 3 orders of magnitude range at temperatures from -20 to -30°C. Good consistency is also seen amongst direct sampling methods as a group and post-processing methods taken together, for shared impinger samples, and whether post-processed samples were collected by impinger or separate polycarbonate filters (IS and FRIDGE-IMM) that were subsequently rinsed of particles. Recall that the IS filter was collected simultaneously with the impinger

sample, while the FRIDGE-IMM filter was collected over a shorter time frame. The largest discrepancies, in

consideration of measurement uncertainties (see Supplement for explanation of measurement uncertainties for each device) occur at the coldest temperatures. In this region, data from the PIMCA-PINC instrument, which activates individually-grown droplets on particles prior to cooling, falls at the upper end of measured INP concentrations in comparison to the few other immersion methods that extended to the heterogeneous ice nucleation limit, just warmer than homogeneous freezing temperatures. We note again here that some scatter may occur in the direct processing flow diffusion chambers due to investigators deciding in each case what $RH_w$ above 100% to report data for as representing immersion freezing of the entire particle population.

Measurements for a Tunisian desert dust sampling experiment in the APC are shown in Fig. 5. Fewer overall measurements are available for this comparison. Nevertheless, results are similar to those obtained for the Argentinian soil dust sample, albeit with a slightly higher than one order of magnitude overall range of values measured by all methods at any particular temperature. A somewhat steeper inflection in data near -20°C is noted in this case, which may exacerbate discrepancies between methods due to temperature uncertainties alone. Within the immersion freezing methods sharing the impinger sample, variance increases from a factor of a few to more than an order of magnitude at colder temperatures. Two FRIDGE-IMM samples were collected 15 minutes apart for this experiment, and demonstrate results that span the range of INP concentrations measured by all methods at temperatures near -20°C. The two FRIDGE-IMM filter samples also bracket the results from the IS filter collection that spanned the same time frame as the impinger sample. At the coldest temperatures examined, a separation develops between the directly sampled (higher) and post-processed (lower) INP concentration ranges. And as for the Argentinian soil sample, the PIMCA-PINC data cap the direct processing instrument measurements of Tunisian soil dust at the coldest temperatures, leading to nearly a two order of magnitude discrepancy of measured INP concentrations at below -32°C. Thus, this experiment is consistent with the experiment for Argentinian soil dust particles in showing good agreement amongst INP measurements, but with the largest uncertainties typically occurring at the very warmest and coldest ends of the temperature spectrum, where ice nucleation activity is lowest and highest, respectively.

Although both of the soil dust examples show a sigmoidal ice nucleation activation temperature spectrum, this is more pronounced for the less "desert-like" sample from Argentina. This likely reflects the activity of different sized particles and the presence of multiple INP types or ice active sites, with the warmest freezers possibly from proteinaceous and other heat labile organic INPs achieving a plateau of activation at temperatures warmer than -20°C (see, e.g., O'Sullivan et al., 2014; Hill et al., 2016; Beydoun et al., 2017). This sigmoidal behavior is also seen in the upper bound of precipitation water immersion freezing spectra (Petters and Wright, 2015) and in natural particle samples collected over arable soil regions, where it has been attributed to soil and plant emissions (Delort and Amato, 2018). The levelling-off of the ice nucleating activity at low temperatures is also similar to desert dust laden air observed around Cape Verde (Price et al., 2018).

For the conditions of overlap between directly sampled and post-processing methods (<-20°C), neither of the soil dust examples tested show the types of discrepancies noted for the mineral illite NX (Emersic et al., 2015; Hiranuma et al., 2015; Beydoun et al., 2016). Co-location of instruments, limitation of the size range of particles collected, and sharing of common samples collected simultaneously onto filters or into liquid may have all contributed to the consistency of results for natural soil dusts in this study. If true, it does not mean that different methods will agree in

atmospheric measurements, but rather that the differences that do occur are influenced by other sampling limitations (e.g., sizes of particles that can be assessed, etc.) (Burkert-Kohn et al., 2017; DeMott et al., 2017). This would also apply to intercomparisons in which laboratories in different places are free to dispense samples independently prior to comparison. In addition, it may be the case that the soil dusts examined in this study have specific features for activation that differ from minerals or proxy dusts like illite NX, and these are less influenced by water immersion and storage either cool or frozen.

Results for illite NX as a test aerosol will be addressed below in the discussion of sampling experiments directly from AIDA and comparison of all results by active site density. K-feldspar was examined as an additional example of a mineral aerosol in this study. This K-feldspar sample is referred to as FS02, as described in Atkinson et al. (2013) and Peckhaus et al. (2016), and has similar ice nucleating activities to other K-feldspars with microtexture (Whale et al., 2017). A comparison of INP concentrations in Fig. 6 shows similar results as for the soil dust samples, but the INP activation curve of K-feldspar particles is much steeper with a pronounced levelling-off below about -25°C (i.e., it reaches a maximum and is only weakly dependent on temperature). This steepness is associated in this case with an up to two order of magnitude spread among bulk sample immersion freezing methods at around -25°C, greater than for the natural soil dusts at this temperature. This may be partly explained by temperature uncertainties, which range from ±0.2 to ±0.5°C for the immersion freezing methods (see specific Supplement sections for each device). Confidence intervals are also seen to be relatively large in this case, probably reflecting the high sensitivity of freezing to temperature. The leveling-off of INP concentrations below -25°C is consistent with previous measurements for K-feldspars summarized in Harrison et al. (2016) and Niedermeier et al. (2015). The separate filter sample (IS and FRIDGE-IMM) results are again consistent with those from instruments that shared the same impinger sample, although falling mostly to the upper side of these other measurements. A potential difference in this case is the time that particles may have spent stored in water, as the IS and FRIDGE-IMM results were presumably processed immediately after placing particles into liquid, thus minimizing time for flocculation of the clay. Most direct sampler results in Fig. 6 show INP concentrations that are consistent with the upper bounds of post-processed immersion freezing measurements at below -20°C. Exceptions are the SPIN-MIT instrument data, elevated at -21.3°C, and the PIMCA-PINC data elevated at colder temperatures, as was seen for PIMCA-PINC data in the Argentinian and Tunisian soil dust experiments.

The experiments for soil dusts and minerals do not offer comprehensive comparisons of the consistency of all of the different measurement methods at cloud temperatures warmer than about -20°C. Using a more active INP type within this temperature regime, Snomax® bacterial INPs, offers the opportunity for such assessment, as shown in Fig. 7. The unique activation properties of these INPs suggest separating the discussion around two temperature regions of Fig. 7; for temperatures warmer and colder than about -9°C. Online measurements were only obtained at temperatures colder than -9°C, where the ice nucleation activity is found to be maximized and only weakly dependent on temperature. Thus, biases should be solely due to uncertainty in derived INP concentration in this colder temperature regime. The excellent agreement in INP concentrations between all direct and post-processed methods suggests biases of at most a factor of 5 in this case.

All bulk immersion freezing methods capture the strong rise in activation due to the presence of the most active biological ice nucleators, those within the realm of the Groups I and II as defined by Yankofsky et al. (1981). This is expressed as a pronounced shoulder in all freezing spectra at temperatures warmer than -8°C in Fig. 7. We may note here that all bulk freezing methods shared the same impinger sample in this case, including the IS. This warm temperature shoulder of ice nucleation activity has also been demonstrated by Wex et al. (2015), Budke and Koop (2015), and Polen et al. (2016). Nevertheless, discrepancies are most strongly apparent in this region where these larger and more fragile aggregates of ice nucleating proteins are responsible for the ice nucleation activity. Measurement discrepancies across all immersion freezing methods are seen to range from one to four orders of magnitude (up to 4°C equivalent difference), increasing toward the warmest temperatures in this temperature region in Fig. 7. This appears to result largely from a bifurcation of freezing behavior of the (warmest) first-freezers in multiple freezing scans of the thawed CMU-CS impinger sample, and a similarly strong increase in the activation of first freezers in a few NIPI scans that were processed without prior storage as a frozen sample (i.e., processed immediately after collection, the only group to do so), including following dilution of the sample performed in order to access colder freezing temperatures for droplet arrays. The CMU-CS results in two of four scans appear to reflect the instability of Group I freezers noted in previous studies, possibly dependent on the time delay involved in conducting a freezing experiment following thawing of the impinger sample (Polen et al., 2016). How the individual freezing assays were separated for averaging is described in the Section S.2.2. In summary, the strong variability in activity seen the warmest activation temperature regime for Snomax® particles brings into question the ability to utilize the warmest temperature (> -10°C) freezing behavior of Snomax® reliably for calibration purposes. This has been noted previously by Polen et al. (2016), and attributed to batch-to-batch variability and the loss of activity following long-term ~~in~~ storage~~)~~. The freezing behaviors at colder than -10°C are quite stable, and a simple conclusion from this experiment is that there is no fundamental limitation or apparent bias in the ability of any method to measure immersion freezing activation in the modestly supercooled temperature regime warmer than -15°C versus below -20°C, at least for detecting biological INPs in relatively high numbers. Hence, if disagreements occur between direct and post-processing methods in this temperature regime, one possibility is that such disagreement relates to the impact of immersion in water on ice nucleation activity for certain particle types whose morphology can be altered in water (e.g., Grawe et al., 2016) and/or other differences in activation of single particles by direct processing methods versus particle populations placed into bulk water, sometimes stored frozen for later processing.

**3.2 Sampling of INPs from the AIDA chamber and comparison to subsequent AIDA freezing results**

Data collected in coordination with AIDA experiments provided additional intercomparisons. These data were first analyzed as active fractions, which is the number fraction of all particles freezing when normalized to the total number concentrations of particles (potential INP) present at the onset of expansions. Because of the large volume of the AIDA chamber, only modest differences in aerosol particle concentrations existed in the time prior to expansion start. Despite more limited participation, most direct measurement systems and a few diffusion chamber systems (using collected filters or substrates) processed particles in these experiments. Generally lower INP number concentrations in AIDA limited any chance that INP number concentrations achieve values that might lead to vapor

depletion in the continuous flow instruments (Levin et al., 2016). Use of active fraction allowed for inclusion and comparison of data from multiple AIDA experiments, and from APC experiments, to examine for consistency and repeatability. As discussed in Section 2.4, the active fraction data could be readily converted to active site density.

In Fig. 8, results are included for the illite NX, for which comprehensive experiments in the APC were not examined in Section 3.1. Figure 8a shows active fraction data from various ice nucleation instruments in multiple AIDA experiments (listed in Fig. 8 caption and Table S1), and from the ice concentrations measured in subsequent AIDA expansions. These results for illite NX show a scatter of INP active fraction at selected temperatures of more than two orders of magnitude, consistent with Hiranuma et al. (2015). Data from two FRIDGE-STD wafer collections demonstrate a variability factor of several fold despite collection of the particles in close temporal proximity. This could reflect the negative impacts of excess particle loading on causing water vapor depletion in the diffusion chamber as freezing ensues, limiting full activation at 1% supersaturation. Consistent with such an assumption, we note that sample 73 (lower set of FRIDGE-STD data points at -25 and -30°C) had four times the volume of sample 74 (higher active fraction data points at -25 and -30°C). The convergence of the FRIDGE-STD results toward the FRIDGE-IMM results is also noted for the latter sample. This issue of determining the suitable volume of air for collection for substrate ice nucleation studies given its dependence on the concentration of INPs has been recognized for many years. Nevertheless, correspondingly low active fractions at -25°C are also measured by PINC, a flow diffusion chamber that should have no issues with water vapor depletion. We may note, however, the strong sensitivity to processing supersaturation in flow diffusion chambers for sampling illite NX particles as reflected by the CFDC-CSU results at 105% and maximum $RH_w$ prior to onset of water droplet breakthrough. We may further note that AIDA activated ice number fractions for illite NX are bracketed by these CFDC-CSU results. These results are consistent with the findings of DeMott et al. (2015) for certain natural and desert dusts, suggesting that underestimates of INP concentrations active in the water supersaturated regime where immersion freezing sometimes dominates in CFDCs could be a general feature, also discussed by Garimella et al. (2017). A strong sensitivity of illite NX to $RH_w$ may be partly responsible for the wider range of INP active fraction for diffusion chambers in this case, and the shorter residence time of the PINC instrument may contribute to its lower estimate in comparison to other continuous flow chambers at certain temperatures. Each continuous flow chamber also may stimulate different responses in regard to $RH_w$ sensitivity, dependent on a variety of factors in addition to residence time that may include the evaporation section control. These things may require special study for the flow diffusion chambers as a group.

Data are plotted as $n_{s,geo}$ (T) values in Fig. 8b, and parameterizations developed from the study by Hiranuma et al. (2015) are overlain. This demonstrates that the single DFPC-ISAC chamber data point at -20°C and the upper bound values of FRIDGE-STD measurements are consistent with the $n_{s,geo}$ curve (log-space version shown as long-dashed curve) function found to represent immersion freezing measurement data by Hiranuma et al. (2015). A fair amount of the direct sampling instrument data from AIDA sampling is also consistent with this function. Nevertheless, it is also the case that some portion of the these instrument data, particularly the CIC, CFDC-CSU measurements at maximum water supersaturation, AIDA expansion results, and all lower temperature flow diffusion chamber data including the PIMCA-PINC, generally align with the Hiranuma et al. (2015) parameterization for results obtained from dry

dispersion measurements in that study (short-dashed line in Fig. 8b). The FRIDGE-IMM results at warmer temperatures appear as the outlier, splitting the two parameterizations at the warm end of measurements.

When $n_{s,geo}$ $(T)$ values derived from APC experimental results on illite NX particles are added in Fig. 8c, it becomes clear that most post-processed immersion freezing results in the present study align quite well with the parameterization of previous immersion freezing results from Hiranuma et al. (2015). Furthermore, it is seen that the DFPC-ISAC and FRIDGE-STD results, and the lower range of CFDC-type measurements (PINC, CFDC-TAMU) are most consistent with the immersion freezing data. However, we note the addition in Fig. 8c of CFDC-type measurements from APC experiments, including data from the CFDC-CSU, INKA, and SPIN-TROPOS instruments, which trend toward the dry suspension parameterization from Hiranuma et al. (2015). Most strikingly, these data, while limited to a few additional experiments, support the extension of this dry suspension relation to temperatures near -20°C, with the consequence that a three-order magnitude or more discrepancy occurs between direct and post-processed measurements at this temperature. The data noted in blue for CFDC-CSU and INKA were the only data collected in the March 13 experiment. Nothing peculiar stands out for the aerosol generated on that day, with sizes that did not reach close to ice crystal detection sizes. Hence, the bifurcation of INP behaviors of illite NX at different times and potentially by different methods are confirmed in the present study, and with no special new insights as yet into their source nor of the relevance of these discrepancies as a potential concern for atmospheric INP measurements. While proposed as an atmospheric dust surrogate, the ice nucleation behaviors of illite NX assessed by different methods contrast with the general equivalency of measurements of INP behaviors of natural soil dusts found in this study.

Agreement of methods for measuring the INP activity of Snomax® particles as shown in Fig. 7 is repeated in the AIDA experiments, as shown by $n_{s,geo}$ $(T)$ calculations presented in Fig. 9a. Ice active site densities derived from fractional activation and particle surface areas in the AIDA experiments (listed in Fig. 9 caption) fall to the high side of other direct processing measurements, but only by a modest factor of no more than a few, and within experimental uncertainties. Little difference is seen between CFDC-CSU results at 105% or the maximum $RH_w$ achieved before water droplet breakthrough. As well, there is no discrepancy seen between the FRIDGE-STD and other results. This may be because Snomax® INPs have been observed to achieve their maximum activated fraction by 100% $RH_w$ at temperatures below -10°C (DeMott et al., 2011), and so no strong artificial supersaturation dependence occurs. Prediction of $n_{s,geo}$ $(T)$ on the basis of the $n_m$ $(T)$ (active site density per unit mass) determined in the Snomax® particle ice nucleation studies of Wex et al. (2015) is also presented in Fig. 9a. This conversion uses $n_m$ $(T)$ as given in Eq. 6 of Wex et al. (2015), divided by the surface area to mass concentration ratio, following Eq. 3 from Hiranuma et al. (2015). A surface area to mass concentration ratio value of 7.99 $m^2$ $g^{-1}$ was derived from the Snomax® particle size distribution measurements made in association with the AIDA results reported in Wex et al. (2015). Particle generation methods for Snomax® used in the present AIDA experiments were identical to that prior study. It is seen that although the peak predicted $n_{s,geo}$ $(T)$ values exceed the values measured by most methods in this study, it is by only a small amount. Since this demonstrates close consistency of the present experiments with past Snomax® experiments, we did not pursue the exercise of re-deriving the surface to mass concentration ratio particular to the FIN-02 studies. APC data were used to derive $n_{s,geo}$ $(T)$ in Fig. 9b. This demonstrates repeatability during the FIN-02 studies for assessment

of the ice nucleation activity of Snomax® INPs and a level of consistency with prior results that suggests the potential suitability of Snomax® as a bacterial INP calibrant surrogate, albeit with the mentioned caveats on the instability of detection of first freezers at the warmest temperatures.

AIDA experimental results converted to $n_{s,geo}$ (T) for Argentinian and Tunisian soil dusts are shown in Fig. 10a. As expected, the range of site density measured by the continuous flow chambers prior to expansion, and based on AIDA ice activation measurements during expansion, mimics a similar spread in INP number concentrations observed by all measuring systems in sampling from the APC. We also note the agreement between AIDA ice crystal activation in cloud parcel simulations and the INP measurements from the portable instruments in these two cases at near -25°C. The non-continuous-flow diffusion chamber results from these AIDA chamber experiments fall moderately to the low side of $n_{s,geo}$ (T) values for these dusts. The two dusts have similar activation properties at below -20°C, and the range of $n_{s,geo}$ (T) is at least partly consistent with multiple natural soil dust $n_{s,geo}$ (T) parameterizations, including O'Sullivan et al. (2014) for "fertile soil dust", Tobo et al. (2014) for "Wyoming soil dust" and Steinke et al. (2016) for "agricultural soil dust". Derived $n_{s,geo}$ (T) based on the APC experiments on Argentinian dust (Fig. 4) are overlain in panel b, and $n_{s,geo}$ (T) derived from APC Tunisian dust experimental data (Fig. 5) is overlain in panel c of Fig. 10. In contrast to larger discrepancies found for illite NX, $n_{s,geo}$ (T) results for both Argentinian and Tunisian dust shown in Fig. 10 demonstrate much greater consistency. Larger discrepancies occur only at the coldest temperatures, where the PIMCA-PINC measurements of direct freezing of single particles within droplets diverge to much higher values than most of the immersion freezing measurements. This is especially the case for the Tunisian dust results, where $n_{s,geo}$ (T) based on the maximum $RH_w$ INP data from the CFDC-CSU do not clearly align with the PIMCA-PINC results in the same manner that they do for Argentinian dust at colder temperatures. The WISDOM data also diverge strongly from other immersion freezing data at colder than -25°C. Finally, we may note that the $n_{s,geo}$ (T) results for the more loamy Argentinian dust align quite well with values predicted from previous studies of arable soil dusts in the studies of O'Sullivan et al. (2014) and Tobo et al. (2014), but not well with those predicted from Steinke et al. (2016). The Tunisian dust results in Fig. 10c show less consistency with the fertile soil dust parameterization, which may be expected due to the more arid nature of the Tunisian sample.

Finally, $n_{s,geo}$ (T) results for sampling K-feldspar particles from AIDA prior to expansions are shown in Fig. 11a, and the same data are overlain with APC data for K-feldspar in Fig. 11b. Additionally, an $n_{s,geo}$ (T) parameterization is added on the basis of the $n_{s,BET}$ (T) fit to immersion freezing ice nucleation data published by Atkinson et al. (2013), where the BET refers to the fact that the surface area employed is based on Brunauer–Emmett–Teller (BET) gas adsorption data rather than an estimate of geometric surface area. To convert the parameterization, we use the laser diffraction-based surface-to-mass conversion factor of 0.89 $m^2$ $g^{-1}$ determined by Atkinson et al. (2013) and the specific BET surface area measured for the samples used in this study of 2.6 $m^2$ $g^{-1}$. Hence, the normalization factor is 2.6/0.89. While all of the data parallel the Atkinson et al. (2013) parameterization, agreement with it quantitatively is seen for selected direct sampling instrument data and the limited AIDA data available for which water saturation was achieved in expansion tests for FS02. Exceptionally large spread in inferred $n_{s,geo}$ values occurs at -20°C. Note here that only the 105% $RH_w$ data was usable for the CFDC-CSU and INKA instruments in this case because of an issue that was associated with and exacerbated by the steep activation curve of K-feldspar at this temperature. In

particular, it was seen that very steep $n_{s,geo}$ *(T)* led to the appearance of small ice crystals in the optical particle counter spectra at just above the 3 μm size used to separate smaller liquid from larger ice particles. This is an unusual feature for this type of device, with nucleated ice crystals typically growing to larger optical channels (sizes), and it likely reflects the late freezing of liquid particles as they were evaporating and cooling upon entry into the evaporation region of the instruments. This possibility is unique to the present configuration of the CSU CFDC due to the adjustment of the walls in evaporation region to match the inner (cold) wall temperature. This issue could similarly be realized in any diffusion chamber if there is a "cold point" anywhere along the flow path. Consequently, the SPIN-MIT data shown were reprocessed to report their data at the coldest wall temperature measured in the instrument growth region. Further discussion of this issue is provided for the CFDC-CSU in the Supplement to this paper (Section S.1.2). An additional (red) data point is shown in Fig. 11 for the CFDC that is considered erroneously attributed to the activation temperature near -20°C, even though it aligns close to the AIDA chamber data. In this case, it was found that the rate of $RH_w$ change during scanning from lower to higher values was too fast, and exacerbated the over-estimation of INPs on the basis of OPC particle size. We note that the INKA instrument used a larger channel (size) to count INPs, and at the reported water supersaturation 4%, smaller the ice crystals were not being counted. Hence, the ice size channel might have been redefined for the CSU instrument in order to report additional data, but we choose here to use the data instead to make a point about instrument design considerations. As a final note, it should be understood that scans of $RH_w$ are not a typical operational practice when collecting atmospheric data. In this case, constant $RH_w$ or step-wise values are used.

While the various $n_{s,geo}$ *(T)* data trend well overall with the previous parameterization for FS02 particles, correspondence amongst results in Fig. 11 is not as good as for the soil dust samples in the -20 to -25°C range, and more resemble the spread of results for illite NX, with separation of $n_{s,geo}$ *(T)* of up to three orders of magnitude. Again, the variance amongst measurements follows the steepness of the INP activity versus temperature. The steep ice activation function of K-feldspar in the region from -15 to -25°C has already been noted in Fig. 6. INP activity rises at least $10^6$ times over the 10°C for K-feldspar in this range, whereas the steepest rise for the natural soil dusts is $10^3$ to $10^4$ units per 10°C. For illite NX the activity rises about $10^5$ per 10°C. Thus, modest differences in temperature, or their control within instruments, equate to large differences in ice activation for K-feldspar.

### 4 Summary and Conclusions

Through careful coordination and collaboration in a laboratory setting, most of the objectives of the second phase of the Fifth Ice Nucleation Workshop were strongly advanced if not fully achieved, and the existence of the data set should continue to serve explorations of measurement consistencies and issues for applying different techniques in isolation or in tandem for making atmospheric ice nucleation measurements. Extensive comparisons involving a large number of teams and using multiple INP types were made within just a three-week workshop. Some operational issues occurred for investigators at times (obvious errors, measurement biases, inability to achieve comparative conditions for proximal immersion freezing) and where these were recognized, data were either not entered into comparisons or in a few cases were revised. Some issues were investigated, such as the appearance of small ice in the CSU CFDC data for INPs with steep activation functions. Others remain the subject of active investigation.

We may summarize the workshop results generally around the stated objectives as follows:

1) *Compare ice nucleation measurement systems for conditions considered to be equivalent as much as possible, across a wide dynamic temperature range, including temperatures warmer than -15°C.*

To simplify this first analysis of FIN-02 data, a focus was placed in this paper on immersion freezing nucleation and activation within continuous flow chambers in the water supersaturated regime, across a wide temperature range including temperatures warmer than -15°C through the use of bacterial INPs in selected experiments. The proximal behavior model for comparing immersion freezing by direct processing instruments versus bulk immersion freezing methods worked reasonably well, excepting cases noted later in this summary. Very good correspondence was obtained between many measurements for soil dusts and bacterial INPs, both amongst instruments that directly-processed single particles and those that post-processed bulk aerosol collections for assessing immersion freezing INP concentrations (Figures 4, 5, 7). Agreement of INP number concentrations and geometric active site density within less than about 1 order of magnitude was achieved under most circumstances analyzed herein for these three materials. This was strictly demonstrated for both direct and post-processed samples over a more limited temperature range, approximately -20 to -30°C for the soil dusts and -10 to -30°C for the bacterial INPs.   For these atmospherically-relevant particle types, no strong biases between the two basic types of measurement systems were evident in this range of overlap.

The fact that agreements were quite good overall in this study may have been strongly assisted by the combination of co-sampling the same aerosol particle sources in the same laboratory, sharing similar collected aerosol samples, and limiting the largest particle sizes assessed in workshop experiments to those that could readily be measured by all techniques. The nature of active sites for the various INPs examined may also have influenced comparability of direct particle sampling versus post-processed bulk collection. Consequently, it appears that soil dust particles are much more equally assessed for INP content than some minerals and mineral mixtures, and may better serve as potential calibration INPs. This was supported by the worst agreement between methods, up to three orders of magnitude, for illite NX and the FS02 samples that have a very steep activation spectra versus temperature, which exacerbates disagreements that otherwise represent only a few degrees of temperature change. In the case of illite NX, discrepancies seen in Hiranuma et al. (2015) were reproduced at temperatures warmer than -25°C. The steep activation behavior of the FS02 also led to the finding that when sampling such INP types, cooling to achieve evaporation in the exit section of a CFDC (CFDC-CSU, CFDC-TAMU and INKA in this study) can express "late" activation of ice crystals that remain at small sizes and should not be attributed to the set point temperature of the instrument growth. This may be an issue primarily for laboratory measurements of such INPs, since most natural INP T-spectral slopes are lower than for many of the samples tested, often only approximately 2 orders of magnitude per 10°C, rather than 5 orders or more per 10°C (DeMott et al., 2017; Price et al., 2018).

Assessment of agreement between direct processing of single particles and post-processing measurement systems was mostly only possible below -20°C since flow chamber devices have a limit of detection which restricts measurements at warmer temperatures.  The exception is in cases where the higher concentrations of bacterial INPs were assessed. Since biological/biogenic INPs are the most likely contributors to freezing at modest supercooling (e.g., Murray et al., 2012; Hoose and Möhler, 2012), it would seem valid that combining bulk aerosol sampling

measurements to capture INPs at very modest supercooling with direct measurements extending to colder temperatures within the same atmospheric study will lead to a reasonably valid representation of immersion freezing INPs (e.g., DeMott et al., 2017; Welti et al., 2018).

2) *Gain insights into how detection of ice nucleation is influenced by the specific configuration of similar measurement systems.*

Among measurements on samples collected for post-processing, there was no particular or consistent bias between different approaches to bulk suspension measurements. Furthermore, there appears to be little discrepancy between measurements made with particles collected directly into liquid versus collection onto filters followed by resuspension into liquid. There also appear to be no discernable impacts of freezing samples versus processing them immediately, on the basis of the μl-NIPI versus other methods apart from impacts on the warmest temperature freezing of bacterial particles (e.g., results from Snomax® experiments). Factors affecting reproducibility, such as accuracy of temperature attributed to sample freezing and instability of the warmest bacterial INPs, are the most important factors affecting the agreement between methods, which often spans an order of magnitude overall. Most measurement groups have likely performed careful assessments of their temperature measurements attributed to droplet volumes, but there is evidence that errors may occur due to the inability to perfectly assess temperature at the point of freezing (Beall et al., 2017).

For diffusion chamber measurements of collected particles, the need for awareness of volume effects on processed INPs remains as a requirement. Results in a few cases showed these measurements to fall to the low side in assessing immersion freezing nucleation. It may be necessary to collect varied volumes to assure that particle loading in different cases is not influencing accurate assessment of INP number concentrations.

Differences between INP measurements in the water-supersaturated regime by continuous flow chambers were seen, and these differences likely relate to the need of these systems to achieve higher than expected $RH_w$ in order to fully activate aerosols to facilitate their subsequent immersion freezing on the full particle population within the diffusion chambers (DeMott et al., 2015; Garimella et al., 2017; Burkert-Kohn et al., 2017). These instruments may universally have an issue in focusing aerosol particles reliably into the center of the imposed $RH_w$ field, among other factors that depend on particle types, including their hygroscopicity and ability to activate ice nucleation already in the sub-water saturated regime (not discussed in this paper). Solving the issue(s) involved could provide the guidance on correcting these data for the $RH_w$-sensitivity factor present in the water supersaturated regime for all of these devices. Different systems have varied ability to achieve higher $RH_w$, depending on the different water breakthrough $RH_w$ as imposed by device design (see Section 3 and Section S.1.2). For example, it was noted that the PINC instrument more commonly measured INP concentrations at the lower range of the flow chamber devices, which may be attributable to its shorter residence time. These systems will continue to be used in this manner to measure atmospheric INP activation, but will struggle to equivalently capture activation to the same degree until issues are solved. Such solutions could involve redesign of how samples are introduced to the chambers. This is clearly deserving of a special study, which was beyond the scope of this workshop. Study of the use of different evaporation region temperatures also merits attention as it may impact detection of ice formation at higher $RH_w$. Limitations on assessing the impacts of larger aerosols as INPs in continuous flow instruments will remain, unless special inlets are developed.

Of note in this study is the agreement between most direct sampling and post-processing measurements at the colder temperatures in comparison to the large discrepancies found in a recent study comparing measurements of ambient particles (DeMott et al., 2017). We believe that this is attributable to assuring comparability of measurement methods in FIN-02 by restricting particle sizes, as mentioned above. This likewise implies that discrepancies between direct and post-processing methods can be expected to occur in ambient sampling when larger particles are present, although the source of those discrepancies as true impacts (i.e., of larger particles acting as individual INPs versus breakup of INPs after time in bulk suspensions) must remain a topic of future research. Both bulk sample immersion freezing and proximal immersion freezing in the flow diffusion chambers sometimes underestimate freezing in comparison to the PIMCA-PINC single particle immersion freezing method. For CFDC type instruments, this is partly understood as the need to achieve much higher $RH_w$, sometimes practically unachievable, to effectively simulate and capture immersion freezing (previous paragraph), requiring corrections that were not applied in this study. Whether such corrections are the only reason for discrepancies with PIMCA-PINC require further investigations. Reasons why the bulk immersion freezing methods do not always agree with PIMCA-PINC may relate in some unresolved manners to factors at play during extended bulk immersion, such as breakup, sedimentation, and alteration of active sites. It would be helpful if the PIMCA-PINC method could be extended to lower active fractions and INP concentrations, but this appears to be a fundamental limitation of the phase discrimination technique.

3)    *Utilize different INP types to investigate if differences between instruments occur with these different types.*

The use of varied INP types was clearly vital in achieving the first two objectives summarized above. For example, the use of a highly active biological INP type clearly helped to demonstrate that there is no fundamental limit on INP detection by any method if limits of detection are met. It is not known if this conclusion is peculiar to the biological INPs, although these may be the most common and important type to detect in the warmer supercooled temperature regime. While the utility of Snomax® as a calibration INP was again demonstrated here, issues in achieving well-defined active fractions at temperatures above -9°C via post-processing of bulk collections for immersion freezing found in previous studies were repeated herein. Comparisons with direct processing methods was not obtained in this temperature regime, making this an important topic for future studies.

The use of both natural soils and mineral samples as INPs allowed for seeing that the soil INPs were more consistently measured within and across measurement methods compared to the mineral component K-feldspar and a material representing key mineral compounds of desert dust aerosols (illite NX). The steeper nucleation rate functions of the minerals were key to identifying the potential bias in production of ice nucleation in the evaporation sections of the CSU and INKA CFDC's, likely due to the additional cooling occurring there. Other devices that warm the airflow while reducing the relative humidity toward ice saturation to evaporate activated cloud droplets did not see small ice crystal production. Detection of the small ice crystals produced in this manner can be largely biased against through adjustment of the channel size used for ice detection in the optical particle counter, although mitigation through redesign may provide a more satisfactory long term solution. Further investigation of this issue is merited.

Through the use of multiple INP types, results from this workshop could also be compared versus previously published parameterizations. These comparisons were very encouraging for demonstrating reproducibility of

laboratory study results in general, further supporting the picture of general consistency of present INP measurements within identified uncertainties.

The FIN-02 archive will remain for additional scientific investigations, such as at least limited comparisons in the ice nucleation regime below water saturation (see below), analysis of experiments regarding homogeneous freezing and the role of particle pre-activation for ice formation. While the FIN-02 workshop objectives were generally achieved, a number of topical research needs remain and some recommendations are suggested.

- Although the coordinated sampling protocols during FIN-02 worked very well, and the possibility of establishing certain calibration standards was suggested, it is not practical for the majority of members of the international INP measurement community to gather with high frequency for such activities. It may be possible that similar correspondence of measurements can be obtained through the distribution of some standards and the use of defined aerosol generation protocol. A basic attempt at such an exercise that restricts sample types to a natural dust and bacterial INPs is worth exploring, as a wide distribution exercise has only thus far occurred for illite NX. The general correspondence of present workshop data with $n_s$ parameterizations derived in previous laboratory studies provides a positive outlook.

- Special investigation of detection of ice formation in the regime below water saturation remains as a need that will be only partially addressable with FIN-02 data due to somewhat limited range of temperatures assessed by most direct processing instruments. While the number of instruments involved in such an assessment is more limited, it is no less important for evaluation in regard to the use of ice nucleation instruments in the colder regions of the troposphere. For example, are similar results obtained or is there a wider discrepancy between shorter residence time diffusion chamber, substrate-based processing devices, and controlled expansion cloud chambers in this regime?

- Ice nucleation measurements are also needed for fundamental understanding of ice nucleation and of the nature of INPs, which an array of measurement devices can address better than a single technique. Questions remain on differences between condensation and immersion freezing (Vali et al., 2015; Burkert-Kohn et al., 2017), the nature of ice nucleation in the regime below water saturation (Higuchi and Fukuta, 1966; Marcolli, 2014), connecting practical measurements with molecular scale understanding and many other topics.

- Other focused studies involving instruments within direct and post-processing communities are recommended for addressing needs specific to these communities. For direct sampling, examples are a careful comparison of the operational characteristics of continuous flow instruments as a function of $RH_w$, and a rigorous comparison of use of optical size for detecting ice in CFDC-style instruments versus use of depolarization and machine learning methods. For post-processing methods, the role of sample storage aggregation and breakup as a function of particle loading and size in bulk immersion freezing studies deserves study.

- Establishing best practices for handling of bulk immersion freezing samples, and for limiting and correcting for the background freezing counts introduced in the water used for collection (for impingers) or for rinsing (of filters) is a topic that was not covered directly in FIN-02. Improvement and standardization of protocol

would only help to improve the good results obtained across these methods in this study. This is a topic of a separate paper published in this special issue (Polen et al., 2018).

- The role of INP size and more careful quantification of biases involved in assessing this factor deserves more focused attention. The use of monodisperse aerosols in a workshop like FIN-02 would present logistical challenges, but would add an important dimension for study and greatly assist interpretation of results.

- The low INP concentration regime still presents a strong challenge for the measurement community, one that becomes critically important in atmospheric studies. Low INP concentrations are ubiquitous at modest supercooling, but can also occur at lower temperatures in the atmosphere. For existing direct sampling devices like flow chambers, no current comparisons have focused on their abilities to control and correct for background frost artifacts. While ambient measurement campaigns such as FIN-03 allow some focus on this topic, a laboratory campaign could do the same. Only post-processing methods for immersion freezing can access the regime at modest supercooling. This limits temporal and spatial resolution, especially when sampling on aircraft. Can new methods be developed for directly assessing INP concentrations in larger sample volumes? Even modest improvement to direct methods to provide more overlap of measurement methods in the regime > -20°C would help to further evaluate the validity of meshing direct and post-processing methods to characterize INPs over the full mixed-phase temperature regime. Aerosol pre-concentration has been applied to extend the dynamic range of direct INP measurements in atmospheric studies (e.g., Tobo et al., 2013; Boose et al., 2016), but aerodynamic concentration methods bias against particle sizes much below 1 μm. Hence, it is worth investigating the possible use of other concentration methods applicable to the full aerosol size distribution, such as pre-condensation. Novel ideas are needed.

- Further comparisons for which the sampling groups are "blind" to the nature and concentrations of INPs being sampled could be useful toward giving confidence to the wider community that the INP measurement community is capable of recognizing issues and properly interpreting data. This will assist confidence and utility of larger global data sets. Such a comparison from FIN-02 will be reported on in a separate publication in preparation.

- Similar exercises as FIN-02 are also needed in sampling under ambient atmospheric conditions. This is the subject of the FIN-03 campaign that will be reported on separately.

Workshops such as FIN-02 will continue to play a large role in assessing measurement biases and ultimately improving the comparability of INP measurements made by a large community of researchers sampling on a global scale. The shared experience of these workshops is irreplaceable in providing special insights into the status of and issues involved in obtaining INP data in different scenarios that may be dominated by certain aerosol types. FIN-02 demonstrates that the INP measurement community remains on a progressive track towards assessing convergence between different methods used for INP quantification.

*Data availability.* Tables of all data used and plotted in this manuscript are included as Table S1 and Table S2 in the Supplement of this manuscript. These tables are also included with data archived in the KITopen data respository under doi:10.5445/IR/1000082906 (available upon publication). Other data are available upon request.

*Competing interests.* The authors declare no competing interests.

*Special issue statement.* This article is part of the special issue "Fifth International Workshop on Ice Nucleation (FIN)". It is not associated with a conference

*Acknowledgments.* The FIN-02 campaign was partially-supported by U.S. National Science Foundation Grant # AGS-1339264, and by the U.S. Department of Energy's Atmospheric System Research, an Office of Science, Office of Biological and Environmental Research program, under Grant No. DE-SC0014487. P. J. DeMott, E. J. T. Levin, and K. J. Suski acknowledge additional support from NSF Grant # AGS-1358495. The following authors acknowledge funding by the German Science Foundation (DFG) through the research unit FOR 1525 (INUIT): O. Möhler, N. Hiranuma, A. Peckhaus and A. Kiselev under MO 668/4-1, C. Budke and T. Koop under KO 2944/2-2, M. Szakáll and O. Eppers under SZ260/4-2, and H. Bingemer and D. Weber under BI 462/3-2. Y. Rudich acknowledges funding by the DFG Mercator fellowship. T.B. Kristensen acknowledges funding from the German Federal Ministry of Education and Research (BMBF) project 01LK1222B. A. Welti, P. Herenz, F. Belosi, G. Santachiara, J. Vergara-Temprado, H. Bingemer and J. Schrod acknowledge support funding for their research from the European Union's Seventh Framework Programme (FP7/2007-2013) project BACCHUS under grant agreement No. 603445. F. Belosi and G. Santachiara also acknowledge the Institute for Atmospheric and Environmental Sciences (Goethe-University Frankfurt) for support in filter collections. M. Burkert-Kohn was funded by grant no. ETH-17 12-1, ETH Zurich. H. Grothe, L. Felgitsch, and T. Seifried acknowledge funding from The Austrian Science Fund, FWF project number P26040. The University of Leeds team (T. F. Whale, H. P. Price, J. Vergara-Temprado, D. O'Sullivan, T. W. Wilson, and B. J. Murray) acknowledge support from the European Research Council (ERC, 240449 ICE; 632272 IceControl; 648661 MarineIce, 713664 CryoProtect) and the Natural Environment Research Council (NE/K004417/1, NE/I019057/1). S. D. Brooks, K. N. Collier, and J. Zenker acknowledge additional support from the U.S. National Science Foundation, Grant # ECS-1309854. M. J. Polen and R. C. Sullivan were supported by NSF Grants # CHE-1213718 and CHE-1554941, and M. J. Polen by an NSF Graduate Research Fellowship. M. D. Petters, S. S. Petters, and H. P. Taylor acknowledge additional support from NSF Grants # AGS-1010851 and # AGS-1450690. G. Kulkarni acknowledges support by the Office of Science of the U.S. Department of Energy (DOE) as part of the Atmospheric System Research Program. Pacific Northwest National Laboratory is operated for the U.S. DOE by Battelle Memorial Institute under contract DEAC05-76RL0 1830. Finally, all authors wish to acknowledge support from the AIDA team for preparing and operating the AIDA chamber, and supporting other operational logistics for this workshop.

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

**Table 1.** Direct processing INP instruments

| Instrument | Type | Institute | References |
|---|---|---|---|
| AIDA | Expansion cloud chamber | Karlsruhe Institute of Technology | Möhler et al. (2003); Möhler et al. (2005); Niemand et al. (2012) |
| CFDC-CSU | Continuous flow diffusion chamber (cylindrical) | Colorado State University | Rogers (1988); Rogers et al. (2001); Eidhammer et al. (2010) |
| CFDC-TAMU | Continuous flow diffusion chamber (cylindrical) | Texas A&M University | Glen and Brooks (2014); Zenker et al. (2017) |
| INKA | Continuous flow diffusion chamber (cylindrical) | Karlsruhe Institute of Technology | Schiebel (2017) |
| SPIN-MIT | Continuous flow diffusion chamber (parallel) | Massachusetts Institute of Technology | Garimella et al. (2016) |
| SPIN-TROPOS | Continuous flow diffusion chamber (parallel) | Institute for Tropospheric Research | Garimella et al. (2016) |
| CIC-PNNL | Continuous flow diffusion chamber (parallel) | Pacific Northwest National Laboratory | Friedman et al. (2011) and Kulkarni et al. (2016) |
| PINC PIMCA-PINC | Continuous flow diffusion chamber (parallel) Immersion mode adaptation | ETH-Zurich | Chou et al. (2011); Kanji et al. (2013); Kohn et al. (2016) |

5    **Table 2.** Post-processing INP instruments

| Instrument | Type | Institute | References |
|---|---|---|---|
| NCSU-CS | Cold stage droplet freezing array | North Carolina State University | Wright and Petters (2013); Hader et al. (2014); Hiranuma et al. (2015). |
| CMU-CS | Cold stage droplet freezing array in oil | Carnegie Mellon University | Polen et al. (2016); Beydoun et al. (2017) |
| KIT-CS | Cold stage droplet freezing array in oil | Karlsruhe Institute of Technology | Peckhaus et al. (2016) |
| μL-NIPI | Cold stage droplet freezing array | University of Leeds | Whale et al. (2015) |
| BINARY | Cold stage droplet freezing array in compartments | Bielefeld University | Budke and Koop (2015) |
| IS | Aliquot array freezing | Colorado State University | Hiranuma et al. (2015); Hill et al. (2016) |
| VODCA | Cold stage emulsion freezing | Technical University of Vienna | Pummer et al. (2012) |
| WISDOM | Microfluidics droplet freezing train | Weizmann Institute | Reicher et al. (2017) |
| M-AL | Acoustic levitator | University of Mainz | Diehl et al. (2014) |
| FRIDGE-STD | Low pressure diffusion chamber (Si wafers) | Goethe University of Frankfurt | Klein et al. (2010); Schrod et al. (2016) |
| FRIDGE-IMM | Cold stage droplet freezing array (on wafers) | Goethe University of Frankfurt | Hiranuma et al. (2015) |
| DFPC-ISAC | Dynamic filter processing chamber | ISAC-CNR | Santachiara et al. (2010); Belosi et al. (2014) |

**Table 3.** List of aerosol types, particle generation techniques and aerosol properties.

| Aerosol type[†] | Generator | AIDA Expt. ID | APC Expt. ID | BET specific sfc. ($m^2$ $g^{-1}$)[*] | Density ($g$ $cm^{-3}$)[**] | Dynamic Shape Factor[**] | Reference[§] |
|---|---|---|---|---|---|---|---|
| Illite NX (IS03) | Rotating brush (PALAS, RGB1000) | FIN02_4,10,22,25 | APC_1-4,7-8 | 124.4 | 2.6 | 1.3 | Hiranuma et al. (2015) |
| Argentinian soil dust (SDAr01) | Rotating brush (PALAS, RGB1000) | FIN02_5,9,24,26 | APC_9-10,20-21,29 | 13.1 | 2.6 | 1.2 | Steinke et al. (2016) |
| Saharan desert dust (SD6) | Rotating brush (PALAS, RGB1000) | FIN02_3 | APC_5-6 | 6.9 | 2.6 | 1.2 | Niemand et al. (2012) |
| Tunisian soil dust (SDT01) | Rotating brush (PALAS, RGB1000) | FIN02_7,12 | APC_11-12,27 | 7.0 | 2.6 | 1.2 | Lafon et al. (2006); Di Biagio et al. (2014) |
| K-rich feldspar (FS02) BCS376 | Rotating brush (PALAS, RGB1000) | FIN02_8,11,14 | APC_13-14 | 2.6 | 2.6 | 1.1 | Atkinson et al. (2013); Peckaus et al. (2016) |
| Snomax (SM04) | Atomizer (TSI, 3076) | FIN02_6,13,27 | APC_15-16,18-19,28 | N/A | 1.4 | 1.1 | Wex et al. (2015) |

[†]IDs in parentheses represent the AIDA-INUIT code names.

[*]The BET method to measure specific surface area of bulk powder is described in Hiranuma et al. (2014a). Note that our measurements have ± 5% uncertainty.

[**]For our geometric surface area estimations, we used the optimized effective densities and dynamic shape factors provided in this table.

[§]Bulk composition data of aerosols are available in these references. For SD6 and SDT01, XRD data are available upon request.

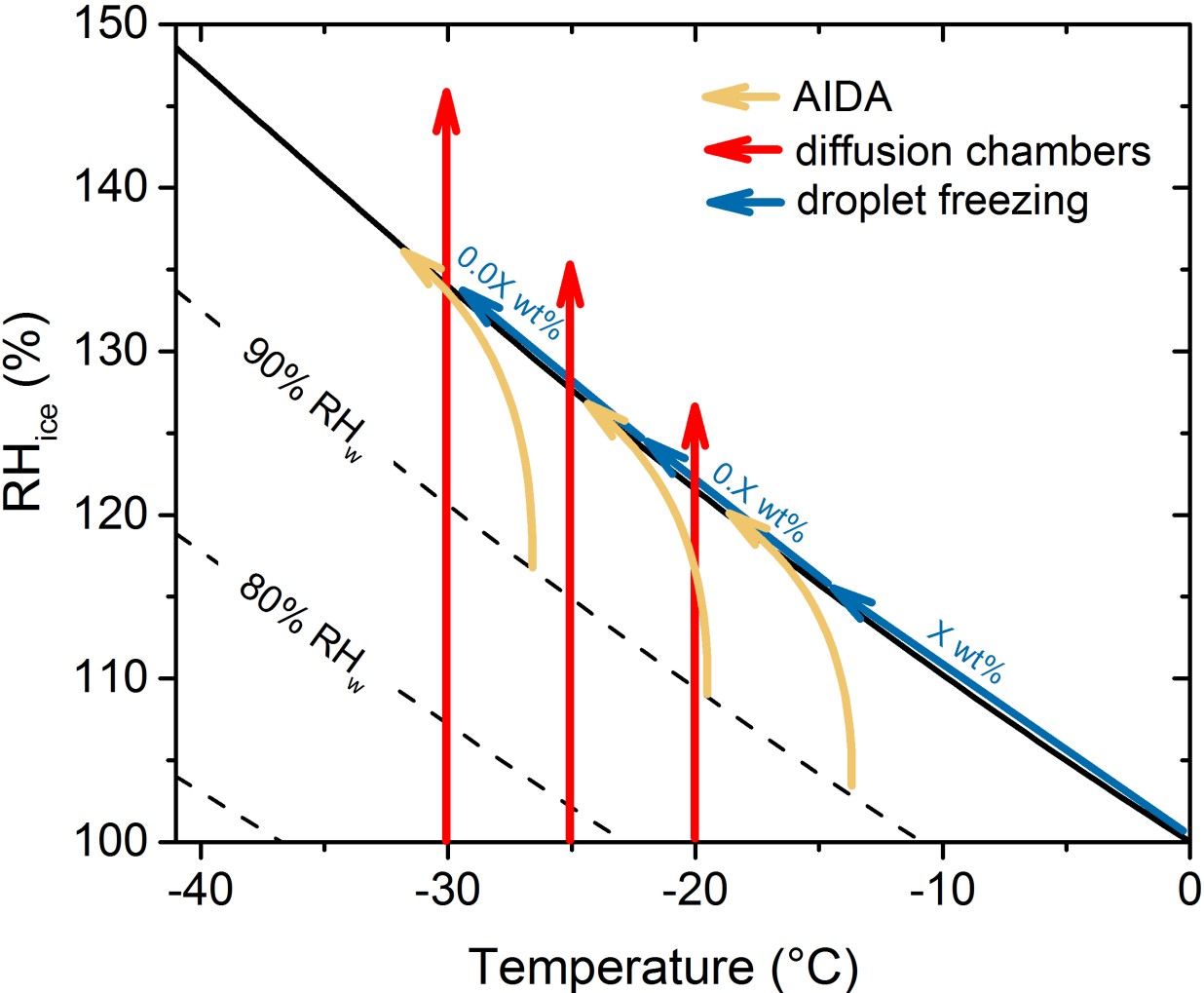

**Figure 1.** Schematic representation of different experiments conducted in the mixed-phase cloud temperature regime using different measurement systems during FIN-02. Yellow curves and arrows show the typical thermodynamic path of AIDA cloud chamber experiments. Red lines with arrows indicate the typical trajectory of direct particle sampling, continuous flow instrument systems and systems that process substrate-collected initially dry particle populations under controlled humidity and temperature conditions. The blue arrow following the water saturation line in $T$-$RH_w$ space shows the trajectory of subsequently diluted samples (generically and schematically referred to as X, 0.X an 0.0X weight percent suspensions) of collected aerosols measured by immersion freezing methods. Such dilution is required in many cases for the laboratory samples tested, but the need for dilution or not also depends on the droplet size/volume used. The PIMCA-PINC instrument follows the trajectory of the bulk aerosol immersion freezing devices, but does so for water droplets activated originally on single dry aerosol particles (and hence without the varied weight percent).

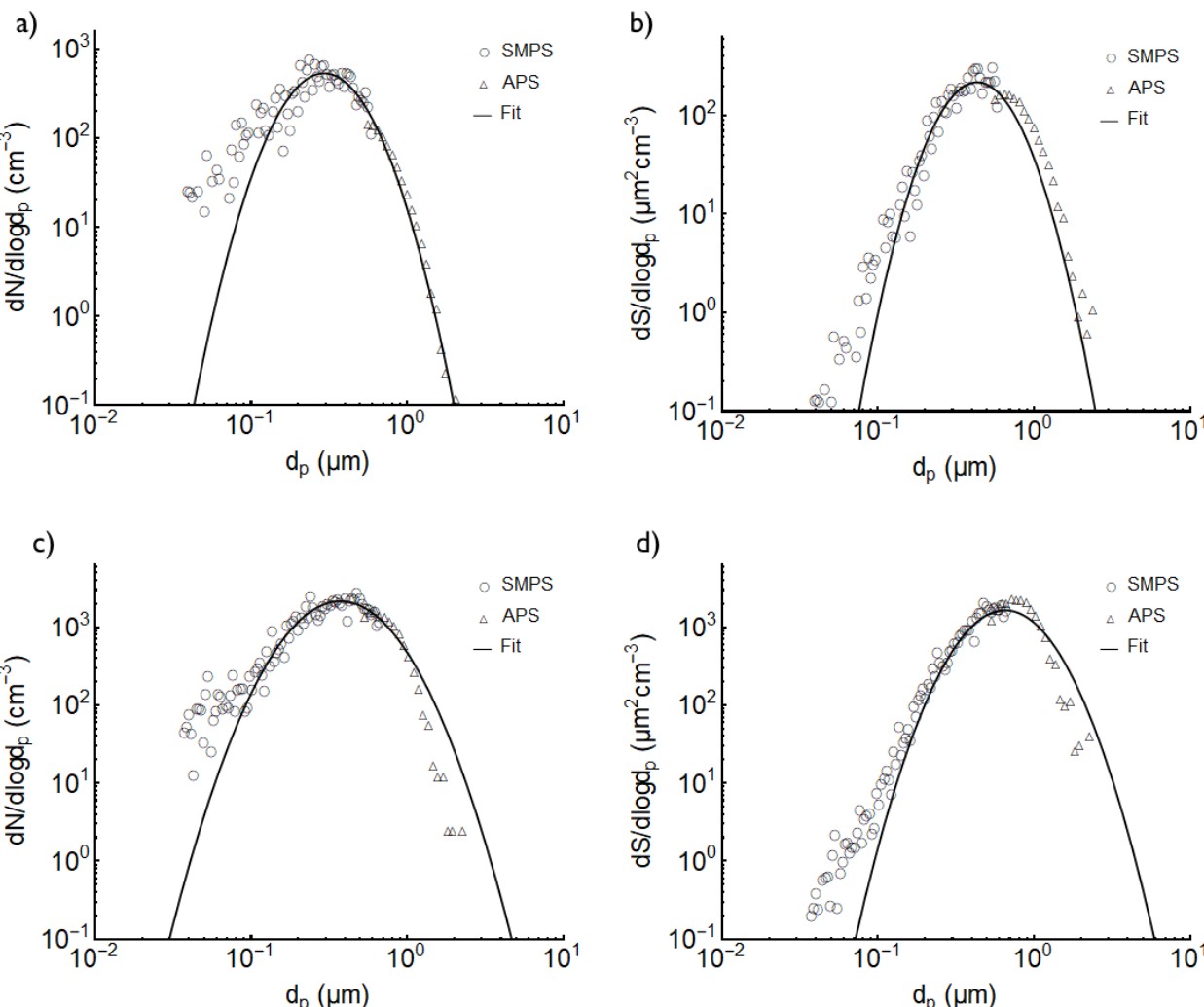

**Figure 2.** Number, in a), and surface area distribution, in b), of SDT01 dust particles generated into the AIDA chamber, measured with the SMPS and APS instruments, as well as the lognormal fits to the size distributions for exemplary AIDA experiment 12 on March 20, 2015, 648 s prior to expansion start. APS data have been converted based on assumed density (2.6 g cm$^{-3}$) and dynamic shape factor (1.2). The same data for SDT01 particles generated in APC experiment 12 on March 18, 2015, 785s following aerosol particle injection, are shown in panels c) and d). Lognormal fits are used to obtain total surface areas and these were tabulated as a function of time in each experiment.

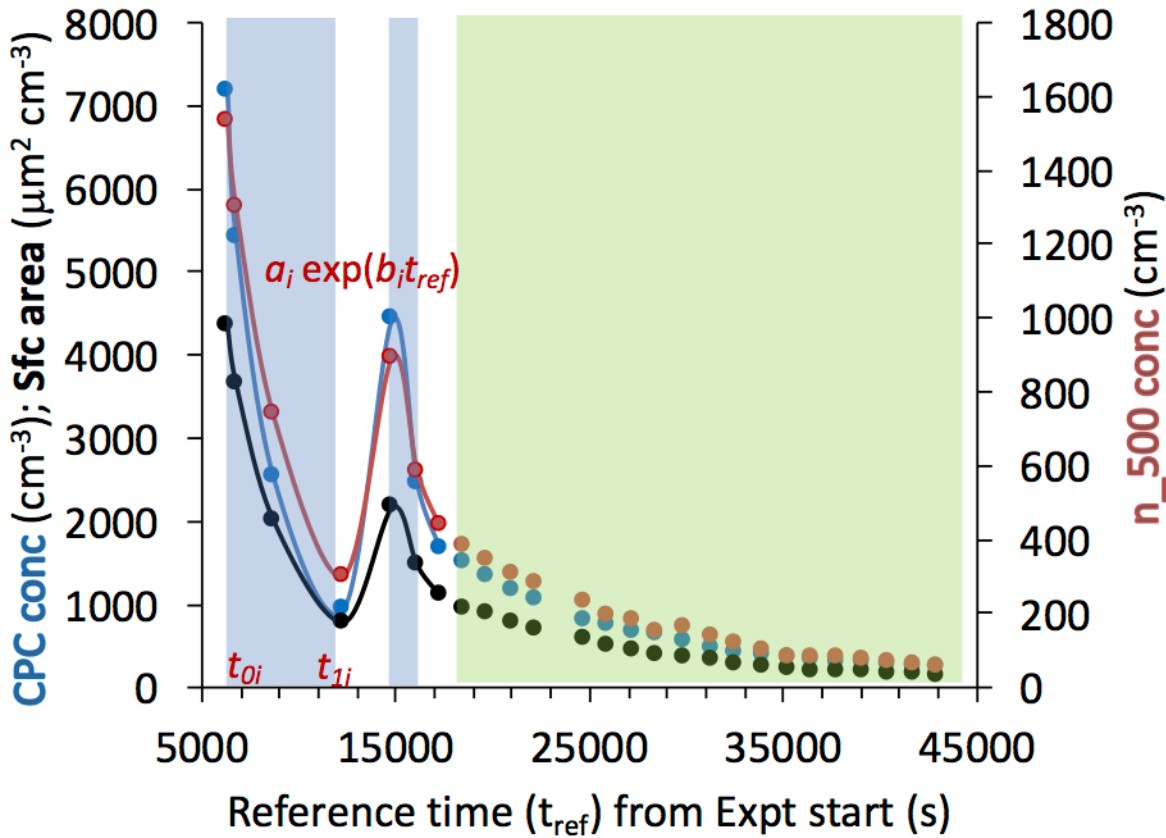

**Figure 3**. Data from the first half of a daily experimental series, after an initial fill of Tunisian soil dust (Experiment 11 and 12 of the FIN-02 APC series on March 18, 2015) into the APC at a time (time 0) taken to be the experimental start time. Impinger and filter sample periods after two separate chamber fills are highlighted in light blue shading. To obtain a reference aerosol concentration ($cm^{-3}$) via the CPC (total particles) (blue points, left scale) or optical particle counter at sizes larger than 500 nm (n_500; brown points, right scale), as well as surface area ($\mu m^2$ $cm^{-3}$) (black points, left scale) for the impinger/filter sampling period, the time-weighted average of the two sampling periods was determined. This period-integrated concentration value could then be used to ratio versus the concentrations of particles present at later sampling times during the direct processing instrument sampling of aerosols from the APC (green shaded region) in order to back-correct the directly-sampled INP number concentrations to those derived for the collections for post-processed samples.

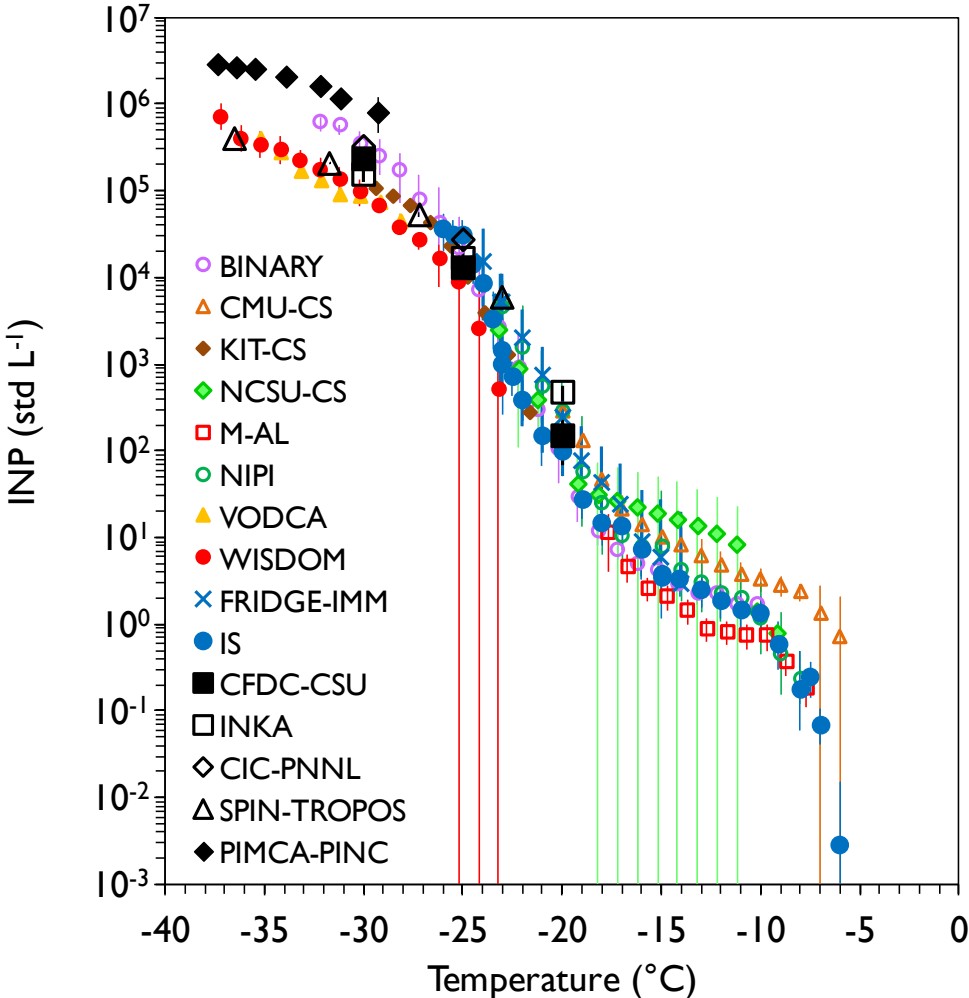

**Figure 4.** Combined results of all measurements of Argentinian soil dust sampled from the APC under conditions representative of those favoring immersion freezing nucleation in all devices. Note that most of the bulk immersion freezing data have been sub-sampled at ~1°C intervals. These immersion freezing methods are distinguishable here by their smaller data points (using different symbols and colors) over a broad range of temperatures, whereas direct sampling devices using conditions supersaturated with respect to water (immersion freezing assumed as a major contributor) at specific temperatures are indicated by sparse (larger) blue data points. The KIT-CS, NCSU-CS, NIPI, VODCA, M-AL, BINARY and WISDOM instruments shared impinger samples, while the IS and FRIDGE-IMM used separate filter collections (distinguished here by blue data points). As discussed in the text and Eqs. (1) - (3), corrections have been applied to real-time instrument data depending on the time of sampling, constituting a ratio of total particle concentrations present at the time of impingers/filter collections to those present at the time of specific flow chamber measurements. Error bars represent 95% confidence intervals, unless stated otherwise in the Supplement for each specific instrument. When error bars are not present, they may be subsumed within the marker (i.e., small errors) or alternately the binned point represents a single observation. The data herein correspond to FIN-02 APC experiments 9 and 10 on March 17, 2015.

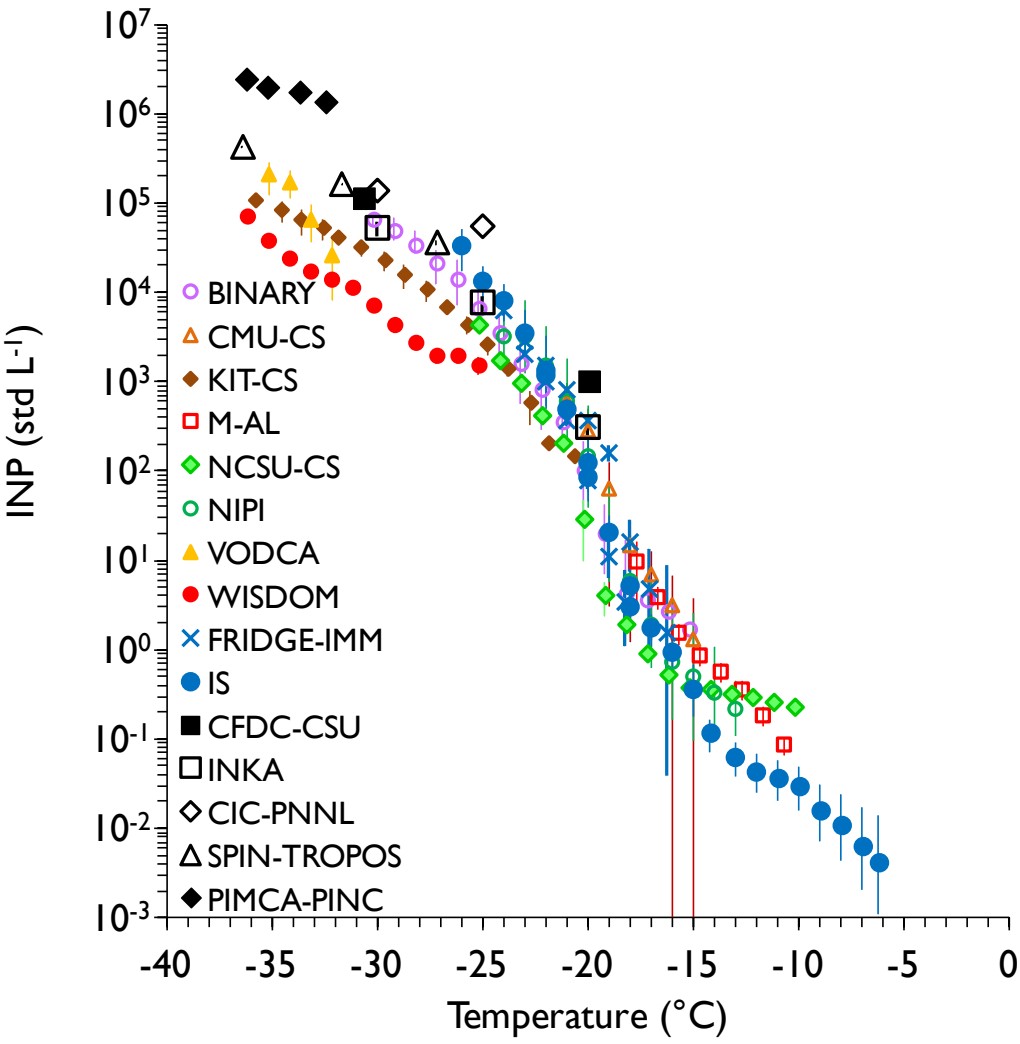

**Figure 5.** As in Fig. 4, but for sampling of Tunisian soil dust from the APC. All distinguishing features discussed with regard to Fig. 4 apply here as well. Note that two FRIDGE-IMM filter samples are represented here as two data points at a given temperature. The data herein correspond to FIN-02 APC experiments 11 and 12 on March 18, 2015.

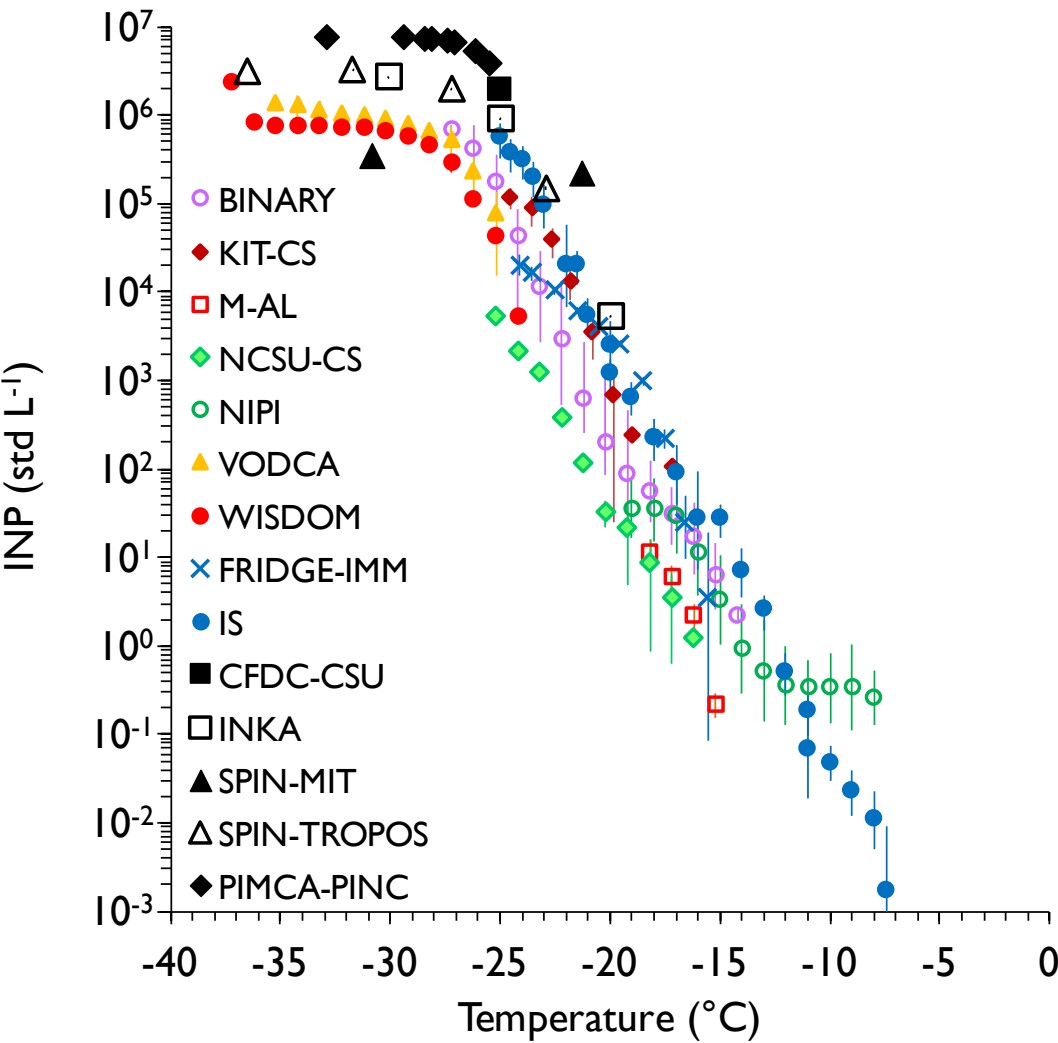

**Figure 6.** As in Fig. 4, but for sampling of K-feldspar particles from the APC. All distinguishing features discussed with regard to Fig. 4 apply here as well. The data herein correspond to FIN-02 APC experiments 13 and 14 on March 19, 2015.

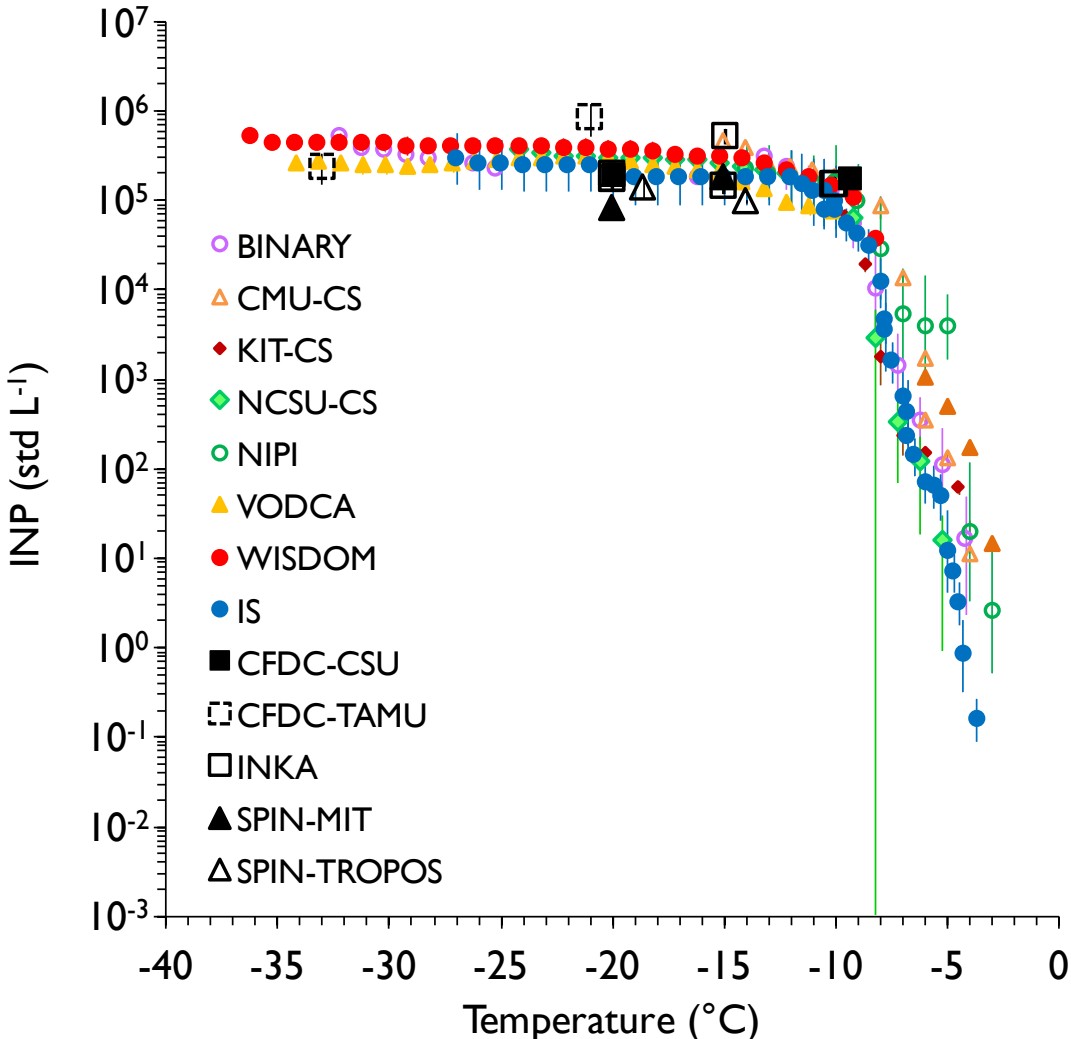

**Figure 7.** As in Fig. 4, but for sampling of aerosolized Snomax[®] particles from the APC. Two sets of measurements are plotted separately from the CMU-CS system (open and filled) due to variability observed between replicate samples as discussed in the text. The closed orange triangles are from experiments run immediately after the sample had thawed, while the open triangles are for runs that occurred within a few hours of thawing. In this case, all post-processed data are from the impinger sample. INKA data are included for all water supersaturated conditions not exceeding water breakthrough $RH_w$. The data herein correspond to FIN-02 APC experiments 15 and 16 on March 20, 2015.

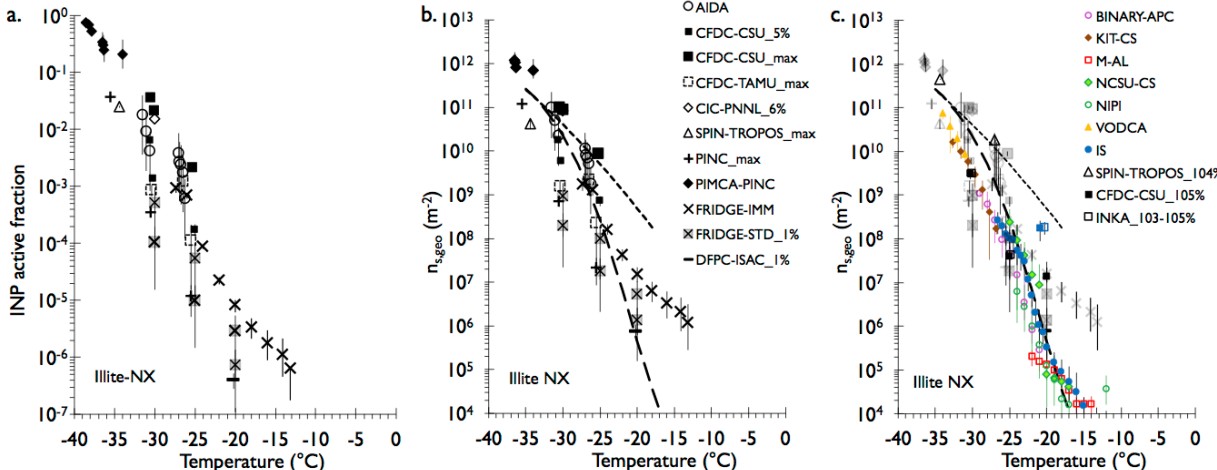

**Figure 8.** Panel a: Ice active fraction from multiple experiments performed on illite NX aerosols sampled directly from the AIDA chamber prior to expansion cloud experiments. In these cases, the subsequent AIDA measurements of activated ice crystal concentrations are included for comparison when water supersaturation was achieved during the expansion cycle. Instrument symbols are shown in Panel b. Panel b: Conversion of data to geometric surface active site density parameter, $n_{s,geo}$ using AIDA aerosol distribution data, and data within Table 3. Where specific instrument water supersaturations were selected for comparison, these are indicated after the instrument label (e.g., CIC-PNNL_6%, implies 106% $RH_w$). The term "max" means the highest $RH_w$ achieved in a scan (102-110%), as listed in Table S1. FIN-02 AIDA experiments (unnumbered, March 11, 2015), 4 (March 16, 2015), 10 (March 19, 2015), 22 (March 26, 2015), and 25 (March 27, 2015) are represented. Gumbel cumulative distribution (log-space) fit curves for illite NX are from Hiranuma et al. (2015; cf., Table 3), representing wet suspension (long dash) and dry-dispersion experiments (short dash) in that study. Panel c: Wet suspension (colored points) and flow chamber (black points) $n_{s,geo}$ data derived from APC experiment 7 (March 16, 2015) for all instruments listed in the legend are overlain on greyed-out AIDA experiment points from Panel b. Additional CFDC-CSU and INKA data points from APC experiment 3 (March 13, 2015) are included as blue data points for these instruments

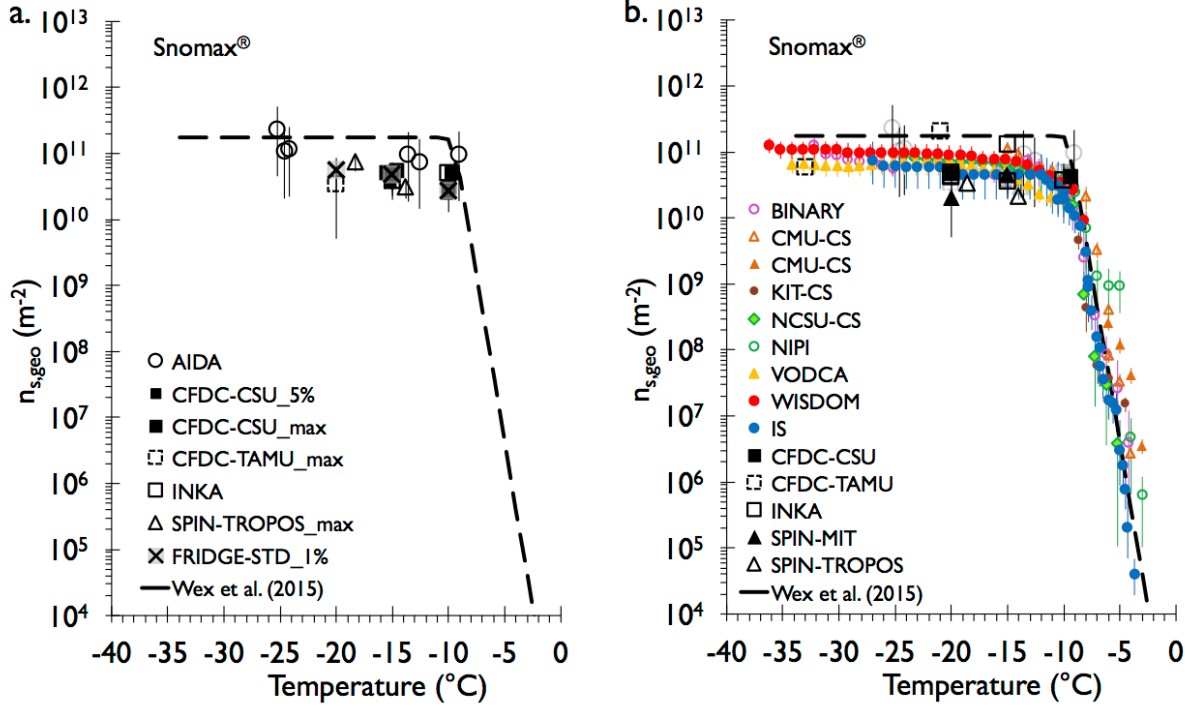

**Figure 9.** As in Fig. 8. Panel a: Ice active site density in multiple experiments performed on Snomax[®] aerosols sampled directly from the AIDA chamber. AIDA experiments 6 (March 17, 2015) and 13 (March 21, 2015) are represented, as listed for different instruments in Tables S1 and S2. The fit for $n_{s,geo}$ is derived from the Wex et al. (2015) fit for nm, the number of molecular INPs per dry mass of Snomax[®], as explained in the text. Panel b: $n_{s,geo}$ derived from FIN-02 APC experiments 15 and 16 on March 20, 2015 (Fig. 7) are overlain on greyed-out AIDA experiment data points from Panel a.

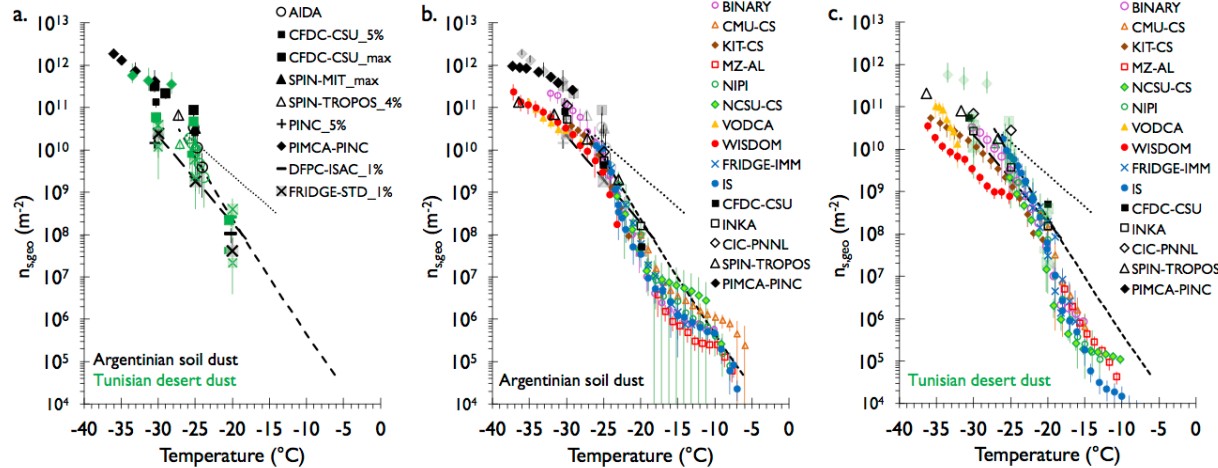

**Figure 10.** As in Figures 8 and 9, but for Argentinian soil dust and Tunisian soil dust sample experiments. Panel a: Data from FIN-02 AIDA experiments 5 (March 16, 2015), 9 (March 19, 2015), 24 (March 26, 2015), and 26 (March 27, 2015) are represented for Argentinian soil dust, and AIDA experiments 7 (March 18, 2015) and 12 (March 20, 2015) are represented for Tunisian soil dust, as listed for different instruments in Table S1. AIDA cloud expansion results are represented only for AIDA experiments 7 and 9, when mixed-phase clouds formed and persisted. Two fits of $n_{s,geo}$ *(T)* for previous surface soil dust particle types reported in the literature are from Tobo et al. (2014) ("Wyoming soil dust", long-dashed) and O'Sullivan et al. (2014) ("fertile soil dust", short-dashed), and Steinke et al. (2016) ("agricultural soil dust", dotted). APC data from Fig. 4 for Argentinian dust and Fig. 5 for Tunisian dust are overlain in panels b and c, respectively.

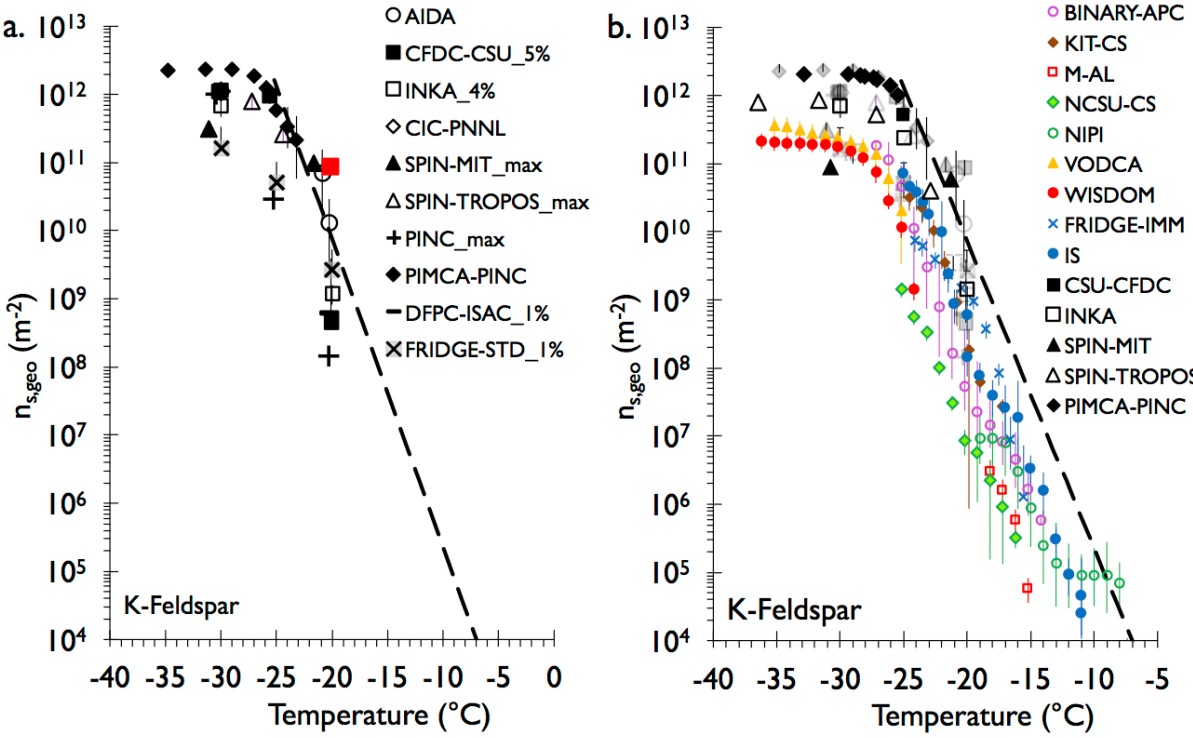

**Figure 11.** As in Fig. 9, but for K-feldspar aerosols sampled from the AIDA chamber prior to expansion-cooling experiments 8 (March 18, 2015), 11 (March 201, 2013), and 14 (March 23, 2013) for panel a, and overlay of data from APC experiments 13 and 14 on March 19, 2015 (Fig. 6) in panel b. The K-feldspar fit from Atkinson et al. (2013) is shown for comparison, after conversion from $n_{s,BET}$ to $n_{s,geo}$, as described in the text. The experiment represented by the red data point in panel a from the CFDC-CSU instrument is discussed in Supplement Section S.1.2 in relation to experimental detection issues.