# Peer review of "The Fifth International Workshop on Ice Nucleation phase 2 (FIN-02): Laboratory intercomparison of ice nucleation measurements"

_Atmospheric Measurement Techniques, 2018_

## Referee Comment (RC2)

**Review of "The Fifth International Workshop on Ice Nucleation phase 2 (FIN-02): Laboratory intercomparison of ice nucleation measurements" by DeMott et al.**

In the DeMott et al. FIN-02 manuscript the authors collate and present data from a large scale instrument intercomparison effort. The work presented here attempts to focus on a certain subset of measurements and/or measurement conditions to present a summary of the level of agreement to the community. This manuscript is important for a number of reasons, but primarily I believe it is of benefit in that it presents an assessment of measurement variability between instruments in highly idealized experimental systems. Thus, one would expect the range of results presented here to represent a baseline of uncertainty between measurements in the field.

In such an endeavor of course many choices are made and here overall I think that the authors have made good choices, and the manuscript is well written and tells a cohesive story. I do find some small areas where I think the reading is confusing and some clarifications could be made. Those and other editorial comments are listed in an itemized fashion below.

As a general comment, I do think the Summary and Conclusion section could have expanded the "do's and don't's" a bit. The authorship represents a large number of those active in the field and I do not think it would be giving away any secrets if they discussed a bit what failed, or was discovered not to have worked during the campaign. Also, they do make a few suggestions for the future, but I am left wondering if the community came no closer to formulating certain experiments that could be done side-by-side anytime two or more instruments are co-sampling. In other words, does any suggested minimum protocol emerge to assess the agreement between instruments? Perhaps a complete picture and set of recommendations emerging from FIN-02 will be detailed in another publication?

In general, if reviewer comments are addressed I recommend this paper for publication without a need for further review.

Itemized Editorial Comments (page/line):

• 1/38 item b only has a departmental affiliation it is missing an institutional affiliation.

• 6/16-18 What comes after the comma at: "...Fig. 1, and why..." does not make sense as written. The why results are given at certain RH is not noted? I think the clause needs to be rewritten.

• 7/28 I recommend you change "in this manuscript" to 'presented here'.

• 7/30 This first sentence of the section is long and awkward. I recommend it is changed. It can be split into 2 sentences.

• 8/4-5 thermodynamic paths not path

• 8/10 would limit particle loading

• 8/18 activation temperatures instead of "temperatures of activation". Also, add 'and referenced' after "...are listed". I was looking for the instrument references as I was reading and did not realize they were in the table until getting there. Finally, I suggest striking "Details on all of" rather begin the next sentence with "The specific ..."

• 9/1 I recommend "besides" is changed to 'in addition to'

• 9/3-5 The sentence beginning " In this regard..." is awkward and should be rephrased. Is it the recommendation of the 2007 workshop, or previous workshops and it was followed in 2007?

• 9/5 recommend change to 'Schematic thermodynamic paths of the AIDA chamber experiments are shown by the ....'

• 9/20 suggest '....cloud regime (i.e., 0 to -36 C). Thus they provide....'

• 10/12 suggest wording is changed to 'However, in some experiments larger particles were present ...'

• 10/13 should "some cases" be 'those cases'

• 12/2 I suggest that when $n_{s,geo}(T)$ is defined the units $[m^{-2}]$ are included.

• 13/6 suggest wording is changed to 'A striking feature of these results is the general correspondence between all methods ...'

• 13/14 suggest 'The greatest discrepancies' replace "Greatest discrepancies..."

• 13/15 "Supplemental" should be 'Supplement'

• 13/19 suggest wording is changed to '...understanding of what $RH_w$ value might...."

• 13/23 The final sentence of the paragraph ending, "... Fig. 4 is encouraging" is oddly placed in a results section. I think it is extraneous.

• 14/15 replace "at the same time" with 'simultaneously'

• 14/27 replace "Comparison" with 'A comparison'

• 15/17-20 This is a very long and confusing sentence, I suggest rewording.

• 15/29 Should it not be CMU-CS?

• 15/33 awkward wording. The source of variations question?

• 22/1 "provide the decision" is awkwardly worded.

• Figure 3 caption – The caption fails to define what the differently colored points indicate. Furthermore, I suggest the 2 light blue shadings be somehow differentiated. If my understanding is correct then I think the 2nd sentence would read, "An initial impinger and filter sampling period is highlighted in light blue 1 and is followed by an APC refill and subsequent sampling period (light blue 2). In any case I think the caption should be carefully reread and reworded for clarity.

• Figure 4 caption – Suggestions: (1) strike "on some points" (2) replace "buried within" with 'subsumed by' (3) add '...errors) or alternatively the binned...'

• Figure 7 caption – bring "in this case" to beginning of sentence

• Figure 8 may be better presented vertically for such a 2 column journal? Perhaps the authors intend to stretch this over the entire page? Currently the text seems somewhat small. (also Figures 9-11).

General figure comments: Can the chosen color schemes for the instruments remain consistent between those figures that represent APC and AIDA data? Also, I think making many of the figures box plots with at least the major tick marks represented on the minor axis would be more reader friendly.

---

## Referee Comment (RC1) · G. Vali (Referee) · 23 Aug 2018

Reviewer comments on "The Fifth International Workshop .... " by DeMott and Co-authors. (Atmos. Meas. Tech. Disc., https://doi.org/10.5194/amt-2018-191)

Gabor Vali

Aug. 22, 2018

Phase 2 of the Fifth Workshop (FIN-02) is the most comprehensive of the ice nucleation instrument inter-comparisons that have been organized since the first one in 1967. Significant progress has been made in recent years in instrument development, in understanding of the processes relevant to ice nucleation and in knowledge about ice nucleating materials. Thus, the paper is of great interest to many scientists. The authors are to be congratulated on the work described.

The paper summarizes the work of a large number of individuals. The long author list makes reviewing the paper somewhat intimidating. How can anything else be noted that such a large group hasn't already considered? On the other hand, perhaps it is harder to make sure that all's in order when so many hands are involved. The reviewer's task is further burdened by having so much material to review. In fact, this reviewer focused more on general aspects of the paper than on detail.

Essentially all in this review addresses what is presented in the paper, not what and how the experiments were done. Most of the instruments participating in the Workshop have already been described in refereed publications. The operating conditions, the selection of aerosol types and the procedures followed during the Workshop have been arrived at after extensive deliberations and with the benefit of experience at earlier workshops. These issues are not addressed here except regarding the consequences of the procedures in interpreting the data obtained.

**Broad comments:**

1. While it is clear that no classification can reflect perfectly the variety of instruments used, it is worth thinking about how useful is the one applied in this paper. No doubt, the online vs. offline designation has its origin in common references to the instruments in that some can be deployed in the field as a self-standing unit, while the others have essential work done in the laboratory. Thus, 'field' vs. 'laboratory' instruments could also be applied. The current designation has disadvantages. It doesn't inform people unfamiliar with the details of what the essential difference is. Is it the operating principle of the devices? Is it aerosol flow through the device versus samples captured and processed? Portable versus fixed in not necessarily applicable, as a filter processing unit is no larger and difficult to transport as a CFD chamber. A drop freezer can be the smallest of all the devices. Also, field deployment of a device impacting a flow of aerosol on water drops would be an online instrument for which much of the RH discussion about CFD's would be irrelevant. So, while the offline/online distinction isn't all bad, perhaps "direct sampling" versus "post-processing" would be more meaningful. Another useful direction would be to focus on variable RH instruments versus those with liquid water. What characteristics is the most important? Tables 1 and 2 do a pretty good job in listing the main features of the various devices. But the titles of subsections 2.1 and 2.2, and the headings in the Tables could be improved by removing the hesitation in the designations and, preferably before 2.1, explain the reason for the classification. Dry versus wet in the table headings is not the best. My recommendation is to use 'direct sampling' versus 'post-processing' and define in the Introduction how these terms are applied. This is also appears to be the intention with lines 30-32 on page 7. These designations would allow the AIDA chamber to be in the group of instruments with clouds forming on the sampled aerosol, just like in the CFD's. Omit 'portable' as part of the definition. (Apologies for this rambling paragraph.)

2. I am somewhat reluctant to raise this issue, but this paper needs special scrutiny in this regard because it weighs

heavily on the principal results of the paper. Screening of data to be included or excluded is perhaps the trickiest and most sensitive part of the paper. There is a sense of some censoring of the data throughout Section 2. Undoubtedly some judgements had to be made about data validity by some criteria. One can see the attempt to do this, and the authors certainly deserve to be given the credit of proper judgement. However, the first question that comes to mind when looking at the degree of agreement shown in the results is how much screening of the data was done? How many discrepant cases were excluded due to errors or uncertainties? In other words, how much subjectivity entered into the analysis. Some comments on this issue would help the readers gauge the value of the results.

3. It becomes gradually clear as one reads on in the paper, that the focus of the analyses was freezing nucleation, at least as much as it could be assumed to be the dominant mode of nucleation if sufficient supersaturation was achieved to ensure that INPs are in liquid droplets. Perhaps, I missed it, but if this is true, shouldn't it be clearly spelled out in the Introduction, and be part of the goals? Similarly, the size range covered (9/28-32) is a pre-defined constraint and would be best stated up front.

4. Unless I missed it, the promise made at 8/32 to describe the aerosol collection and transfer methods for testing as suspensions are missing from Section 2.4. Problems associated with the method used and the final efficiency of the capture are not found. See also 13/4-5.

5. Some re-phrasing of conclusions such as in lines 6 to 23 on page 13 should be considered. The degree of agreement in the data is remarkable and deserves to be celebrated.. But to say that the agreement extends over 7 orders of magnitude over the whole temperature range is misleading. Except in the center of the temperature range only 4-5 data sets are represented. Thus, in a strict sense, the agreement among all data sets is not quantitative over the whole range. They all coincide in defining a common trend and agree in overlap regions. None of the data sets extend over the whole range. Also, there is a division of the types of instruments for which data are available at the upper and lower ends of the temperature regime. The main novelty in these results is the agreement between CFD type and drop freezing types of measurements in the overlap region. This is a real accomplishment and demonstrates that the two types of measurements can be combined to yield atmospheric measurements over a wide temperature range. The extent to which that is a practical solution for low INP air samples is worth addressing.

6. Only in S.2.1 is the method for calculations of the INP concentrations stated. How were the presented data derived for each device should be part of the description for each one. This also applies to how uncertainty ranges were determined.

7. A suggestion I make reluctantly, because of the amount of re-writing it implies, the authors should consider including a Discussion section, moving some material from Results there, and going a little further in some themes. Having a Discussion section would allow for better perspective on the various aspects of the results (e.g. 14/3-22), and give good opportunities for comparisons with previous publications (e.g. 18/2-28).

8. Parts of the description of results refer to the sigmoid shapes and the central slopes of the INP vs. T curves. I mention here that similar considerations, and a definition for a slope parameter were given in Vali (ACP 14, 5171-5194, 2014.). The rich material in this paper could very usefully add and expand on the analyses in that paper. That may go beyond the scope the authors set for this paper, but in reality it would not be a large step.

9. There are many references to the Supplement. It would be helpful if, in each case, the reference pointed to some specific part of the Supplement.

10. It is regrettable that the ACP (3.1) and AIDA (3.2) sampling results are in different forms, i.e. per air volume and as site density. Was this unavoidable? Considered unimportant? It comes as a surprise to the reader. Having the two different aerosol processing paths provided some assurance of minimizing problems related to each one. Perhaps there were other operational advantages or limitations as well. Yet, one would have like to see the results in the same format. For inter-comparisons the two data sets provide additional support and that is the main goal of the Workshop.

But it isn't clear why the results need to be given in different manner. More careful reading of the paper may reveal the reasons, but perhaps readers (like me) could be helped by a statement of those reasons up front.

11. The focus on measurements of atmospheric INPs in the recommendations on page 22 is well placed. But ice nucleation measurements are also needed for fundamental understanding of ice nucleation and of the nature of INPs. Another important direction that ice nucleation measurements should explore in future workshops is to perrform tests with mono-disperse aerosols.

**Specific items:**

page/line

3/32      'constrain' INP population???

4/7      The fact that the more general aspects of the project are to be published later is a bit of a problem. A brief description of the three stages would help.

4/10-12      The order of these last two sentences of the paragraph should be reversed.

4/22      What do you mean by 'align'?

4/25-32      Goals are well defined. It would be nice if the conclusions responded clearly to each item.

4/33 ->      The intent to define a priori instrument characteristics is somewhat futile. Instruments that participated can be grouped as on-line and off-line but linking that too strongly to dry and wet is too much of a simplification. It would be more effective to describe the methods starting with Fig. 1 and then detail what the instruments do (above water saturation and immersion freezing) and do not do (deposition, condensation-freezing, contact). Indeed, reference to Vali et al, 2015 can help with the definitions.

5/16      The operating principles are in in Tables 1. and 2 to some extent and are described in 2.1 and 2.2. The Supplement is to describe the *detailed implementations* of the basic idea in each device.

Fig. 1      The inclusion of X wt% designation here is unhelpful. The inclusion of this factor complicates interpretation. Indicated range may have been correct for the INPs chosen for test but have no general meaning. There are other materials that cover the whole temperature range with a single wt%. Adding the specific INP for which the graph is valid would add unnecessary detail. I suggest to leave those wt% indications off the graph. Of course, solute effects are ignored here.

6/14      The definition of max $RH_w$ practicable before 'breakthrough' is made difficult by the use of $RH_w$ for many different aspects of the processes.

6/18-20      Is this caveat not in conflict with the goals defined earlier? May be just say that experiments included in the evaluations all stayed below the breakthrough $RH_w$.

6/31-33      This sentence should start the paragraph to explain why the choice of 102% needs discussion.

6/37-7/1      This is problematic. How is this known? The fraction of aerosol activated as a measure of the RH achieved in an instrument? The nominal RH and actual volume-weighted one differ, as just described in earlier part of the paragraph.

7/3-7/5      Recommendation here is out of place and heavy-handed.

7/7      Shouldn't this 'primary comparison' be defined as a 'goal' up front?

7/11      Quite unclear. Highest vs. maximum. Could the question be said to have been to select the RH point in the scans that the operators thought was most representative for freezing nucleation? Again, this objective should be stated before discussing details of RH variations in the instruments. All those variations are caveats on the validity of the values chosen/used for the comparisons. *In any case, isn't it likely that the activated INP were in drops, due to the high hygroscopicity of the aerosol?* In that case, the INP's don't experience 102% or whatever supersaturation. Clearly, these considerations need to be addressed here.

8/5      Is this because only ice crystals forming from frozen droplets on the surface are counted? How certain is that? Same issue as in previous point.

8/7      " ... limits reduction of ..." is unclear. The influence of vapor competition, and its dependence on INP density on the substrate has extensive literature, and perhaps should not be glossed over so easily. The operational definition used here for making the comparisons deserves more explanation. The concluding sentence on 8/13-15 is unclear - do you mean cases when vapor competition was only inferred to have been present but not evaluated?

8/8      '...were allowed to ..' inserts a hierarchical tone that is unnecessary. An objective tone would be more appropriate. The working arrangements of the workshop are not of interest to the reader. Else, make it clear and explain in the Introduction that there was some sort of checks and balances arrangement in order to improve the final result. I don't know if there was or not. See also comment on 7/3-5.

8/14      "wet suspension" -- ??  wouldn't liquid suspension or water suspension be more direct? Actually, the first two words could well be omitted and the sentence say "Measurements of immersion freezing ...."

8/31      " ... wet suspension groups ...? -- one can deduce what is meant but it could be said better

9/1      "... basic methods ... " may need to be re-considered if changes are made in response to the first item in this review.

9/29      suggest leaving out the word 'mode'

10/3      suggest replacing 'wet dispersion of particles'  by something like "aerosol generation from aqueous suspensions" or  "particle dispersion from aqueous suspensions" or "dispersion of particles via spraying and drying"

10/32      Was the depletion of aerosol in the APC by sample withdrawal avoided or neglected? Also, how justified was it to assume the decrease in concentration to be valid for all sizes?

12/30      perhaps the intention was to put the words 'basis for"  in the sentence, to read : .... primary basis for the comparison of ...

12/31      replace 'most' by 'largest number' ??

13/18-20      The sentence is somewhat garbled.

14/3-22      These comments would be better placed in a Discussion, not in the Results section.

15/12-14      I think it is too uncertain to explain results in terms of the numbers of proteins. Snomax contains full cells and evidence for separation of the INP material (protein) from the cell wall is unclear. Leveling off  the INP curves shows a limit in the number of INPs as a fraction of the total number of cells. This is a matter of expression of the INP protein under induced conditions and of the processing of the bacterial culture. Again, the emphasis here should be on the agreement among measurement systems. Interpretation of the shape of the INP curves is a matter for Discussion (see Broad comments 7 and 8).

15/25->   Reference here to first-freezers etc. is confusing without more detailed knowledge of what is being discussed. This type of error analysis for specific devices should be part of the apparatus description (cf. Broad comment 6).

15/32   Supporting Information =? Supplement. Where is the material referred to in this line?

16/10   Why wasn't the fraction expressed as active INP versus total number of potential INP particles introduced? Maybe that is what is meant, but I am not sure from the wording given here.

16/16-17   The conversion to active site density is the same for all measurements, provided particle sizes are known. Why is this introduced here? (cf. Broad comment 10)

20/28-31   Do these two sentences refer to the same or two separate findings?

21   Some numerical values for the errors discussed would be useful.

21/23   'de-agglomeration' has a simpler alternative:  'breakup' and removes the implication that all large INPs are aggregates of many smaller ones

21/28   " ... full immersion of all particles in the same liquid volume .." is too vague to focus on, as other factors like time in suspension may also come into play. Also, are the differences beyond the error bars of the PIMCA-PINC and cold-plate methods?

21/32   why is that need artificial?

22/4   '.. uniformly capture activation ...' is awkward wording

---

## Referee Comment (RC3) · G. Vali (Referee) · 2 Sep 2018

G. Vali (Referee)

vali@uwyo.edu

Please disregard point #6 of the Broad Comments in my review. Contrary to my comment, information about how the INP concentration were derived and the methods of error estimation are included in the Supplement.

---

## Author Comment (AC2) · 4 Oct 2018

**Response to anonymous Reviewer 2 comments on "The Fifth International Workshop on Ice Nucleation phase 2 (FIN-02): Laboratory intercomparison of ice nucleation measurements"**

We thank the reviewer for both the positive comments as well as detailed additional comments and corrections.

**General Comment:** As a general comment, I do think the Summary and Conclusion section could have expanded the "do's and don't's" a bit. The authorship represents a large number of those active in the field and I do not think it would be giving away any secrets if they discussed a bit what failed, or was discovered not to have worked during the campaign. Also, they do make a few suggestions for the future, but I am left wondering if the community came no closer to formulating certain experiments that could be done side-by-side anytime two or more instruments are co-sampling. In other words, does any suggested minimum protocol emerge to assess the agreement between instruments? Perhaps a complete picture and set of recommendations emerging from FIN-02 will be detailed in another publication?

**Authors' response to General Comment:** While the goal of the workshop was not necessarily to fashion protocol for all future such instrumental comparisons, nor to be judgmental, we agree with the reviewer that we could expand our recommendations, and point to things that could/should be done in a comparison involving any particular number of participants. At the same time, it is clear to those involved that the composite results of all three phases of the workshop will be needed to provide a complete picture and set of recommendations. Additional papers in that regard are forthcoming. We have completely reorganized and added to the Summary and Conclusions of this paper on the basis of the requests of both reviewers.

**Changes in manuscript re: General Comment:**
The Conclusions have been reorganized, first by summarizing and augmenting the initial discussion within the structure of the objectives listed in the Introduction of the paper, and secondly by adding a list of topics recommended for further research. As this section amounts to nearly 6 pages, we append it specially to the end of this response after first completing responses to itemized editorial comments.

**Itemized Editorial Comments (page/line):**

*Comment 1*. *Page/line* 1/38 item b only has a departmental affiliation it is missing an institutional affiliation.
**Comment 1 author response/manuscript change:** Thanks, corrected.

*Comment 2*. Page/line 6/16-18 What comes after the comma at: "...Fig. 1, and why..." does not make sense as written. The why results are given at certain RH is not noted? I think the clause needs to be rewritten.
**Comment 2 author response/manuscript change:** We have rewritten this section for clarity: *"Therefore, a single $RH_w$ level for this condition to occur is not noted in Fig. 1. Instead, results are stated as being associated with*

*specific RH$_w$ values (or % supersaturation values, which equal RH$_w$-100) that are simply a value that was lower than the droplet breakthrough condition in some cases. In other cases, the maximum RH$_w$ achievable prior to droplet breakthrough is reported."*

**Comment 3**. Page/line 7/28 I recommend you change "in this manuscript" to 'presented here'.
**Comment 3 author response/manuscript change:** Changed as requested.

**Comment 4:** Page/line 7/30 This first sentence of the section is long and awkward. I recommend it is changed. It can be split into 2 sentences.
**Comment 4 author response/manuscript change:** Done

**Comment 5:** Page/line 8/4-5 thermodynamic paths not path • 8/10 would limit particle loading
**Comment 5 author response/manuscript change:** Changed as requested.

**Comment 6:** Page/line 8/18 activation temperatures instead of "temperatures of activation". Also, add 'and referenced' after "...are listed". I was looking for the instrument references as I was reading and did not realize they were in the table until getting there. Finally, I suggest striking "Details on all of" rather begin the next sentence with "The specific ..."
**Comment 6 author response/manuscript change:** Changed as requested. Reference to Table 1 was also added on Page 5.

**Comment 7***: Page/line 9/1 I recommend "besides" is changed to 'in addition to'*
**Comment 7 author response/manuscript change:** Changed as suggested.

**Comment 8:** Page/line 9/3-5 The sentence beginning "In this regard..." is awkward and should be rephrased. Is it the recommendation of the 2007 workshop, or previous workshops and it was followed in 2007?
**Comment 8 author response/manuscript change:** Rewritten as, *"In this regard, we follow the example of the 2007 workshop, and a key recommendation from ice nucleation workshops prior to that time, that an expansion cloud chamber be utilized to provide a simulation of cloud activation (DeMott et al., 2011)."* This is discussed in the 2011 paper and references to prior workshops are provided.

**Comment 9:** Page/line 9/5 recommend change to 'Schematic thermodynamic paths of the AIDA chamber experiments are shown by the ....'
**Comment 9 author response/manuscript change:** Changed as suggested.

**Comment 10:** Page/line 9/20 suggest '....cloud regime (i.e., 0 to -36 C). Thus they provide....'
**Comment 10 author response/manuscript change:** Changed as suggested.

**Comment 11:** Page/line 10/12 suggest wording is changed to 'However, in some experiments larger particles were present ...'

**Comment 11 author response/manuscript change:** Changed as suggested.

**Comment 12:** Page/line 10/13 should "some cases" be 'those cases'

**Comment 12 author response/manuscript change:** Corrected.

**Comment 13**: Page/line 12/2 I suggest that when ns,geo(T) is defined the units [m−2] are included.

**Comment 13 author response/manuscript change:** Done.

**Comment 14**: Page/line 13/6 suggest wording is changed to 'A striking feature of these results is the general correspondence between all methods ...'

**Comment 14 author response/manuscript change:** Changed as suggested.

**Comment 15:** Page/line 13 /14 suggest 'The greatest discrepancies' replace "Greatest discrepancies..."

**Comment 15 author response/manuscript change:** Changed to "The largest…"

**Comment 16:** Page/line 13/15 "Supplemental" should be 'Supplement'

**Comment 16 author response/manuscript change:** Done.

**Comment 17:** Page/line 13/19 suggest wording is changed to '...understanding of what RHw value might...."

**Comment 17 author response/manuscript change:** Changed as requested.

**Comment 18:** Page/line 13/23 The final sentence of the paragraph ending, "... Fig. 4 is encouraging" is oddly placed in a results section. I think it is extraneous.

**Comment 18 author response/manuscript change:** Removed.

**Comment 19:** Page/line 14/15 replace "at the same time" with 'simultaneously'

**Comment 19 author response/manuscript change:** Changed.

**Comment 20:** *Page/line 14/27 replace "Comparison" with 'A comparison'*

**Comment 20 author response/manuscript change:** Done.

**Comment 21:** Page/line 15/17-20 This is a very long and confusing sentence, I suggest rewording.

**Comment 20 author response/manuscript change:** This section is rewritten as, *"All immersion freezing methods capture the strong rise in activation due to the presence of the most active biological ice nucleators, those within the*

*realm of the Groups I and II as defined by Yankofsky et al. (1981). This is expressed as a pronounced shoulder in all freezing spectra at temperatures warmer than -8°C in Fig. 7. We may note here that all bulk freezing methods shared the same impinger sample in this case, including the IS. This warm shoulder of ice nucleation activity has also been demonstrated by Wex et al. (2015), Budke and Koop (2015), and Polen et al. (2016)."*

**Comment 22:** *Page/line 15/29 Should it not be CMU-CS?*
**Comment 20 author response/manuscript change:** Yes, corrected.

**Comment 23:** Page/line 15/33 awkward wording. The source of variations question?

**Comment 23 author response/manuscript change:** Rewritten as, *"In summary, the strong variability in activity seen at the warmest activation temperature regime for Snomax® particles brings into question the ability to utilize the warmest temperature (> -10°C) freezing behavior of Snomax® reliably for calibration purposes. This has been noted previously by Polen et al. (2016), and attributed to batch variability and the loss of activity during storage."*

**Comment 24:** Page/line 22/1 "provide the decision" is awkwardly worded.
**Comment 24 author response/manuscript change:** Replaced with "…provide guidance…"

**Comment 25:** Figure 3 caption – The caption fails to define what the differently colored points indicate. Furthermore, I suggest the 2 light blue shadings be somehow differentiated. If my understanding is correct then I think the 2nd sentence would read, "An initial impinger and filter sampling period is highlighted in light blue 1 and is followed by an APC refill and subsequent sampling period (light blue 2). In any case I think the caption should be carefully reread and reworded for clarity.

**Comment 25 author response/manuscript change:** We had hoped for the colored scales to direct eyes to the appropriate axes, but for clarity we have rewritten it all as, *"Data from the first half of a daily experimental series, after an initial fill of Tunisian soil dust (Experiment 11 and 12 of the FIN-02 APC series on March 18, 2015) into the APC at a time (time 0) taken to be the experimental start time. Impinger and filter sample periods after two separate chamber fills are highlighted in light blue shading. To obtain a reference aerosol concentration (cm⁻³) via the CPC (total particles) (blue points and left scale) or optical particle counter at sizes larger than 500 nm (n_500 (brown points and right scale), as well as surface area ($\dot{r}$ m² cm⁻³) (black points and left scale) for the offline sampling period, the time-weighted average of the two sampling periods was determined. This period-integrated concentration value could then be used to ratio versus the concentrations of particles present at later sampling times during the online sampling of aerosols from the APC (green shaded region) in order to back-correct the online INP number concentrations to those interpreted from the offline samples."* The text section describing the procedure is already quite detailed, and does mention the fact that there were sometimes two sample chamber fills, and sampling periods for which the integrated particle number values and surface areas are determined.

**Comment 26**: *Figure 4 caption – Suggestions: (1) strike "on some points" (2) replace "buried within" with 'subsumed by' (3) add '...errors) or alternatively the binned...'*

**Comment 26 author response/manuscript change:** Changed as suggested.

**Comment 27:** *Figure 7 caption – bring "in this case" to beginning of sentence*

**Comment 27 author response/manuscript change:**
Done.

**Comment 28:** Figure 8 may be better presented vertically for such a 2 column journal? Perhaps the authors intend to stretch this over the entire page? Currently the text seems somewhat small. (also Figures 9-11).

**Comment 28 author response/manuscript change:** Due to the length of the caption, we will suggest to place these figures across the entire page, unless the editors suggest otherwise. And although the text is somewhat small, there were many different instruments to indicate, and it should be possible for most to expand the figures in PDF format. But again, we will go with the decision of the copy editors in this case.

**Comment 29:** General figure comments: Can the chosen color schemes for the instruments remain consistent between those figures that represent APC and AIDA data? Also, I think making many of the figures box plots with at least the major tick marks represented on the minor axis would be more reader friendly.

**Comment 29 author response/manuscript change:** It was indeed the intent to maintain color schemes for different data across figures. We will recheck that this is done. Once we began to introduce AIDA data at Figure 8 and beyond, and when we desired to show both soil dusts together in Figure 10, it complicated matters, so we simply made all of the online instruments black (or green and black in the case of Fig. 10). This also assisted with shading the APC versus AIDA sampling points in the later figures. We revise now to use black symbols for flow chambers all the way through the figure series, and we distinguish IS and FRIDGE-IMM series as blue in the APC figures now. The exception to this overall scheme will continue to occur in Figure 10, where green is used to distinguish the two dusts amongst the direct sampling instruments. As for the scaling on box plots, it would be too difficult to add the extra scale in the AIDA figures and simultaneously keep the long legends that are perhaps for important for easily distinguishing instruments. For that reason, we do not add the secondary scales on any of the figures.

**New Conclusions section:**

[revised manuscript text omitted]
. There is also no current check on the accuracy of post-processing methods for immersion freezing at modest supercooling where this is the only method practically available at present, one that necessarily involves the loss of time and space resolution when sampling on aircraft. Can new methods be developed for directly assessing INP concentrations in larger sample volumes? Even modest improvement to direct methods to provide more overlap of measurement methods in the regime > -20°C would help to further evaluate the validity of meshing direct and post-processing methods to characterize INPs over the full mixed-phase temperature regime. Aerosol pre-concentration has been applied to extend the dynamic range of direct INP measurements in atmospheric studies (e.g., Tobo et al., 2013; Boose et al., 2016), but aerodynamic concentration methods bias against particle sizes much below 1 μm. Hence, it is worth investigating the possible use of other concentration methods applicable to the full aerosol size distribution, such as pre-condensation. Novel ideas are needed.

- Further comparisons for which the sampling groups are "blind" to the nature and concentrations of INPs being sampled could be useful toward giving confidence to the wider community that the INP measurement community is capable of recognizing issues and properly interpreting data. This will assist confidence and utility of larger global data sets. Such a comparison from FIN-02 will be reported on in a separate publication in preparation.

- Similar exercises as FIN-02 are also needed in sampling under ambient atmospheric conditions. This is the subject of the FIN-03 campaign that will be reported on separately.

Workshops such as FIN-02 will continue to play a large role in assessing measurement biases and ultimately improving the comparability of INP measurements made by a large community of researchers sampling on a global scale. The shared experience of these workshops is irreplaceable in providing special insights into the status of and issues involved in obtaining INP data in different scenarios that may be dominated by certain aerosol types. FIN-02

demonstrates that the INP measurement community remains on a progressive track towards assessing convergence between different methods used for INP quantification.

---

## Author Response (AR1)

**Response to Gabor Vali's comments on "The Fifth International Workshop .... " by DeMott and Co-authors**

We thank Dr. Vali for his comprehensive and insightful comments, and his understanding of the complexities of such workshops.

**Broad comment [1]:** While it is clear that no classification can reflect perfectly the variety of instruments used, it is worth thinking about how useful is the one applied in this paper. No doubt, the online vs. offline designation has its origin in common references to the instruments in that some can be deployed in the field as a self-standing unit, while the others have essential work done in the laboratory. Thus, 'field' vs. 'laboratory' instruments could also be

10 applied. The current designation has disadvantages. It doesn't inform people unfamiliar with the details of what the essential difference is. Is it the operating principle of the devices? Is it aerosol flow through the device versus samples captured and processed? Portable versus fixed in not necessarily applicable, as a filter processing unit is no larger and difficult to transport as a CFD chamber. A drop freezer can be the smallest of all the devices. Also, field deployment of a device impacting a flow of aerosol on water drops would be an online instrument for which much

15 of the RH discussion about CFD's would be irrelevant. So, while the offline/online distinction isn't all bad, perhaps "direct sampling" versus "post-processing" would be more meaningful. Another useful direction would be to focus on variable RH instruments versus those with liquid water. What characteristics is the most important? Tables 1 and 2 do a pretty good job in listing the main features of the various devices. But the titles of subsections 2.1 and 2.2, and the headings in the Tables could be improved by removing the hesitation in the designations and, preferably

20 before 2.1, explain the reason for the classification. Dry versus wet in the table headings is not the best. My recommendation is to use 'direct sampling' versus 'post-processing' and define in the Introduction how these terms are applied. This is also appears to be the intention with lines 30-32 on page 7. These designations would allow the AIDA chamber to be in the group of instruments with clouds forming on the sampled aerosol, just like in the CFD's. Omit 'portable' as part of the definition. (Apologies for this rambling paragraph.)

**Authors' response to Broad Comment [1]:**
It was a most difficult decision, in considering how to separate the discussion pre-submission. However, the workshop was separated in a distinct manner that the discussion mimics. The samplers for offline immersion freezing measurements required the largest particle loadings and so sampled first each day. This sometimes included

30 two separate fills of the APC. At the end of this period, online instruments would collect particles for a period from the APC, over a period that extended a couple of hours beyond the period of sampling for post-processing. Finally, an aerosol fill was done directly into the AIDA chamber and this accommodated the few of the post-processing samples for devices that did not desire quite as high particle loadings (e.g., FRIDGE-STD and DFPC), and then other online samplers that would sample from AIDA just prior to the AIDA expansion. The concentrations were

35 never the same in the APC period and in AIDA. Importantly, the aerosol fill to the AIDA chamber was not always the same INP type used in the morning sampling period from the APC. This was a conscious (and deemed necessary) decision to accommodate the ability of all instruments to sample all aerosols, and for the direct sampling

instruments to be able to do so over a range of temperatures (e.g., only limited temperatures could be assessed by flow chambers in the two hours after the collections from the APC into liquid or onto filters were done, and before the AIDA expansion had to begin). We mention this here to preface our responses to later comments. Nevertheless, understanding this may help to explain a lot about how the paper is organized, and hence we have decided to add

5      such discussion to the Methods section. We hope that this discussion helps all readers to understand why we use concentration comparisons in some plots and then revert to active fraction and/or active site density in later figures that include data from both APC and AIDA experiments.

As for the exact description of device types, field versus laboratory would not work, because some of the immersion freezing methods are being taken to field facilities for immediate processing, mitigating in some cases

10     the need for freezing samples. Direct sampling versus post-processing is more or less what we intended as the meaning for online versus offline, so we accept Dr. Vali's suggestion on using this terminology.

**Changes in manuscript re: Broad Comment [1]:**
This discussion will appear as follows as the 4th paragraph in revised Section 2.3, with the new title, "*Generation of*

15     *varied INP types and general study procedures*". We write, "*The daily protocol determined for aerosol generation and measurements is an aspect of these studies that bears strongly on the organization and discussion of results in this manuscript. Especially, sampling periods were organized to optimize the opportunities and conditions for all instruments to sample the variety of aerosols. Each day over the three-week workshop period typically began with fills of one INP type into the APC, and sampling of that aerosol into liquid and onto filters over a two-hour period*

20     *for later assessment by the post-processing devices, as detailed below. This was followed by the direct samplers processing the same INP type from the APC over another two-hour period. This typically permitted direct measurement at a couple specific temperatures, with data at other temperatures being acquired on another day for the same INP type (from the APC or AIDA). Then, at midday, the AIDA chamber would be filled, typically with a different INP type than used in the morning APC experiments. Direct sampling from AIDA by the flow chambers*

25     *would occur over a period of time just prior to the start of expansion cooling experiments. As well, collections onto filters or wafers used by the DFPC and FRIDGE device (standard method) would be made only from the AIDA aerosol fill on each day, since these methods required very short sampling times in order to limit particle loading. For example, the DFPC-ISAC filters were collected for periods of 10's of seconds. Other collections into liquid or onto filters for immersion freezing post-processing would not occur from the AIDA chamber prior to expansions. A*

30     *consequence of these procedures is that we will find it convenient to present results on different bases when discussing sampling from the APC and from AIDA. While we might ideally wish to present all data on the same basis as measurements are reported for atmospheric sampling, as number concentrations, we choose to do so only for the APC experiment period that offers the opportunity for comparing the most measurement systems. We describe how that is done next. For AIDA sampling, we will display results as active fractions and ice active site*

35     *density, which then allows integrating APC results along with AIDA results for the same INP types over the course of the workshop.*"

Additionally, we widely employ the terminology of direct sampling and post-processing for the different methods used. We introduce this in Section 2 with the statement, "*These categories are, firstly, instruments operating online or for direct processing of aerosol particles and, secondly, those utilizing collections of particles for subsequent offline or post-processing.*" We also revise section headings, table titles, and consolidate the AIDA chamber description into the direct sampling category discussion in Section 2.1 and remove the original Section 2.3.

**Broad comment [2]:** I am somewhat reluctant to raise this issue, but this paper needs special scrutiny in this regard because it weighs heavily on the principal results of the paper. Screening of data to be included or excluded is perhaps the trickiest and most sensitive part of the paper. There is a sense of some censoring of the data throughout Section 2. Undoubtedly some judgements had to be made about data validity by some criteria. One can see the attempt to do this, and the authors certainly deserve to be given the credit of proper judgement. However, the first question that comes to mind when looking at the degree of agreement shown in the results is how much screening of the data was done? How many discrepant cases were excluded due to errors or uncertainties? In other words, how much subjectivity entered into the analysis. Some comments on this issue would help the readers gauge the value of the results.

**Authors' response to Broad Comment [2]:**
It is disconcerting to hear the word censoring raised in regard to judgements on instrument operation and data submission. While full participation was encouraged, even for groups doing such a thing for the first time, there was no requirement for individual groups to submit data for every case covered in this paper. In fact, there was no requirement to collect data in every case, and it is the case that some groups collected much more data than others. We have tried to be upfront about the fact that this workshop had both formal and informal parts, and this paper deals with the latter. Investigators who recognized a certain issue in a given experiment that was not part of the formal intercomparison (a separate paper) were free to remove the data pending their own investigations (some that are continuing). This was not a frequent occurrence. In at least one case, a temperature calibration issue was recognized by one team (M-AL) following the workshop, and all of their data were subsequently reanalyzed. The MIT SPIN group did not always extend RH scans to levels sufficient to express immersion freezing in all experiments during the informal workshop phase. Other groups chose to focus their efforts on the formal period of comparisons, and to use the other part of the workshop for development activities (PNNL-CIC). We do not wish to state all of these details in the paper. It should be clear by the volume of data shown in many of the figures that this allowance was not in any way an attempt to unify the data sets. The participation of different groups at different levels can also be recognized in the data table listing. We considered that if 15 measurements are used to demonstrate correspondence or not, that this is a reasonably sufficient examination.

**Changes in manuscript re: Broad Comment [2]:**
We re-emphasize at the end of the Introduction that this paper describes the informal component of FIN-02. We write, "*This paper is intended as an overview of the informal activities of the workshop while addressing the*

*majority of these objectives. It is not intended to answer all of the goals and objectives that are better addressed in separate studies. It is not our intent to rigorously test the capabilities of different measurement systems, but rather to point to areas of success and areas for needed development or further research."*

**Broad comment [3]:** It becomes gradually clear as one reads on in the paper, that the focus of the analyses was freezing nucleation, at least as much as it could be assumed to be the dominant mode of nucleation if sufficient supersaturation was achieved to ensure that INPs are in liquid droplets. Perhaps, I missed it, but if this is true, shouldn't it be clearly spelled out in the Introduction, and be part of the goals? Similarly, the size range covered (9/28-32) is a pre-defined constraint and would be best stated up front.

**Authors' response to Broad Comment [3]:**
We are thankful for this comment because it was not intended that this be a gradual realization. In the abstract, we had stated the different measurements with a focus on immersion freezing (modified here to account for the revised naming of the different methods: "The results presented here use data from the workshop to assess the comparability of immersion freezing measurement methods activating INPs in bulk suspensions, methods that activate INPs in condensation and/or immersion freezing modes as single particles on a substrate, continuous flow diffusion chambers (CFDCs) directly sampling and processing particles well above water saturation to maximize immersion and subsequent freezing of aerosol particles, and expansion cloud chamber simulations in which liquid cloud droplets were first activated on aerosol particles prior to freezing." In the first paragraph of the Methods section we state (again, slightly revised here in new form), "No measurements of contact freezing were included in this study, and neither will we discuss results herein from workshop measurements that were made in the regime associated with deposition nucleation (including at temperatures below the homogeneous freezing temperature or pure water droplets), but we will focus on inter-comparisons of particles acting via immersion freezing or proximal behaviors. By proximal behaviors, we follow the terminology of Vali et al. (2015),…" We simply could not pull all of the workshop studies into this overview paper, or it could have become unwieldy. We attempted to point out that the database remains for future papers by the participants, or inspection by any interested parties.

**Changes in manuscript re: Broad Comment [3]:**
In addition to the modest changes mentioned in our response above, we continue to add a direct statement at the end of the Introduction pointing out the emphasis decided for comparisons presented in this paper.
*"For these reasons, and to include as many measurements as possible in comparisons, we focus primarily on measurements relevant to immersion freezing nucleation, as discussed further below. This allows for integrating the most possible measurements into comparisons made for assessing one important aspect of the state of the art of ice nucleation measurements."*

**Broad comment [4]:** Unless I missed it, the promise made at 8/32 to describe the aerosol collection and transfer methods for testing as suspensions are missing from Section 2.4. Problems associated with the method used and the final efficiency of the capture are not found. See also 13/4-5.

**Authors' response to Broad Comment [4]:**

We thank Dr. Vali for pointing out that we indeed neglected to describe the details of particle collection into liquid.

**Changes in manuscript re: Broad Comment [4]:**

We have added this now to Section 2.3 (formerly 2.4). The segment below has been integrated into a revision of the overall discussion of experimental protocol for APC sampling. In doing so, some repetitive statements are now removed.

*"Collection of particles into liquid suspensions for shared use by a suite of immersion freezing devices was performed by impinging a flow of particles from the APC into a glass bioaerosol sampler (SKC Inc.) (Hader et al., 2014; DeMott et al., 2017), referred to here as impinger samples. Two impinger samples were collected for ~120 min with a flow rate of 12.5 L min$^{-1}$ from the APC. Flows were checked daily. Impingers were cleaned by wiping, rinsing with ultrapure water (18 MΩ-cm), and soaking in isopropyl alcohol overnight (2-propanol, ≥99.8%, ROTH). Before assembly the impingers were rinsed using ultrapure water once more. Following the sampling of Snomax$^{®}$ particles, the impingers were baked overnight at 200°C instead of soaking in alcohol. This was done to eliminate the possibility of carryover by ice active due to these biological samples. During sampling the water was replenished every ~30 minutes to keep the water level near 20 mL. Due to evaporation, the final bottled volume was typically about half of the added water. The two impinger suspensions were combined into one sample and topped off to a total of ~36 mL. The sample was divided into 4 ml aliquots, bottled in pre-cleaned DNA free cryovials and stored locally in a freezer at -20°C. The same procedure was applied to the handling of blanks. At the end of the campaign, blanks and samples were placed on dry ice and shipped overnight to participating groups. Shipment to Israel was delayed by customs, allowing the sample to thaw en-route. After receiving the sample, each group decided on their own sample storage and handling strategies. Again, we note that the University of Leeds group performed NIPI measurements of these suspensions on-site in Karlsruhe immediately after collection (i.e., without freezing)."*

**Broad comment [5]:** Some re-phrasing of conclusions such as in lines 6 to 23 on page 13 should be considered. The degree of agreement in the data is remarkable and deserves to be celebrated. But to say that the agreement extends over 7 orders of magnitude over the whole temperature range is misleading. Except in the center of the temperature range only 4-5 data sets are represented. Thus, in a strict sense, the agreement among all data sets is not quantitative over the whole range. They all coincide in defining a common trend and agree in overlap regions. None of the data sets extend over the whole range. Also, there is a division of the types of instruments for which data are available at the upper and lower ends of the temperature regime. The main novelty in these results is the agreement between CFD type and drop freezing types of measurements in the overlap region. This is a real accomplishment and

demonstrates that the two types of measurements can be combined to yield atmospheric measurements over a wide temperature range. The extent to which that is a practical solution for low INP air samples is worth addressing.

**Authors' response to Broad Comment [5]:**

5    We have revised the statements accordingly, since indeed, the point was to show the meshing of results for this particular sample, as well as correspondence overall within groupings. We alter the 7 order of magnitude statement for this reason, to reflect the context of what direct overlap is captured.

**Changes in manuscript re: Broad Comment [5]:**

10   The paragraph now reads: "*The comparisons obtained for sampling Argentinian soil dust particles (Fig. 4) were among the best in this study. A striking feature of these results is the general correspondence between all methods and sampling techniques in ranges of overlap, as well as the apparent meshing of results from direct sampling and post-processing of immersion freezing to capture 7 or more orders of magnitude of INP activity in the temperature regime from -5 to -35°C. Direct overlap showing correspondence of the continuous flow methods with a minimum of*

15   *four different bulk methods occurs over 3 orders of magnitude range at temperatures from -20 to -30°C. Good consistency is also seen amongst direct sampling methods as a group and post-processing methods taken together, for shared impinger samples, and whether post-processed samples were collected by impinger or separate polycarbonate filters (IS and FRIDGE-IMM) that were subsequently rinsed of particles. Recall that the IS filter was collected simultaneously with the impinger sample, while the FRIDGE-IMM filter was collected over a shorter time*

20   *frame. The largest discrepancies, in consideration of measurement uncertainties (see Supplement for explanation of measurement uncertainties for each device) occur at the coldest temperatures. In this region, data from the PIMCA-PINC instrument, which activates individually-grown droplets on particles prior to cooling, falls at the upper end of measured INP concentrations in comparison to the few other immersion methods that extended to the heterogeneous ice nucleation limit, just warmer than homogeneous freezing temperatures. We note again here that some scatter*

25   *may occur in the direct processing flow diffusion chambers due to investigators deciding in each case what $RH_w$ above 100% to report data for as representing immersion freezing of the entire particle population.*"

Additionally, we have added a discussion point in the Conclusions on the issue of low INP sampling. This was mentioned in regard to the likely need to pre-concentration INPs for direct processing at >-20°C. This is also a prime topic of FIN-03 and its overview paper. We will not add the Conclusion revision here, as the entire new Conclusions

30   are appended to the response to Reviewer 2.

**Broad comment [6]:** Only in S.2.1 is the method for calculations of the INP concentrations stated. How were the presented data derived for each device should be part of the description for each one. This also applies to how uncertainty ranges were determined.

**Authors' response to Broad Comment [6]:**

This comment was retracted in an updated note by the reviewer. As noted, this information is available in the extensive Supplemental material, repeating in most cases information that is already published.

**Changes in manuscript re: Broad Comment [6]:** None.

**Broad comment [7]:** A suggestion I make reluctantly, because of the amount of re-writing it implies, the authors should consider including a Discussion section, moving some material from Results there, and going a little further in some themes. Having a Discussion section would allow for better perspective on the various aspects of the results (e.g. 14/3-22), and give good opportunities for comparisons with previous publications (e.g. 18/2-28).

**Authors' response to Broad Comment [7]:**
We can see how this might work, and appreciate the suggestion. Nevertheless, we also believe that the flow of the paper, and the manner that the AIDA studies are used to continue the comparison of data and to then fold in APC results, favor the present structure. With the introduction of plots in $n_s$ space, it is natural to add the parameterizations and discuss them in place. It would take major rewriting to extract discussion and have it stand alone in an organized manner. Expanded discussion may not have an advantage over a more concise description of results and reference to previous studies.

**Changes in manuscript re: Broad Comment [7]:** We have renamed the section as "Results and Discussion" and have chosen to expand the Summary and Conclusions.

**Broad Comment [8]:** Parts of the description of results refer to the sigmoid shapes and the central slopes of the INP vs. T curves. I mention here that similar considerations, and a definition for a slope parameter were given in Vali (ACP 14, 5171- 5194, 2014.). The rich material in this paper could very usefully add and expand on the analyses in that paper. That may go beyond the scope the authors set for this paper, but in reality it would not be a large step.

**Authors' response to Broad Comment [8]:**
We agree that such an exercise could be useful, but as a science focused and separate topic. However, we wish to limit the scope of this present paper to intercomparisons and issues that arose during this focused objective, in line with the previous comment.

**Changes in manuscript re: Broad Comment [8]:** None.

**Broad Comment [9]:** There are many references to the Supplement. It would be helpful if, in each case, the reference pointed to some specific part of the Supplement.

**Authors' response to Broad Comment [9] and changes in manuscript re: Broad Comment [9]:**
We have checked all references to the Supplement. Most of the references were already to specific sections of the supplement or to the tables. The rest were found to reference uncertainties, in which cases we have added the words "for each specific device". Each section describes the manner in which these were defined, and most of these are uniform.

**Broad Comment [10]:** It is regrettable that the ACP (3.1) and AIDA (3.2) sampling results are in different forms, i.e. per air volume and as site density. Was this unavoidable? Considered unimportant? It comes as a surprise to the reader. Having the two different aerosol processing paths provided some assurance of minimizing problems related to each one. Perhaps there were other operational advantages or limitations as well. Yet, one would have like to see the results in the same format. For inter-comparisons the two data sets provide additional support and that is the main goal of the Workshop. But it isn't clear why the results need to be given in different manner. More careful reading of the paper may reveal the reasons, but perhaps readers (like me) could be helped by a statement of those reasons up front.

**Authors' response to Broad Comment [10]:** This decision has been discussed in response to previous comments. We understand now that we needed to describe the manner of operations of the workshop that was jointly determined by the workshop leads in consult with participants. The operational protocol determined what we deem as the best means of comparing the different chamber results. It was not meant to leave unrevealed any particular thing about comparisons. Indeed, the decision made to show the AIDA results in terms of active fraction (identical in all respects to a number concentration intercomparison) and $n_s$ was to permit explicit comparison of results from the two sampling periods for the same INP types (even though they were different aerosolization experiments, sometimes on different days). This led to the multi-panel figure comparing active fraction and $n_s$ values from both the AIDA sampling and from the APC, using shading in one panel to underlay one type of data and hopefully not making the figures unreadable. We decided that nothing especially revealing would be lost by not repeating the active fraction plot in each example give in Figures 8 to 11. The timing of sampling largely drove this decision, already mentioned above. There was no way to sample the INP concentration by all samplers, including AIDA, for each aerosol generation experiment.
**Changes in manuscript re: Broad Comment [10]:** See response to Broad Comment [1]. We definitely want readers to be able to navigate the paper smoothly, so we have additionally added small statements in the Methods section and reminders in the Results section to assist understanding why different formats are used for APC and AIDA sampling results, and how these are brought together ultimately in the later figures.

**Broad Comment [11]:** The focus on measurements of atmospheric INPs in the recommendations on page 22 is well placed. But ice nucleation measurements are also needed for fundamental understanding of ice nucleation and of the nature of INPs. Another important direction that ice nucleation measurements should explore in future workshops is to perform tests with mono-disperse aerosols.

**Authors' response to Broad Comment [11]:**

We definitely agree. However, producing monodisperse aerosols for many investigators is yet another logistical

challenge. Probably there will need to be a number of different and smaller workshop approaches with different

5     teams.

**Changes in manuscript re: Broad Comment [11]:** Both recommended topics, and others are listed in an expanded

Summary and Conclusions section, appended to the end of the response to Reviewer 2.

10    **Specific items**

**Specific item 1.** page/line 3/32: 'constrain' INP population???

**Specific item 1 author response/manuscript change:** Removed as superfluous.

15    **Specific item 2.** page/line 4/7: The fact that the more general aspects of the project are to be published later is a bit

of a problem. A brief description of the three stages would help.

**Specific item 2 author response:** While we eliminated this material as a possible distraction, we add some words

now about the different parts of FIN so that readers can understand the activity and the FIN-02 part of it. The

overview article is in preparation.

20    **Specific item 2 manuscript change:** In the Introduction, we now add, *"Briefly, and distinct from most previous*

*workshops in its comprehensive scope, FIN sought to perform comprehensive operational comparisons of ice*

*nucleation instruments for sampling calibration type INPs (representative of different atmospheric classes) in a*

*laboratory setting and for sampling ambient atmospheric aerosols in a natural setting. In addition, the component*

*FIN-01 (first study in late 2014) sought to intercompare single particle mass spectrometer instruments that are*

25    *sometimes used to assess the detailed chemical composition of INPs by sampling the residues of ice crystals*

*nucleated in flow diffusion chambers or aerodynamically segregated from atmospheric clouds. FIN-01 tested these*

*instruments for their determined reference mass spectra on some of the INPs also planned for use in comparing ice*

*nucleation instruments, it compared the different clustering algorithms used by the aerosol mass spectral*

*community, and it repeated testing on ice crystal residues. FIN-02, the workshop phase discussed herein, was the*

30    *laboratory ice nucleation instrument intercomparison. FIN-03, the field phase, was conducted at Storm Peak*

*Laboratory in Steamboat Springs, CO. A final aspect of both FIN-01 and FIN-02 was to provide a minor period*

*within the overall informal gatherings for scientific study that would feature a formal intercomparison of*

*measurements. FIN was a volunteer activity on the part of participants, who agreed to participate to their fullest*

*extent in both the informal and formal components, but were also free to explore new developments. Referees were*

35    *solicited for organizing and analyzing the results of formal comparisons in FIN-01 and FIN-02. These formal or so-*

*called "blind" experiments were conducted to investigate the degree to which the informal results presented in*

*papers such as this one could be independently reproduced."*

**Specific item 3.** Page/line 4/10-12: The order of these last two sentences of the paragraph should be reversed.

**Specific item 3 author response/manuscript change:** This section has been rewritten, as noted just above.

**Specific item 4.** *Page/line 4/22: What do you mean by 'align'?*

**Specific item 4 author response/manuscript change:** We meant "coordinate." Wording changed.

**Specific item 5.** Page/line 4/25-32: Goals are well defined. It would be nice if the conclusions responded clearly to each item.

**Specific item 5 author response/manuscript change:** We now explicitly focus our summary of results around the stated objectives in the new Summary and Conclusions section.

**Specific item 6.** Page/line 4/33: The intent to define a priori instrument characteristics is somewhat futile. Instruments that participated can be grouped as on-line and off-line but linking that too strongly to dry and wet is too much of a simplification. It would be more effective to describe the methods starting with Fig. 1 and then detail what the instruments do (above water saturation and immersion freezing) and do not do (deposition, condensation-freezing, contact). Indeed, reference to Vali et al, 2015 can help with the definitions.

**Specific item 6 author response/manuscript change:** The dry and wet designations have been used now in a number of prior published intercomparison studies. Nevertheless, we have refocused all discussion around direct and post-processed sampling systems. The noted sentence now reads, *"These categories are, firstly, instruments operating online or for direct processing of aerosol particles and, secondly, those utilizing collections of particles for subsequent offline or post-processing."* Additional small changes to the entire paragraph remove reference to wet and dry or online and offline, and clarifies what is measured in each case.

**Specific item 7.** Page/line 5/16: The operating principles are in in Tables 1. and 2 to some extent and are described in 2.1 and 2.2. The Supplement is to describe the detailed implementations of the basic idea in each device.

**Specific item 7 author response/manuscript change:** Wording is changed. The operating principles are already described in basic publications about most devices, so we attempted to reduce clutter. We write, *"Names, basic descriptions, and general operating principles are provided in Tables 1 and 2, and sections 2.1 and 2.2. Detailed implementations of the basic principles in each device are given in the Supplement sub-sections."*

**Specific item 8.** Fig. 1: The inclusion of X wt% designation here is unhelpful. The inclusion of this factor complicates interpretation. Indicated range may have been correct for the INPs chosen for test but have no general meaning. There are other materials that cover the whole temperature range with a single wt%. Adding the specific INP for which the graph is valid would add unnecessary detail. I suggest to leave those wt% indications off the graph. Of course, solute effects are ignored here.

**Specific item 8 author response/manuscript change:** We respectfully disagree, feeling that this point is useful for noting a fundamental difference in how the parameter space is covered by different instruments, even within the same general class. These values were indeed not intended to have a specific meaning except to indicate dilutions of samples to increase dynamic range, a general procedure that is needed for some sample types and droplet volumes. We write in the caption of Fig. 1, *"The blue arrow following the water saturation line in T-RH$_w$ space shows the trajectory of subsequently diluted samples (generically and schematically referred to as X, 0.X an 0.0X weight percent suspensions) of collected aerosols measured by immersion freezing methods. Such dilution is required in many cases for the laboratory samples tested, but the need for dilution or not also depends on the droplet size/volume used."*

**Specific item 9.** Page/line 6/14: The definition of max RH$_w$ practicable before 'breakthrough' is made difficult by the use of RH$_w$ for many different aspects of the processes.

**Specific item 9 author response/manuscript change:** We have streamlined this discussion by removing so many references to water relative humidity. We write, *"When the temperature gradient in the growth section is adjusted to generate water supersaturated conditions that activate cloud droplets within the aerosol lamina, the lower relative humidity in the evaporation section shrinks droplets back toward haze particle sizes. This method works up to some high value of RH$_w$ in the growth section whereupon activated cloud droplets survive through to detection, often referred to as the water droplet breakthrough RH$_w$. The breakthrough value varies with temperature, geometry and flow rate for different devices. Therefore, a single RH$_w$ level in the growth region for breakthrough to occur is not noted in Fig. 1. Instead, results are stated as being associated with specific RH$_w$ values (or % supersaturation values, which equal RH$_w$-100) that are simply a value that was lower than the droplet breakthrough condition. In some cases, this was the maximum RH$_w$ achievable in the growth region prior to droplet breakthrough."*

**Specific item 10.** Page/line 6/18-20: Is this caveat not in conflict with the goals defined earlier? May be just say that experiments included in the evaluations all stayed below the breakthrough RH w.

**Specific item 10 author response/manuscript change:** This is a good suggestion, and one we implement. The sentence is eliminated in preference to the rewrite of the sentences in the comment above.

**Specific item 11.** Page/line 6/31-33: This sentence should start the paragraph to explain why the choice of 102% needs discussion.

**Specific item 11 author response/manuscript change:** There was no intention of focusing on 102%, as it was simply put out that as a value that one might not expect to be relevant in the atmosphere. It is relevant for CFDCs. We have removed this sentenced and reorganized/rewritten this paragraph. It was too wordy and somewhat repetitive. This entire paragraph now reads, *"The focus on reporting of flow chamber data at highly supersaturated conditions as best representing proximal immersion freezing behaviors is motivated by recent research and publications. Presently, continuous flow diffusion chamber instruments in general do not expose particles to uniform water supersaturations with the precision achieved by cloud condensation nuclei (CCN) instruments. Rather, the*

*transition into the immersion freezing regime above water saturation does not occur sharply in line with the supersaturation calculated for the aerosol central lamina, but ensues completely only at higher $RH_w$ as controlled by aerosol particle properties and instrument characteristics. For example, hygroscopicity and kinetic factors control water uptake, chambers have different flow rates and growth section lengths, there is a finite difference in*

5     *$RH_w$ across the aerosol lamina, and many devices appear (for as yet unclear reasons) often to induce a proportion of all particles to escape the defined aerosol lamina and expose these particles to lower $RH_w$ outside of the intended central lamina (DeMott et al., 2015; Garimella et al., 2017). Hence, higher $RH_w$ is used in these instruments to bypass limitations in achieving CCN activation on the entire particle population, and to increase the condensation rate and thus water content of the formed droplets. The justification is to make the measurement conditions (most*

10    *particles placed in cloud droplets larger than a few μm prior to freezing) more similar to the cloud parcel simulations in the AIDA cloud chamber (see next section) with more typical cloud supersaturations and time scales. In practice, continuous flow instruments processed dry particle samples by slowly "scanning" $RH_w$ from near ice saturation conditions to water supersaturated conditions (see DeMott et al., 2011 for discussion of these methods, and the Supplement Section S.1.2 for a few examples. Investigators were then asked to select those data they felt*

15    *represented the highest (not necessarily maximum) immersion freezing activity possible to assess in their $RH_w$ scans, and reported the INP concentrations and $RH_w$ values selected."*

**Specific item 12.** Page/line 6/37-7/1: This is problematic. How is this known? The fraction of aerosol activated as a measure of the RH achieved in an instrument? The nominal RH and actual volume-weighted one differ, as just

20    described in earlier part of the paragraph.
**Specific item 12 author response/manuscript change:** We believe that the point may have been misunderstood. It was poorly stated. The point is that the instrument lamina RH cannot be interpreted as the exposure RH. As pointed out, this was somewhat repetitive, and so the paragraph has been rewritten. See above.

25    **Specific item 13.** Page/line 7/3-7/5: Recommendation here is out of place and heavy-handed.
**Specific item 13 author response/manuscript change:** We have removed this. It is a clear need for the field, but out of place for this section.

**Specific item 14.** Page/line 7/7: Shouldn't this 'primary comparison' be defined as a 'goal' up front?
30    **Specific item 14 author response/manuscript change:** We believe this was mentioned upfront, and only wish here to point out that additional data are available.

**Specific item 15.** Page/line 7/11: Quite unclear. Highest vs. maximum. Could the question be said to have been to select the RH point in the scans that the operators thought was most representative for freezing nucleation? Again,
35    this objective should be stated before discussing details of RH variations in the instruments. All those variations are caveats on the validity of the values chosen/used for the comparisons. In any case, isn't it likely that the activated

INP were in drops, due to the high hygroscopicity of the aerosol? In that case, the INP's don't experience 102% or whatever supersaturation. Clearly, these considerations need to be addressed here.

**Specific item 15 author response/manuscript change:** All of the INPs are not in the drops in CFDCs up to some RH that exceeds the expectation, and that is the point of the discussion, that every group operating such instruments has noted this. The suggestion is well taken though, and this section has been rewritten, as noted above (specific item 11 response).

**Specific item 16.** Page/line 8/5: Is this because only ice crystals forming from frozen droplets on the surface are counted? How certain is that? Same issue as in previous point.

**Specific item 16 author response/manuscript change:** We are not sure that we fully understood this comment, but hopefully some added words and reorganized discussion help here. We were only remarking that data will be included from the substrate-based diffusion chambers up to reported values of 101-102% RH with respect to water, but acknowledging that the equivalence of INP response to immersion freezing remains under evaluation, especially as this may depend on particle loading. We did not wish to gloss over this issue, only to limit extensive discussion of it when we are referring to two particular instruments. We have added overall to this discussion, including some past references, at least ones that reference the substantial literature on this topic, as note by the reviewer. See next comment.

**Specific item 17.** Page/line 8/7: " ... limits reduction of ..." is unclear. The influence of vapor competition, and its dependence on INP density on the substrate has extensive literature, and perhaps should not be glossed over so easily. The operational definition used here for making the comparisons deserves more explanation. The concluding sentence on 8/13-15 is unclear - do you mean cases when vapor competition was only inferred to have been present but not evaluated?

**Specific item 17 author response/manuscript change:** Upon review, only one of the submitted cases was removed from consideration by the DFPC group, and this related to potential heating (of the paraffin) impacts on Snomax activation, still under investigation. Hence, we have revised this section to emphasize matter-of-fact discussion of the two devices, and to mention that we are including them in this intercomparison of immersion freezing despite a continuing need to evaluate their ability to represent immersion freezing. We write, *"The thermodynamic path of measurements using these instruments is the same as for the continuous flow diffusion chambers in Fig. 1 (red lines), but typically terminate 1-2% $RH_w$ above water saturation. Both devices were designed with the intention to overcome the so-called "volume effect" on freezing (e.g., Bigg, 1990; Schrod et al., 2016 and references therein) which describes the underestimation of INPs that can occur when processing particles on a substrate in a diffusion chamber due to vapor pressure reduction by the first particles freezing, especially when larger volumes are collected that result in larger numbers of INPs per surface area of the substrate. The FRIDGE instrument seeks to limit this effect using a low-pressure diffusion chamber to enhance vapor deposition over particles collected onto silicon wafer substrates, while the DFPC instrument follows the methods of Langer and Rodgers (1975) to focus a flow of humid air over filter substrates, and using the best practices outlined by Bigg (1990). For both devices,*

*attempts were made to limit particle collections to shorter times during FIN-02, in order to keep particle loading light on the substrates. An additional fundamental difference between FRIDGE and the DFPC is the use of the filter substrate in the latter case, which is placed on a paraffin layer that is heated to establish thermal contact with a cold plate prior to ice nucleation measurements. The uncertain difference between condensation freezing and immersion freezing mechanisms (Vali et al., 2015) argues for an evaluation of results obtained near water saturation in this intercomparison as representative of proximal immersion freezing, for both instruments."*

**Specific item 18.** Page/line 8/8: '...were allowed to ..' inserts a hierarchical tone that is unnecessary. An objective tone would be more appropriate. The working arrangements of the workshop are not of interest to the reader. Else, make it clear and explain in the Introduction that there was some sort of checks and balances arrangement in order to improve the final result. I don't know if there was or not. See also comment on 7/3-5.

**Specific item 18 author response/manuscript change:** We have attempted to remove any hierarchical tone, and revise as noted above. Again, this was the informal component of the workshop, and the intent was to offer the best chances to compare instruments on a fair basis. We included as much of the data as was possible, and truly tried to limit and omissions.

**Specific item 19.** Page/line 8/14: "wet suspension" -- ?? wouldn't liquid suspension or water suspension be more direct? Actually, the first two words could well be omitted and the sentence say "Measurements of immersion freezing ...."

**Specific item 19 author response/manuscript change:** The latter suggestion would ignore that immersion freezing describes a process and not a stock measurement method, in our opinion. We here use *"immersion freezing measurements of collected particles suspended in water."* In other cases, we take the tip to simplify to *"water suspension"*.

**Specific item 20.** Page/line 8/31: " ... wet suspension groups ...? -- one can deduce what is meant but it could be said better

**Specific item 20 author response/manuscript change:** We have revised simply *to "Most groups using liquid suspension freezing methods shared common samples…"*

**Specific item 21.** Page/line 9/1: "... basic methods ... " may need to be re-considered if changes are made in response to the first item in this review.

**Specific item 21 author response/manuscript change:** Done, as mentioned above.

**Specific item 22.** *Page/line 9/29: suggest leaving out the word 'mode'*

**Specific item 22 author response/manuscript change:** Done.

**Specific item 23.** Page/line 10/3: suggest replacing 'wet dispersion of particles' by something like "aerosol generation from aqueous suspensions" or "particle dispersion from aqueous suspensions" or "dispersion of particles via spraying and drying"

**Specific item 23 author response/manuscript change:** We use, *"Aerosol generation from aqueous suspensions..."*

**Specific item 24.** Page/line 10/32: Was the depletion of aerosol in the APC by sample withdrawal avoided or neglected? Also, how justified was it to assume the decrease in concentration to be valid for all sizes?

**Specific item 24 author response/manuscript change:** Depletion of aerosol by sample withdrawal was in fact the primary mechanism for reduction in concentrations. The fits encapsulate this primary mechanism and wall losses.

10 Brownian diffusional losses to the walls was of course limited due to particle mode size being typically a few to several tenths of a micron in diameter. Hence, particle sizes were in the range where large differences in reductions as a function of size over time were not expected. This was a reason for showing a figure to include two different size ranges. We found that using total particle numbers versus using the concentration of particles > 0.5 μm led to no more than a 30% difference in the fractional losses of particles over time during the initial fills with higher total

15 aerosols. This is evident in Fig. 3. During the time period of direct particle sampling, the particle losses losing the same size ranges differed by no more than 10%, also evident in the figure. Hence, we assume a similar range of uncertainties on correction factors to bring the direct sampler INP concentrations into line with the period of sampling by post-processing devices. We have added statements in these regards. *"These loss processes were dominated by the drawing of air from the APC by samplers, replenished in all cases by clean synthetic air. Curves*

20 *are shown for total particle numbers, numbers of particles larger than 0.5 μm, and total particle surface area (spherical equivalency assumed for measured particle diameters). By integrating the exponential fit functions during sampling periods (blue shading), the integrated number concentrations and surface areas were determined for the combination of sample periods for post-processing. While we focus in the following discussion on quantifying the decay of total (CPC) particle numbers in order to correct INP number concentration data during the direct*

25 *sampling periods ("online" used as shorthand in equations) for equivalency with the prior post-processing collection periods ("offline" used as shorthand in equations), we noted (not shown) differences ranging from only 10-30% in fractional loss rates when instead using particle numbers in the larger size range (>500 nm) to characterize particle number decay over time in the APC. These relatively minor differences, evident in Fig. 3, are consistent with the limited physical loss mechanisms existing for particles with mode sizes as shown in Fig. 2, and*

30 *with limited numbers of supermicron particles that might be subject to sedimentation."*

**Specific item 25.** Page/line 12/30: perhaps the intention was to put the words 'basis for" in the sentence, to read : .... primary basis for the comparison of ...

**Specific item 25 author response/manuscript change:** Done.

**Specific item 26.** Page/line 12/31: replace 'most' by 'largest number' ??

**Specific item 26 author response/manuscript change:** Done

**Specific item 27.** Page/line 13/18-20: The sentence is somewhat garbled.

**Specific item 27 author response/manuscript change:** Given the extended earlier discussion, we have removed some discussion here and have revised this sentence to read, *"We note again here that some scatter may occur in the direct processing flow diffusion chambers due to investigators deciding in each case what $RH_w$ above 100% to report data for as representing immersion freezing of the entire particle population."* We may note to the reviewer that this topic will be featured within the referee paper on the formal workshop component of FIN-02.

**Specific item 28.** Page/line 14/3-22: These comments would be better placed in a Discussion, not in the Results section.

**Specific item 28 author response/manuscript change:** We have renamed the section to account for the inclusion of discussion.

**Specific item 29.** Page/line 15/12-14: I think it is too uncertain to explain results in terms of the numbers of proteins. Snomax contains full cells and evidence for separation of the INP material (protein) from the cell wall is unclear. Leveling off the INP curves shows a limit in the number of INPs as a fraction of the total number of cells. This is a matter of expression of the INP protein under induced conditions and of the processing of the bacterial culture. Again, the emphasis here should be on the agreement among measurement systems. Interpretation of the shape of the INP curves is a matter for Discussion (see Broad comments 7 and 8).

**Specific item 29 author response/manuscript change:** We were attempting to explain the behaviors in terms of discussion present already in past literature, but understand the point here and agree that we should omit this statement as tangential to the overall comparison.

**Specific item 30.** Page/line 15/25: Reference here to first-freezers etc. is confusing without more detailed knowledge of what is being discussed. This type of error analysis for specific devices should be part of the apparatus description (cf. Broad comment 6).

**Specific item 30 author response/manuscript change:** We were not discussing an error analysis in this section, but pointing to the fact that in some experiments, at least two of the freezing methods detected discrepancies in freezing conditions of their samples. We were also pointing out that such a result is consistent with the impact of handling on the first freezers detected on the basis of other published work.

**Specific item 31.** Page/line 15/32: Supporting Information =? Supplement. Where is the material referred to in this line?

**Specific item 31 author response/manuscript change:** Section S.2.2, stated now, to point specifically to the CMU methods.

**Specific item 32.** Page/line 16/10: Why wasn't the fraction expressed as active INP versus total number of potential INP particles introduced? Maybe that is what is meant, but I am not sure from the wording given here.

**Specific item 32 author response/manuscript change:** That is what was intended. The fraction was INP over potential INP for each type, representing a different experiment in each case. Reworded for clarity as, *"These data were first analyzed as active fractions, which is the number fraction of all particles freezing when normalized to the total number concentrations of particles (potential INP) present at the onset of expansions."*

**Specific item 33.** Page/line 16/16-17: The conversion to active site density is the same for all measurements, provided particle sizes are known. Why is this introduced here? (cf. Broad comment 10)

**Specific item 33 author response/manuscript change:** The active site density approach was introduced in Methods. It is mentioned here because we are about to present figures including it. The reasoning for this was provided in response to Broad comment 10. It was not possible to compare AIDA and APC experiments in any manner except on the basis of active fraction or $n_s$. We choose to show one active fraction plot and then proceed to reduce figures by showing only $n_s$. By overlaying the AIDA experiment $n_s$ data on $n_s$ calculated from the APC experiments, we seek to tie the two series together.

**Specific item 34.** Page/line 20/28-31: Do these two sentences refer to the same or two separate findings?

**Specific item 34 author response/manuscript change:** Rewritten for clarity. There are two points. The steep activation behavior with temperature exacerbates discrepancies. It was also the steep slope of the FS02 activation versus temperature that led to the finding that CFDC cooling in the evaporation region can express "late" activation of ice crystals that remain at small sizes and should not be attributed to the set point temperature of the lamina in the instrument. We write, *"The steep activation behavior of the FS02 also led to the finding that when sampling such INPs, cooling to achieve evaporation in the exit section of a CFDC (CFDC-CSU, CFDC-TAMU and INKA in this study) can express "late" activation of ice crystals that remain at small sizes and should not be attributed to the set point temperature of the instrument growth region. This may be an issue primarily for laboratory measurements of such INPs, since most natural INP T-spectral slopes…"*

**Specific item 35.** Page/line 21/21: Some numerical values for the errors discussed would be useful.

**Specific item 35 author response/manuscript change:** We are not sure how to answer this request, since it deals with a topic that was not investigated. Namely, we did not purposely attempt to create size distributions of INPs that would challenge instruments with size cuts. Furthermore, to suggest a range of errors that would occur in the atmospheric scenario, we would need to know a typical size distribution of atmospheric INPs. This could vary tremendously. The topic is appropriate for an ambient measurement intercomparison. It could be explored in a laboratory setting, but as mentioned, we did not explore it during FIN-02. Hence, we make no changes here.

**Specific item 36.** Page/line 21/23: 'de-agglomeration' has a simpler alternative: 'breakup' and removes the implication that all large INPs are aggregates of many smaller ones

**Specific item 36 author response/manuscript change:** Very good point, and accepted.

**Specific item 37.** Page/line 21/28: " ... full immersion of all particles in the same liquid volume .." is too vague to focus on, as other factors like time in suspension may also come into play. Also, are the differences beyond the error bars of the PIMCA-PINC and cold-plate methods?

**Specific item 37 author response/manuscript change:** We have added on breakup, sedimentation and active site alteration in bulk suspension as possible impacts on freezing spectra compared to PIMCA-PINC single droplet results. As for the last question, error bars are shown on all plots, so the answer is yes at the level of the experiments performed. If referring to the overall uncertainty evident in experimental results, it is harder to say. This topic would benefit from more overlap in the measurement regimes of the two methods for more instruments, but we have noted that PIMCA-PINC is restricted to assessing higher fractions freezing. We write, *"Reasons why the bulk immersion freezing methods do not always agree with PIMCA-PINC may relate in some unresolved manners to factors at play during extended bulk immersion, such as breakup, sedimentation, and alteration of active sites."*

**Specific item 38.** Page/line 21/32: why is that need artificial?

Agreed. It is artefactual, as in, not fundamental to the original operating principle.

**Specific item 38 author response/manuscript change:** We drop the artefactual part of that statement.

**Specific item 39.** Page/line 22/4: '…uniformly capture activation ...' is awkward wording

**Specific item 39 author response/manuscript change:** We have changed the word to *"equivalently"*.

**Response to anonymous Reviewer 2 comments on "The Fifth International Workshop on Ice Nucleation phase 2 (FIN-02): Laboratory intercomparison of ice nucleation measurements"**

We thank the reviewer for both the positive comments as well as detailed additional comments and corrections.

**General Comment:** As a general comment, I do think the Summary and Conclusion section could have expanded the "do's and don't's" a bit. The authorship represents a large number of those active in the field and I do not think it would be giving away any secrets if they discussed a bit what failed, or was discovered not to have worked during the campaign. Also, they do make a few suggestions for the future, but I am left wondering if the community came no closer to formulating certain experiments that could be done side-by-side anytime two or more instruments are co-sampling. In other words, does any suggested minimum protocol emerge to assess the agreement between instruments? Perhaps a complete picture and set of recommendations emerging from FIN-02 will be detailed in another publication?

**Authors' response to General Comment:** While the goal of the workshop was not necessarily to fashion protocol for all future such instrumental comparisons, nor to be judgmental, we agree with the reviewer that we could expand our recommendations, and point to things that could/should be done in a comparison involving any particular number of participants. At the same time, it is clear to those involved that the composite results of all three phases of the workshop will be needed to provide a complete picture and set of recommendations. Additional papers in that regard are forthcoming. We have completely reorganized and added to the Summary and Conclusions of this paper on the basis of the requests of both reviewers.

**Changes in manuscript re: General Comment:**
The Conclusions have been reorganized, first by summarizing and augmenting the initial discussion within the structure of the objectives listed in the Introduction of the paper, and secondly by adding a list of topics recommended for further research. As this section amounts to nearly 6 pages, we append it specially to the end of this response after first completing responses to itemized editorial comments.

**Itemized Editorial Comments (page/line):**

*Comment 1*. *Page/line* 1/38 item b only has a departmental affiliation it is missing an institutional affiliation.
**Comment 1 author response/manuscript change:** Thanks, corrected.

*Comment 2*. Page/line 6/16-18 What comes after the comma at: "...Fig. 1, and why..." does not make sense as written. The why results are given at certain RH is not noted? I think the clause needs to be rewritten.
**Comment 2 author response/manuscript change:** We have rewritten this section for clarity: *"Therefore, a single $RH_w$ level in the growth region for breakthrough to occur is not noted in Fig. 1. Instead, results are stated as being associated with specific $RH_w$ values (or % supersaturation values, which equal $RH_w$-100) that are simply a value that was lower than the droplet breakthrough condition. In some cases, this was the maximum $RH_w$ achievable in the growth region prior to droplet breakthrough."*

*Comment 3*. Page/line 7/28 I recommend you change "in this manuscript" to 'presented here'.
**Comment 3 author response/manuscript change:** Changed as requested.

**Comment 4:** Page/line 7/30 This first sentence of the section is long and awkward. I recommend it is changed. It can be split into 2 sentences.
**Comment 4 author response/manuscript change:** Done

**Comment 5:** Page/line 8/4-5 thermodynamic paths not path • 8/10 would limit particle loading
**Comment 5 author response/manuscript change:** Changed as requested.

**Comment 6:** Page/line 8/18 activation temperatures instead of "temperatures of activation". Also, add 'and referenced' after "...are listed". I was looking for the instrument references as I was reading and did not realize they were in the table until getting there. Finally, I suggest striking "Details on all of" rather begin the next sentence with "The specific ..."

**Comment 6 author response/manuscript change:** Changed as requested. Reference to Table 1 was also added on Page 5.

**Comment 7:** Page/line 9/1 I recommend "besides" is changed to 'in addition to'

**Comment 7 author response/manuscript change:** Changed as suggested.

**Comment 8:** Page/line 9/3-5 The sentence beginning "In this regard..." is awkward and should be rephrased. Is it the recommendation of the 2007 workshop, or previous workshops and it was followed in 2007?

**Comment 8 author response/manuscript change:** Rewritten as, *"In this regard, we follow the example of the 2007 workshop, and a key recommendation from ice nucleation workshops prior to that time that an expansion cloud chamber be utilized to provide a simulation of cloud activation (DeMott et al., 2011)."* This is discussed in the 2011 paper and references to prior workshops are provided.

**Comment 9:** Page/line 9/5 recommend change to 'Schematic thermodynamic paths of the AIDA chamber experiments are shown by the ....'

**Comment 9 author response/manuscript change:** Changed as suggested.

**Comment 10:** Page/line 9/20 suggest '....cloud regime (i.e., 0 to -36 C). Thus they provide....'

**Comment 10 author response/manuscript change:** Changed as suggested.

**Comment 11:** Page/line 10/12 suggest wording is changed to 'However, in some experiments larger particles were present ...'

**Comment 11 author response/manuscript change:** Changed as suggested.

**Comment 12:** Page/line 10/13 should "some cases" be 'those cases'

**Comment 12 author response/manuscript change:** Corrected.

**Comment 13**: Page/line 12/2 I suggest that when ns,geo(T) is defined the units [m−2] are included.

**Comment 13 author response/manuscript change:** Done.

**Comment 14**: Page/line 13/6 suggest wording is changed to 'A striking feature of these results is the general correspondence between all methods ...'

**Comment 14 author response/manuscript change:** Changed as suggested.

**Comment 15:** Page/line 13 /14 suggest 'The greatest discrepancies' replace "Greatest discrepancies..."
**Comment 15 author response/manuscript change:** Changed to "The largest…"

**Comment 16:** Page/line 13/15 "Supplemental" should be 'Supplement'
**Comment 16 author response/manuscript change:** Done.

**Comment 17:** Page/line 13/19 suggest wording is changed to '...understanding of what RHw value might...."
**Comment 17 author response/manuscript change:** Changed as requested.

**Comment 18:** Page/line 13/23 The final sentence of the paragraph ending, "... Fig. 4 is encouraging" is oddly placed in a results section. I think it is extraneous.
**Comment 18 author response/manuscript change:** Removed.

**Comment 19:** Page/line 14/15 replace "at the same time" with 'simultaneously'
**Comment 19 author response/manuscript change:** Changed.

**Comment 20:** Page/line 14/27 replace "Comparison" with 'A comparison'
**Comment 20 author response/manuscript change:** Done.

**Comment 21:** Page/line 15/17-20 This is a very long and confusing sentence, I suggest rewording.
**Comment 20 author response/manuscript change:** This section is rewritten as, *"All immersion freezing methods capture the strong rise in activation due to the presence of the most active biological ice nucleators, those within the realm of the Groups I and II as defined by Yankofsky et al. (1981). This is expressed as a pronounced shoulder in all freezing spectra at temperatures warmer than -8°C in Fig. 7. We may note here that all bulk freezing methods shared the same impinger sample in this case, including the IS. This warm shoulder of ice nucleation activity has also been demonstrated by Wex et al. (2015), Budke and Koop (2015), and Polen et al. (2016)."*

**Comment 22:** Page/line 15/29 Should it not be CMU-CS?
**Comment 20 author response/manuscript change:** Yes, corrected.

**Comment 23:** Page/line 15/33 awkward wording. The source of variations question?
**Comment 23 author response/manuscript change:** Rewritten as, *"In summary, the strong variability in activity seen at the warmest activation temperature regime for Snomax® particles brings into question the ability to utilize the warmest temperature (> -10°C) freezing behavior of Snomax® reliably for calibration purposes. This has been*

*noted previously by Polen et al. (2016), and attributed to batch-to-batch variability and the loss of activity during storage."*

**Comment 24:** Page/line 22/1 "provide the decision" is awkwardly worded.

5 **Comment 24 author response/manuscript change:** Replaced with "…provide guidance…"

 **Comment 25:** Figure 3 caption – The caption fails to define what the differently colored points indicate. Furthermore, I suggest the 2 light blue shadings be somehow differentiated. If my understanding is correct then I think the 2nd sentence would read, "An initial impinger and filter sampling period is highlighted in light blue 1 and

10 is followed by an APC refill and subsequent sampling period (light blue 2). In any case I think the caption should be carefully reread and reworded for clarity.

**Comment 25 author response/manuscript change:** We had hoped for the colored scales to direct eyes to the appropriate axes, but for clarity we have rewritten it all as, *". Data from the first half of a daily experimental series, after an initial fill of Tunisian soil dust (Experiment 11 and 12 of the FIN-02 APC series on March 18, 2015) into*

15 *the APC at a time (time 0) taken to be the experimental start time. Impinger and filter sample periods after two separate chamber fills are highlighted in light blue shading. To obtain a reference aerosol concentration ($cm^{-3}$) via the CPC (total particles) (blue points, left scale) or optical particle counter at sizes larger than 500 nm (n_500; brown points, right scale), as well as surface area ($\mu m^2\ cm^{-3}$) (black points, left scale) for the impinger/filter sampling period, the time-weighted average of the two sampling periods was determined. This period-integrated*

20 *concentration value could then be used to ratio versus the concentrations of particles present at later sampling times during the direct processing instrument sampling of aerosols from the APC (green shaded region) in order to back-correct the directly-sampled INP number concentrations to those derived for the collections for post-processed samples."* The text section describing the procedure is already quite detailed, and does mention the fact that there were sometimes two sample chamber fills, and sampling periods for which the integrated particle number values and

25 surface areas are determined.

 **Comment 26***:* Figure 4 caption – Suggestions: (1) strike "on some points" (2) replace "buried within" with 'subsumed by' (3) add '...errors) or alternatively the binned...'
**Comment 26 author response/manuscript change:** Changed as suggested.

**Comment 27:** Figure 7 caption – bring "in this case" to beginning of sentence
**Comment 27 author response/manuscript change:**
Done.

35 **Comment 28:** Figure 8 may be better presented vertically for such a 2-column journal? Perhaps the authors intend to stretch this over the entire page? Currently the text seems somewhat small. (also Figures 9-11).

**Comment 28 author response/manuscript change:** Due to the length of the caption, we will suggest to place these figures across the entire page, unless the editors suggest otherwise. And although the text is somewhat small, there were many different instruments to indicate, and it should be possible for most to expand the figures in PDF format. But again, we will go with the decision of the copy editors in this case.

**Comment 29:** General figure comments: Can the chosen color schemes for the instruments remain consistent between those figures that represent APC and AIDA data? Also, I think making many of the figures box plots with at least the major tick marks represented on the minor axis would be more reader friendly.

**Comment 29 author response/manuscript change:** It was indeed the intent to maintain color schemes for different
10 data across figures. We will recheck that this is done. Once we began to introduce AIDA data at Figure 8 and beyond, and when we desired to show both soil dusts together in Figure 10, it complicated matters, so we simply made all of the online instruments black (or green and black in the case of Fig. 10). This also assisted with shading the APC versus AIDA sampling points in the later figures. We revise now to use black symbols for flow chambers all the way through the figure series, and we distinguish IS and FRIDGE-IMM series as blue in the APC figures
15 now. The exception to this overall scheme will continue to occur in Figure 10, where green is used to distinguish the two dusts amongst the direct sampling instruments. As for the scaling on box plots, it would be too difficult to add the extra scale in the AIDA figures and simultaneously keep the long legends that are perhaps for important for easily distinguishing instruments. For that reason, we do not add the secondary scales on any of the figures.

20 **New Conclusions section:**

[revised manuscript text omitted]

**S.1 Direct sampling Online instrument systems**

Online INP measurement systems are cloud chambers or instruments using continuous flow systems to assess INPs. Most have been reported on in the literature or will be in the near future, and some have a record of reported data of more than 20 years. Most of these systems are documented to the extent that they could easily be reproduced, and in some cases certain ones have a legacy in an earlier version of one type. This will be noted in the descriptions below.

**S.1.1 AIDA (Aerosol Interaction and Dynamics in the Atmosphere) cloud simulation chamber**

The Aerosol Interaction and Dynamics in the Atmosphere (AIDA) cloud simulation chamber, which consists of an 84 $m^3$ isothermal aluminum vessel, is a comprehensive experimental facility that can be used for recreating supercooled clouds in the vessel (Möhler et al., 2003). More specifically, the chamber conditions are precisely controlled by mechanically pumping air in the vessel, inducing concurrent and homogeneous reduction of both gas temperature and pressure. The resulting so-called expansion cooling provides a wide range of simulated atmospheric in-cloud conditions, such as temperature (60 to -90°C with an uncertainty ± 0.3°C), pressure (~1000 to below 1 hPa) and relative humidity (from ~0% to above water saturation). In such cloud simulation experiments, spontaneous droplet activation and ice crystal formation occur in simulated supercooled clouds at or above saturation with respect to water and ice (e.g., Möhler et al., 2005; Niemand et al., 2012). Further, AIDA is unique since its cloud experiments are systematically performed with atmospherically relevant droplet sizes (i.e., a few to tens of micrometer diameter at the largest) as well as under atmospherically relevant cooling rate; i.e., an average cooling rate of 1.7 ± 0.1 (standard error) °C $min^{-1}$.

Besides its central function as a cloud emulator, the AIDA chamber can be also utilized as a platform for multiple instruments to investigate aerosol-cloud interactions and cloud microphysics in ice nucleation experiments (*DeMott et al.*, 2011). During FIN-02, similar to previous AIDA studies (e.g., Wagner et al., 2011; Hiranuma et al., 2014a and 2014b), a combination of 6 online instruments, including (1) a condensation particle counter (CPC, TSI, Model 3076), (2) a scanning mobility particle sizer (SMPS, TSI, Model 3080 DMA and Model 3010 CPC), (3) an aerosol particle sizer (APS, TSI, Model 3321), (4) the white light aerosol spectrometer optical particle counters (Welas-OPCs, Palas, Sensor series 2300 and 2500; Benz et al., 2005), (5) a tunable diode laser (TDL) water vapor absorption spectroscopy (Fahey et al., 2014) and (6) a home-built device for scattering intensity measurement for the optical detection of ice (hereafter SIMONE - German abbreviation for Streulicht-Intensitätsmessungen zum optischen Nachweis von Eispartikeln; Schnaiter et al., 2012), used to characterize the physical properties of aerosol and hydrometeors (droplets and ice crystals) formed during the AIDA expansion experiments with a detection limit for number concentration of 0.1 $cm^{-3}$. It is noteworthy that, in dense supercooled liquid clouds, the AIDA-TDL data is offset by +5% in order to match the liquid-phase saturation (Murphy and Koop, 2005) conditions expected for the immersion freezing event. The reason for this systematic deviation of the TDL measurement from the expected saturation is unclear. More technical details on individual instruments and their applications at the AIDA facility are given in above listed publications.

**S.1.2 CFDC-CSU (Continuous Flow Diffusion Chamber - Colorado State University)**

The Colorado State University (CSU) Continuous Flow Diffusion Chamber (CFDC) operating principles are described in the earlier works of Rogers (1988), Rogers et al. (2001) and Eidhammer et al. (2010). The current versions of the CFDC-CSU used in ground based (CFDC-1F) and aircraft studies (CFDC-1H) are geometrically identical and composed of cylindrical walls that are coated with ice via flooding and expelling water from the chamber when the walls are set at a controlled temperature of ~-27°C before each experimental period. The plate separation is 1.12 cm prior to ice application, which has a typical thickness of 0.015 cm. The chamber is divided into two sections vertically, separated by a Delrin collar. A temperature gradient between the colder (inner) and warmer (outer) ice walls in the upper 50 cm "growth" section creates an ice supersaturated field into which an aerosol lamina is directed. Vapor pressure relations used within the analytical equations given in Rogers et al. (2001) are taken from Murphy and Koop (2005). The Delrin inlet manifold has a stainless-steel knife-edge ring threaded into it, so that aerosol flow is directed centrally between two sheath flows of clean and dry air. The ratio of aerosol and sheath flows can be varied, but typically the aerosol lamina represents 15% of the 10 L min$^{-1}$ total flow. Ice crystals forming on INPs in the growth region of the chamber enter the lower 30 cm "evaporation" section of the chamber where the two walls are held equivalently to the cold (inner) wall temperature. As shown by DeMott et al. (2015), residence time in the growth region is approximately 5 s under conditions used in the present study, although residence at prescribed steady state conditions is probably on the order of 3s, followed by 2 s in the evaporation regime. When the temperature gradient in the growth section is adjusted for water supersaturated conditions that activate cloud droplets in the aerosol lamina, these will evaporate to haze sizes in the evaporation section, at least up to some water relative humidity ($RH_w$) where they survive, referred to by many as the droplet breakthrough $RH_w$. Until that high $RH_w$, only ice crystals and haze particles will exit the CFDC. Ice crystals and aerosols exiting the CFDC at sizes above approximately 500 nm are counted with an optical particle counter (OPC), where the two populations are distinguished in different size modes. For the data collected in this work, we count all particles in size bins above 3 μm as ice particles when not encountering droplet breakthrough. The cut-channel used for analysis of activated ice crystals at a calibrated 3 μm size was channel 50 for FIN-02. In usual operation, aerosol particles larger than 2.4 μm are removed by a set of inertial impactors prior to the chamber inlet to eliminate misidentification as ice crystals, but the impactors were removed for all data reported in this paper. Data archive files indicate times when impactors were used in selected experiments. Some experiments with the impactors in place will be reported in the paper summarizing blind inter-comparisons.

CFDC-1F measurements for FIN-02 were ideally made via slowly scanning $RH_w$ (~1% min$^{-1}$) at single temperatures, including below and above water saturation to identify the maximum freezing activity prior to the point that water droplets "breakthrough" the lower evaporation section. The $RH_w$ of droplet breakthrough was also identified whenever possible, but this higher $RH_w$ was not always achieved. Figure S1 shows an example for which d$RH_w$/dt was kept to close to 0.5 % min$^{-1}$.

[Figure]

**Fig. S.1.** Example of data from a slow $RH_w$ scan (~0.5% min$^{-1}$) with the CFDC-CSU instrument while sampling illite NX and processing it at a temperature of -20°C on March 13, 2015. Each OPC channel size spectra (at the $RH_w$ values in the legend) in the left panel or single point in the right panel represents a 10 s interval. In the left panel, the spectra indicating water droplet breakthrough (WDBT) are indicated by the progressive appearance of a high concentration

5  mode of particles becoming more prominent as $RH_w$ is further increase from 111 to 114%. This point is clear also in the cumulative apparent INP concentration at sizes larger than the standard cut-point for ice at 3 μm (Chnl 50). INP concentrations are referenced for this study primarily at 105% $RH_w$, which is computed as an average over the range 105 to 106%. As previously reported (DeMott et al., 2015), the maximum INP concentration prior to WDBT is a factor three to four larger than values referenced to 105% $RH_w$ in this case. The comparability of Chnl 50 and Chnl 165 data

10  at $RH_w$ up to 105% makes it clear that most of the first ice crystals nucleated grow to large sizes. This is the basis for using Chnl 165 in cases where aerosol particles were present already at sizes >3 μm before the start of an $RH_w$ scan.

Higher ice cut-point sizes were applied in cases where sufficient large aerosol particles were present initially to "pollute" standard ice size channels. This was a peculiarity of the laboratory study that is not encountered in

15  atmospheric sampling. A higher cut-size for ice was selected to derive INP concentrations from these data because it was expected that large aerosols might retain some water after liquid particle activation and thereby show up even at sizes between channel 50 and some larger channel in the OPC at the exit of the CFDC. The appropriateness of this procedure is demonstrated for a typical experiment Fig. S1. In all cases, the larger cut-size selected and reported is listed as channel 165 in the CFDC-CSU archive files. The cut size employed in each experiment is also noted in Table

20  S1. Note that the channel size is not linearly related to particle size, but typically follows a square relationship to channel, and hence, channel 165 is likely in the range of 5 μm. This was not calibrated. An example of correction for aerosols polluting typical ice cut-size channels by using Chnl 165 for ice definition is shown in Fig. S2.

[Figure]

**Fig. S2**. As in Fig. S1, but for the Tunisian dust experiment on March 18 (Fig. 5) in which aerosols were already present at "ice" channel sizes in the APC when the $RH_w$ scan was initiated in the CFD-CSU instrument (without the upstream impactor in place to remove >2.5 μm particles). In this case, WDBT occurs sooner, despite the slow d$RH_w$/dt scan. After subtracting the aerosol background, the Chnl 50 data (standard "ice" channel) resemble the Chnl 165 data.

5    Nevertheless, to be conservative, the Chnl 165 data is used to define ice in these cases. It can be seen that the Chnl 165 data also allows definition of a maximum INP concentration in this case.

It was noted in a few experiments, most notably for K-feldspar, that what appeared similar to droplet breakthrough even in the absence of aerosol "pollution", occurred at low water supersaturations. We hypothesize this to be due to

10    small ice crystal formation occurring for evaporating droplets in the lowered temperature region of the evaporation section of the CFDC-CSU instrument, a consequence of both exceeding desired $dRH_w$/dt rates and the steep activation function of K-feldspar versus temperature. This is demonstrated in Fig. S3, where OPC data from contrasting AIDA chamber sampling experiments on March 20 and March 23, the two days plotted in Fig. 11 of the manuscript, are shown. The discontinuous and strong rise in ice activation at moderate supersaturation, especially noted on March 23

15    when $dRH_w$/dt exceeded twice the desired rate for scanning, had not been seen in previous studies (DeMott et al., 2015), just as it is not evident in Fig. S1 or S2. That is, normally, INP freezing in the growth region of the instrument will grow to larger sizes, and not be present frozen at near to the size of activated cloud droplets in the CFDC growth section. While Fig. S3 shows that use of Chnl 165 may derive the most appropriate INP concentrations for relating to the CFDC processing temperature in these cases, Chnl 50 data is plotted in Fig. 11 to emphasize this discovered CFDC

20    sampling issue for K-feldspar in the temperature region near -20°C. $RH_w$ scan rates exceeded 1% min$^{-1}$ in two other K-Feldspar experiments on March 18, one in the APC and one in the AIDA chamber, so these data are not included in Fig. 6 and Fig. 11. Although this situation is considered unusual, as scanning RH$_w$ is not a practice that is often operationally practiced in the field (i.e., constant values or steps of RHw are used), these findings may motivate testing reconfiguration of the CFDC-CSU instrument so that the evaporation section presents warming instead of cooling.

25    This could be accommodated simply by configuring the inner cylindrical wall to be the warmest wall instead of the coldest wall.

[Figure]

**Fig. S3.** Spectral and cumulative INP > specific channel plots as in Fig. S2, but for the K-feldspar experiments used to define the low (left panels, March 20, 2015) and high (right panels, March 23, 2015) CFDC-CSU INP concentration data points in Fig. 11 of the manuscript. The occurrence in these cases of "small ice" by 5% water supersaturation is distinct from Fig. S1 despite the absence of aerosol "pollution" of the initial spectra and similar d$RH_w$/dt rates. The unexpected population of small ice crystals are believed to result from "late" freezing as liquid particles cool before completely evaporating upon entering the evaporation section of this instrument. In the experiment on March 23, d$RH_w$/dt was three times faster, far above the desirable rates. Less steady control of $RH_w$ led to even stronger growth of the small ice crystal mode and likely overestimation of INP (via the standard Chnl 50 ice point definition) attributable to the processing temperature of -20°C.

Interval periods occurred during $RH_w$ scanning in which the aerosol sample was filtered in order to determine background frost influences on ice particle counts in the OPC, as described in prior publications (e.g., DeMott et al., 2015; Schill et al., 2016). Following Schill et al. (2016), sample period concentrations are those with the interpolated background concentrations of adjacent filter periods subtracted. The standard deviation derived from Poisson counting statistics from both the sample and the interpolated background concentrations were added in quadrature to obtain the INP concentration error. Concentrations are considered significant if they are 1.64 times larger than the INP concentration error, which corresponds to the Z statistic at 95% confidence for a one-tailed distribution.

Particle losses in the aerosol impactor and the inlet manifold of the CFDC have been previously estimated as 30% of total condensation nuclei when sampling ambient air (Rogers et al., 2001), but only 10% for aerosols in the 100 to 800 nm size range based on laboratory tests (Prenni et al., 2009). We apply the 10% correction to all CFDC data herein. Temperature uncertainty is ± 0.5°C at the reported CFDC lamina processing temperature (Hiranuma et al., 2015). $RH_w$ uncertainty depends inversely on temperature, and has been estimated as ± 1.6, 2 and 2.4 % at -20, -25, and -30°C, respectively (Hiranuma et al., 2015).

**S.1.3 CFDC-TAMU (Continuous Flow Diffusion Chamber - Texas A&M University)**

The Texas A&M Continuous Flow Diffusion Chamber (CFDC-TAMU), including a Cloud Aerosol Spectrometer with Polarization (CASPOL) is an apparatus designed to measure the concentration, backscatter, and depolarization ratio of ice crystals nucleated under controlled conditions. The CFDC, built at Texas A&M University, measures the nucleation and growth of ice crystals under well-controlled temperature and supersaturation conditions (Rogers et al., 2001; Glen and Brooks, 2014; McFarquhar et al., 2011; Zenker et al., 2017). Leaving the CFDC, samples enter the CASPOL (Droplet Measurement Technologies, Inc.) which counts the ice crystals activated on INIPs and determines the optical properties of individual ice crystals at 680 nm (Glen and Brooks, 2014). An inlet at the top of the CFDC allows for an aerosol stream to enter into an annular chamber. Here, the sample air passes between two laminar flows of dry filtered air. The walls of the chamber are coated with ice and held at different temperatures in order to create a controlled supersaturation field between the walls. The aerosol sample has the potential to nucleate and form ice crystals as it travels through this controlled supersaturation region.

[Figure]

**Figure S4:** Schematic of the CFDC-TAMU with CASPOL illustrating how the particles are analyzed by the CASPOL after exiting the CFDC.

After nucleation and growth of ice crystals have occurred in the CFDC, the ice crystals are counted by the CASPOL based on size discrimination. The CASPOL employs 3 detectors which measure light scattered in the forward (4°-12°) and backward (168°-176°) directions. The concentration and size of IN are determined by the forward

scattering detector. Backscatter light is split between two detectors, which measure the parallel and perpendicularly polarized light, respectively. The back detectors are used to determine the backscattering intensity and the depolarization ratio of individual ice crystals (Fig. S4).

During FIN-02, TAMU CFDC experimental uncertainty in concentration was +/- 39% based on instrument uncertainties in sheath and total flow rates, particle losses within the CFDC and in the transition zone between the CFDC and CASPOL, background corrections, icing thickness, and CASPOL uncertainties (choice of bin size, coincidence of multiple particles). In FIN-02, uncertainty in supersaturation arose mainly from the possible separation of the hydrophobic material on the lower section of the warm (inner) wall (used to induce evaporation) by as much as 2 mm. Assuming laminar flow considerations and that the hydrodynamic flow is fully developed, this causes an uncertainty in SSw of +/- 1.75%. The use of spheres to calibrate the CASPOL may lead to uncertainties in particle sizing for aspherical particles. In an earlier version of the CFDC-TAMU, sample air flowing from the CFDC to the CASPOL was observed to warm, causing an estimated ice loss corresponding to a small decrease in radius of less than 1% below ~ -50°C and unquantified changes in the associated optical properties (Glen and Brooks, 2014). Prior to FIN0-2, the connection between the CFDC and CASPOL has been modified to eliminate this warming.

**S.1.4 INKA (Ice Nucleation Instrument of the Karlsruhe Institute of Technology)**

The new INKA (Ice Nucleation Instrument of the Karlsruhe Institute of Technology) is a continuous flow diffusion chamber that is based on the design of Rogers (1988) and was built in cooperation with Colorado State University. The INKA instrument is most specifically modeled after the Colorado State University "laboratory" CFDC (Archuleta et al., 2005) and is described in detail in Schiebel (2017). INKA is a longer column version of the CSU CFDC-1H, but with cooling baths for temperature control. INKA's ice nucleation chamber consists of concentrically aligned copper tubes that have been ebonized for wetting purpose and are individually cooled via external chillers to temperatures as low as -60°C. During operation, the side walls of the 1 cm annular gap between the copper tubes are coated with a 0.4 mm ice layer. A total flow of 12.5 L min$^{-1}$l passes through the chamber. This flow consists of 5% to 10% of sample flow encased in 90% to 95% of particle free sheath air. The sample to sheath flow ratio was adapted during the FIN-02 measurements according to available aerosol concentration and fresh synthetic air was used as for sheath flow.

The total length of the INKA chamber is 150 cm, with the upper 2/3 acting as so called "nucleation and growth section". Here, the sample is exposed to defined water vapor levels above ice saturation by setting both walls to different temperatures. Following Rogers (1988), the temperature and relative humidity in the sample lamina is calculated for the measured wall temperatures, flow velocity and sample to sheath ratio. Depending on the chosen conditions, deposition and/or immersion freezing mode nucleation can be studied. During FIN-02, we stepwise increased the relative humidity at the sample location while keeping the laminar mean temperature constant. For most samples, we studied a range from 90% to 110% $RH_w$ with respect to liquid water. INP concentrations close to 105% $RH_w$ were used for intercomparison with other immersion freezing methods. In between the investigated $RH_w$-plateaus, the chamber background was measured by effective filtering of aerosols from the sample flow.

The lower 1/3 of the outer wall can be cooled separately from the upper section. Here, a Delrin spacer inhibits thermal contact between the outer wall copper pieces. For the FIN-02 measurements, we chose to couple the cooling of the inner wall and the outer lower wall, thus setting those to equal temperatures (colder than the upper outer wall). With entry into this so called "droplet evaporation section", the ice particles, any existent droplets and interstitial aerosol particles quickly adapt to the abruptly lowered vapor pressure. As the sample is still held at ice saturation, the formed ice crystals are able to grow further while droplets will shrink due to evaporation, thus enabling the identification and counting of ice particles by an optical particle counter (Climet CI-3100) at the chamber outlet.

Although the air samples taken from AIDA or the APC were rather dry, we used silica gel equipped diffusion dryers to ensure consistent conditioning. To prevent the erroneous classification of large inactivated aerosol particles as ice particles by OPC counting, an impactor with a well-defined flow dependent cut-off size of about 2 μm was used in some experiments. To account for particle loss in the impactor, the particle number concentrations were measured with and without the impactor in place, at a position before the chamber inlet, using a condensation particle counter (TSI model 3772). Particle losses in the sampling line were found to be negligible.

A detailed description of the INKA instrument will be given in Schiebel (2017). This paper will also include a thorough analysis of measurement uncertainties. For the FIN-02 campaign we estimate an uncertainty in temperature of 1 K and an uncertainty in particle number concentrations of 20%.

**S.1.5 PINC (Portable Ice Nucleation Chamber – ETH Zurich)**

The Portable Ice Nucleation Chamber (PINC, Chou et al., 2011) is a parallel-plate vertical continuous flow diffusion chamber and the portable version of the Zurich Ice Nucleation Chamber (ZINC, Stetzer et al., 2008). The general operational principle follows that of Rogers (1988) as described in the previous sections. Prior to the ice nucleation experiments, an ice layer is applied to the chamber walls (568 x 300 mm) which have a distance of 1cm between them. A temperature gradient is set between the chamber walls that generates a parabolic supersaturation profile with a peak saturation close to the center plane. The sample aerosol is introduced with a flow rate of 1 lmin$^{-1}$ and layered between two particle-free sheath air flows of 4.5 lmin$^{-1}$ ensuring a narrow, centered sample lamina. Aerosol particles are introduced into the chamber and may nucleate and grow to ice crystals during a residence time of 4-5 s in the ice nucleation and growth section before entering the evaporation section, where both walls are isothermally set to the warm wall temperature. This creates a subsaturated environment with respect to water in which any formed droplets evaporate while ice crystals are maintained at the ice saturated conditions. Exiting aerosol particles and ice crystals are detected at the bottom of the chamber by an optical particle counter (Lighthouse R5104). In this study, particles larger than a set size threshold of 3 μm are counted as ice crystals and no impactor was used upstream of the chamber to limit larger aerosol particles from being sampled. Measurements were performed as $RH_w$ scans (<2 % RH min$^{-1}$) at prescribed temperatures from ice saturation to above water saturation up to an $RH_w$ at which ice crystals cannot be distinguished from droplets based on their size (droplet breakthrough). Above water saturation conditions condensation freezing and immersion freezing cannot be distinguished in this setup. Before and after each scan, background concentrations of ice crystals in the chamber are obtained by sampling filtered air. The background counts are linearly interpolated between two filter periods and subtracted from the sample signal and the INP concentration

is determined. The INP active fraction is calculated as the ratio of ice crystals detected with the OPC to the number of total aerosol particles measured with a CPC on the aerosol chambers (AIDA or APC).

The accuracy of the temperature sensors is ±0.1°C and the variation of the temperature across the theoretically defined sample lamina is ±0.4°C, which corresponds to an uncertainty in $RH_w$ of ±2 % (Chou et al., 2011). The uncertainty in INP concentration is 10 % due to counting by the OPC and uncertainty in active fraction is 14% including an additional uncertainty of 10 % for the measurement of the total aerosol particle concentration. Chamber characterization experiments with PINC revealed particle losses below 5 % without the use of an impactor upstream of PINC (Boose et al., 2016).

**S.1.6 PIMCA-PINC (Portable Immersion Mode Cooling Chamber - ETH Zurich)**

PIMCA is a vertical extension of the PINC instrument and used to investigate the ice nucleating ability of aerosol particles explicitly in the immersion mode (Kohn et al., 2016). After entering PIMCA aerosol particles are activated to cloud droplets at 40°C in supersaturated conditions with respect to water ($RH_w$>115 %) by applying a temperature gradient of $\Delta T = 25°C$ between two chamber walls with constantly wetted filter paper. The droplets with single-immersed aerosol particles are then supercooled to the desired ice nucleation temperature prior to entering PINC, which is held at water saturation conditions. Typical flow rates in the PIMCA-PINC setup are 0.6 L min$^{-1}$ sample flow and 2.2 L min$^{-1}$ particle-free sheath air on either side of the aerosol lamina. A frozen fraction of entering cloud droplets is calculated by the ratio of ice crystals to the number of total particles (cloud droplets and ice crystals) in the sample volume using the Ice Optical DEpolarization detector (IODE, Nicolet et al., 2010), which is attached to PINC. Measurements with PIMCA-PINC are performed by scanning the temperature from homogeneous freezing conditions (e.g., T<233 K) until the detected frozen fraction is not distinguishable from the experimental background at water saturation conditions in the entire sample lamina.

Each reported data point consists of about 2-5 individual measurements at one temperature, which corresponds to more than 3000 measured single particle intensity peaks. The sample concentration is diluted upstream of the chamber to avoid coincidence errors for appropriate peak detection and the frozen fraction is the primary data set. For calculation of the INP concentration, the frozen fraction is multiplied by the total aerosol particle concentration in the aerosol chambers measured in parallel.

Temperature and RH uncertainties are equivalent to those for PINC. The uncertainties in frozen fraction are based on the uncertainty from the potential false classification of ice crystals as cloud droplets and vice-versa. Measurements with PIMCA-PINC were conducted on the same sample line as has been used for PINC and are discussed at the end of section 2.4.

**S.1.7 CIC-PNNL (Compact Ice Chamber – Pacific Northwest National Laboratory)**

The Pacific Northwest National Laboratory (PNNL) Cloud Ice Chamber (CIC) has been previously described by Friedman et al. (2011) and Kulkarni et al. (2016). This device is modeled after the ice nucleation chamber described by Stetzer et al. (2008), and is the predecessor design on which the SPectrometer for Ice Nucleation (see later sections) was modeled. The chamber consists of two parallel plates through which flow is directed in a downward, vertical

direction, with an evaporation section attached at the bottom of the chamber to remove water droplets (Stetzer et al., 2008). The chamber plates are coated with an ice layer ~0.5 mm thick and independently temperature controlled using two external cooling baths (Lauda Binkmann Inc.). Temperature data are logged using a National Instrument CompactRIO programmable automation controller at 1 Hz sampling rate. Cooling baths were operated such that a linear temperature gradient was developed across the chamber plates. The gradient was adjusted to produce the desired ice-supersaturated conditions inside the chamber, and relative humidity with respect to ice ($RH_i$) and liquid water was calculated using the Murphy and Koop (2005) vapor pressure formulations. The chamber design ensures that aerosol particles are placed between the layers of two sheath flows. The sheath and sample flows were 6 and 1 L min$^{-1}$, respectively, which resulted in a particle residence time of the sample stream to ~12 s within the chamber. For FIN-02, the CIC-PNNL instrument was operated for at least part of the time in scanning $RH_w$ mode, wherein the thermal gradient between the chamber walls was slowly increased in a manner that also held the average temperature of the aerosol stream constant. Temperature and $RH_i$ uncertainty limits are ±1.0°C and ±3%, respectively. Ice crystals that grew to sizes greater than ~1.5 μm were detected with an optical particle counter (OPC, CLiMET, Model CI-3100). Background INP counts were calculated by sampling only filtered dry air for at least 15 min at the beginning, middle, and at the end of the experiment. Background INP counts were subtracted from the measured INP counts in each experiment. INP concentrations reported are interquartile means.

**S.1.8 SPIN-MIT (Spectrometer for Ice Nuclei – Massachusetts Institute of Technology)**

The MIT SPectrometer for Ice Nuclei (SPIN) is a commercially-available ice nuclei counter manufactured by Droplet Measurement Technologies in Boulder, CO, technically described by Garimella et al. (2016). SPIN is a continuous flow diffusion chamber with parallel plate geometry similar to the Zurich Ice Nucleation Chamber (ZINC) (Stetzer et al., 2008) and the Portable Ice Nucleation Chamber (PINC) (Chou et al., 2011). Ice supersaturation conditions are created by coating two parallel plates with ice, holding them at different temperatures (both below 0°C), and owing an aerosol lamina in the center of a sheath flow between the walls. After flowing through the main chamber, the air stream enters an isothermal evaporation section to evaporate liquid droplets. It then passes through a linear depolarization optical particle counter (OPC) for particle, droplet, and ice counting. The instrument can be operated at aerosol temperatures as low as -55°C and ice supersaturations exceeding 60%.

SPIN was operated for FIN-02 in both a "static" mode, where the wall temperatures were controlled to provide constant aerosol temperature and supersaturation conditions, and in a "ramp" mode, also referred to as $RH_w$ scan mode in this paper, where the wall temperatures were set to diverge from a starting temperature, resulting in increasing supersaturation at a relatively constant temperature. Ramps were generally performed such that dT/dt for each wall was 0.5°C min$^{-1}$. In all cases, the evaporation section is isothermal and held at the desired aerosol temperature.

The aerosol temperature and supersaturation in the lamina are calculated using the method in Rogers (1988). Uncertainties in the temperatures and supersaturations experienced by the aerosol arise from inhomogeneity in wall temperatures, uncertainties in flows, differences in conditions across the width of the aerosol lamina, and deviations from the analytical calculations in Rogers (1988). Below water saturation, aerosol particles and ice crystals are distinguishable by optical size and light depolarization. At certain conditions above water saturation, dependent on

temperature, droplets do not evaporate fully in the evaporation section and can enter the OPC. Size and depolarization signals are used to distinguish particles, droplets, and ice (Garimella et al., 2016).

**S.1.9 SPIN-TROPOS (Spectrometer for Ice Nuclei – Leibniz Institute for Tropospheric Research)**

The SPIN-TROPOS is nearly identical in configuration, operation, and uncertainties to the SPIN-MIT. A description of the chambers design and functionality can be found in Garimella et al. (2016). During an experiment, a continuous flow of aerosol particles is exposed for 10-12 s to controlled ice supersaturated conditions and temperatures, allowing ice nucleation to occur on the aerosol under investigation. Ice crystals grow subsequently on the fraction of particles acting as ice nuclei. The number of ice crystals formed at a specific condition is detected optically by an OPC measuring the intensity of light scattered by individual particles. The OPC signal of ice crystals is distinguished from the signal of aerosol particles by a higher intensity, caused by the larger size of ice crystals. Sequences of increasing ice saturation ramps up to above water saturation, at several constant temperatures were run during FIN-02. Experimental uncertainties in temperature and saturation, derived from the wall temperature inhomogeneity during each measurement are typically below +/-1K and +/-5%, respectively.

**S.2 Instrument systems for post-processing of bulk particle collections**

All of these instrument systems employ cooled surfaces or wells holding particles exposed to water vapor, or water droplets/water volumes. Each are unique, and many are documented already in the literature, as noted herein.

**S.2.1 NCSU-CS (North Carolina State University Cold Stage)**

The design of the NC State cold stage-supported droplet freezing assay and data reduction methods are described in Wright and Petters (2013), Hader et al. (2014) and Hiranuma et al. (2015). For the experiments reported here, aqueous suspensions from impinger collections were distributed on the cold stage as follows. Approximately 64 drops of $V = 1\ \mu L$, measured with an electronic micropipette, were placed directly on a hydrophobic glass slide. These drops are in contact with a gas-phase composed of dry nitrogen. Squalene oil to immerse the droplets, which was used in previous studies of the NC State CS was not applied. A constant cooling rate of 2°C min$^{-1}$ was applied and the fraction of unfrozen drops was recorded using a microscope camera at incremental $\Delta T = 0.17$°C resolution. Droplet frozen fractions versus temperature data were inverted to first determine the concentration of INPs using the method of Vali (1971):

$$c_{IN}(T) = -\frac{\ln\left(f_{unfrozen}\left(T\right)\right)}{V_{drop}} \tag{S1}$$

where $c_{IN}(T)$ is the concentration of INPs per unit volume of water (m$^{-3}$), $f_{unfrozen}$ is the fraction of unfrozen drops at $T$, and $V_{drop}$ is the population-median drop volume.

Volumetric INP concentrations in air ($C_{INP}(T)$) were calculated via,

$$c_{INP}(T) = \frac{c_{IN}(T)\cdot f\cdot V_{imp}}{V_a} \tag{S2}$$

where $V_{imp}$ is the total impinger water volume collected for distribution to other investigators, $f$ accounts for the dilution of the impinger water ($f = 1$ for undiluted), and $V_a$ is the air volume collected into liquid.

Samples were stored frozen at inside a -80°C freezer in Raleigh before the experiments, which were performed between April and September 2015. For each dilution, the experiment was repeated three times. In addition, experiments with diluted impinger water and on sample blanks for quality control were performed. This resulted in sample data from 4 to 9 individual experiments for a given collection. Results from these experiments were binned into 1°C temperature intervals; 95% confidence intervals for $c_{INP}$ are reported for the binned data.

**S.2.2 CMU-CS (Carnegie Mellon University Cold Stage)**

The Carnegie Mellon University Cold Stage (CMU-CS) is composed of an air-cooled cascade 3-stage thermoelectric chiller (TEC) unit (TECA, AHP-1200CAS), topped with a custom-built aluminum stage described further by Polen et al. (2016) and Beydoun et al. (2017). The stage houses an external single-stage thermoelectric element (TE Technology Inc., VT-127-1.4-1.5-72P) and an associated thermistor (TE Technology Inc., MP-3176) placed beneath a removable aluminum sample dish. The thermistor provides temperature measurement for all experiments and is calibrated as described below. Droplet samples are placed on hydrophobically-coated coverslips (Hampton Research, HR3-231) in an inert squalene oil (VWR, H0097, ≥98.0% purity) environment to prevent contamination and droplet interaction. The chiller's plastic lid encloses the entire aluminum chamber that is placed on the cascade TEC to insulate it from ambient conditions. Dry air is flowed over the top of the plastic lid to prevent fogging, and a beaker of desiccant is used to dry the air inside the enclosure. The cooling ramp cycle is controlled by the TE Tech software. The single-stage element is held at 10°C until the experiment begins; it then begins to ramp to 0°C for 1 min. After this 1 min, the temperature is set to ramp down at 5°C intervals every 5 min to -40°C, producing a 1°C min⁻¹ cooling rate. The cascade chiller that acts as the heat sink for the single-stage TEC is set to -45°C throughout the experiment.

Temperature calibration of the system was performed by attaching a thermistor to a hydrophobic coverslip, placing it into the sample dish, and covering it in oil. The temperature was ramped down at 1°C min⁻¹ using the same program as a typical freezing experiment. The thermistor temperature was recorded at 1 Hz frequency. This was repeated multiple times to insure similar temperature ramp rates for all experiments. Following these ramps, a plot of the thermistor temperature measurement and the system measurement (using the same model thermistor, placed in its usual location in the aluminum stage under the sample dish) is generated. The linear relationship between the system temperature and the cover slip temperature is used to correct the measured system temperature to that of the cover slip during normal droplet freezing assays. Occasional tests of the cover slip temperature during a normal cooling cycle were performed to confirm that the relationship remained the same.

FIN-02 samples were kept frozen at ~ -10°C in their original containers until the droplet freezing experiment was run. Samples were thawed until no ice remained in the sample, but not completely to room temperature. A small volume of the stock sample was then poured into a sterile, unused secondary plastic vial to avoid contamination of the original sample. The original sample was then returned to the freezer. The secondary volume was used for that day's experiments exclusively and then disposed of. The secondary volume was not re-frozen between experiments to avoid repeated freezing and thawing of the sample. The secondary sample was hand shaken briefly to re-suspend any particles that may have settled out.

For Snomax® (FIN02-15-J) dilution experiments, the stock sample was poured into a second vial and returned to the freezer. From the second vial, 100 µL was pipetted into a third vial, which was immediately stored in the freezer until the experiment was performed. Shortly before the cooling experiment, the 100 µL sample was thawed completely and diluted to 50 mL (500X dilution) to examine the colder freezing temperature ice nucleants. The diluted sample was hand-mixed briefly, immediately generated into droplets, and the cooling cycle was started.

The secondary volumes of solutions were used exclusively for droplet generation to avoid contamination of the stock sample. A p2 variable electronic pipette (SEOH, 3824-1LC) was used to create 0.1 µL droplets in squalene oil on top of a hydrophobic glass coverslip. New sterile pipette tips were used between different droplet arrays to avoid contamination. Each droplet was placed on top of the oil and allowed to sink to the bottom to rest on the coverslip. Each array contained 40-60 droplets.

Pure water samples were from an in-house Milli-Q system (18.2 MΩ·cm) which is run for at least 5 min before obtaining a sample. Fourteen independent pure water arrays containing more than 600 droplets total were subjected to the standard droplet generation and temperature program. 10% of droplets had frozen by -25°C and 50% were frozen at ~ -30°C.

For uncertainty analysis, 2-3 replicate droplet arrays were measured for each sample, and their freezing temperature spectra were averaged to produce the averaged droplet freezing spectrum determined from Eqs. (S1) and (S2). A new droplet array was generated from the same secondary volume for each replicate run. The error bars are the 95% confidence intervals determined from the replicate runs for each sample. The Snomax® sample was separated into two different average spectra, due to differences observed between the replicates. The first freezing cycle, performed as soon as droplets were generated immediately after the sample had thawed, was separated from the second and third freezing cycles on new droplet arrays generated from the same secondary volume. The average spectrum from the diluted Snomax® sample was combined with the average of the 2nd & 3rd runs to produce the complete Snomax® spectrum. An additional first freezing cycle was also performed on a later day, by re-thawing out the Snomax® stock sample.

**S.2.3 IS (Colorado State University Ice Spectrometer)**

The Colorado State University Ice Spectrometer (IS) emanates from the developments of Hill et al. (2014; 2016) and is described in the approximate form used in this study by Hiranuma et al. (2015). Immersion freezing temperature spectra are obtained in the IS following dispensing 24 or 32 aliquots of 50 or 60 µL of suspensions of aerosols into sterile, 96-well PCR trays (Life Science Products Inc.) in a laminar flow cabinet. The IS is constructed using two 96-well aluminum incubation blocks (VWR), designed for cooling or heating PCR plates, placed end-to-end and encased on their sides and base by cold plates (Lytron). A ULT-80 low temperature bath (Thermo Neslab) circulating SYLTHERM XLT heat transfer fluid (Dow Corning Corporation) is used for cooling. Loaded PCR plates were placed in the blocks, the device covered with a plexiglass window and the headspace purged with 0.5-1.5 L min$^{-1}$ of filtered (HEPA-CAP, Whatman) nitrogen. Temperature was lowered at 0.33°C min$^{-1}$, measured using a thermistor verification probe (Bio-Rad, Hercules, CA, VPT-0300) inserted into a side well. Frozen wells were counted at 0.2-1°C degree intervals to a limit of -27°C, and cumulative numbers of INPs mL$^{-1}$ of suspension were estimated using Eq. (S1). INPs

per volume of air processed were calculated with Eq. (S2), where $V_{imp}$ in this case was either of the impinger sample or of the filter suspension sample. For filter samples, filter blanks were processed in a similar manner as aerosol samples to obtain a mean background INP spectrum. Binomial sampling confidence intervals (95%) were derived using the formula recommended by Agresti and Coull (1998):

$$\text{CI}_{95\%} = \left( \hat{p} + \frac{1.96^2}{2n} \pm 1.96\sqrt{\left[\hat{p}(1 - \hat{p}) + \frac{1.96^2}{4n}\right]/n} \right) / \left(1 + \frac{1.96^2}{n}\right) \tag{S3}$$

where $\hat{p}$ is the proportion of droplets frozen and $n$ is the total number of droplets. Using this formula, for a single well frozen out of 32 aliquots the $\text{CI}_{95\%}$ ranges from 18% to 540% of the estimated INP concentration, while for 16/32 wells frozen it is 68-132% of the INP concentration.

Most results reported in this paper were from filter samples, rather than the shared impinger samples. APC air was typically filtered for 120-130 min at 15 L min$^{-1}$ through a 47-mm diameter in-line aluminum filter holder (Pall) fitted with a 0.2 μm diameter pore Nuclepore polycarbonate membrane (Whatman). These were collected at the same position as impinger samples, and the collection times typically aligned. Dis-assembled filter holders were cleaned by soaking in 10% H$_2$O$_2$ for 60 min followed by rinses in deionized water (18 MΩ-cm and 0.2 μm diameter-pore filtered) and removal of excess water with a gas duster before drying. Filters were prepared in a laminar flow cabinet (<0.01 particles cc$^{-1}$) by soaking them in 10% H$_2$O$_2$ for 10 min followed by three rinses in deionized water, the last of which had been filtered through a 0.02 μm pore diameter filter (Anotop 25 mm syringe filter, Whatman) and drying on foil.

After particle collection, filters were transferred using clean, plastic forceps to a sterile, 60 mm petri dish (CELLTREAT) and stored frozen at -20°C. For re-suspension of particles, filters were placed in sterile 50 mL Falcon polypropylene tubes (Corning Life Sciences), 6-10 mL of suspension solution added and particles re-suspended by tumbling end-over-end on a Roto-Torque (Cole-Palmer) at 60 cycles min$^{-1}$ for 20 min. The re-suspension solution was 2 mM KCl (to prevent any influence on the ice nucleation activity of K-Feldspar) filtered through a 0.02 μm pore diameter filter (which contained, on average, 1.6 INPs mL$^{-1}$ at -25°C). A series of up to five 20-fold dilutions in 2 mM KCl were used to cover the full temperature range.

For the Snomax® comparison as well as for the blind studies (Hoose et al., 2017), the IS processed samples from the impingers. A sub-sample of the NCSU impinger water was melted and tested neat and after dilution in 20-fold steps in 0.02 μm pore filter deionized water.

**S.2.4 μL-NIPI (Leeds Microliter Nucleation by Immersed Particles Instrument)**

This instrument has been previously described in detail by Whale et al. (2015). Briefly, approximately 40 droplets of 1 μL volume are pipetted onto a hydrophobic glass slide (Hampton Research HR3-23) using an electronic pipette (Picus Biohit). The glass slide is placed onto an Asymptote EF600 Stirling cryocooler, which is used to control the temperature of the slide. The slide is enclosed within a Perspex chamber and a gentle flow of dry nitrogen is used to prevent water condensation during cooling. Freezing of droplets is monitored using a digital camera, allowing the fraction of droplets frozen at a given temperature to be determined. Samples were used for freezing experiments immediately following collection at the AIDA facility. All experiments presented here were conducted at a cooling rate of 1°C min$^{-1}$. Temperature error was calculated by taking the random error of the thermocouple used to measure

cold stage temperature, propagated with the melting point range observed for water, resulting in a maximum error of less than ± 0.4°C. Due to the ice nucleation induced by the slide and other sources of contamination there is a lower limit to the temperature at which this instrument can be used. In order to account for this effect a background freezing curve has been produced, which is subtracted from the cumulative nucleus spectrum for individual experiments. In this way freezing events which are not unambiguously caused by the heterogeneous nucleator under investigation are eliminated from the dataset. This process is described in O'Sullivan et al. (2015). Uncertainties in INP concentration for the binned data shown in this paper were derived from the experiment-to-experiment variability.

**S.2.5 BINARY (Bielefeld Ice Nucleation ARraY)**

A detailed description of the BINARY setup is given by Budke and Koop (2015). Briefly, the setup as it is used in this study consists of a compartment array for 36 droplets ($V_{drop}$ = 0.6 µL) positioned on a Peltier cooling stage (Linkam LTS120). Each compartment is composed of a lower hydrophobic glass slide with contact to the drop, a polydimethylsiloxane (PDMS) spacer at the sides, and an upper acrylic glass. The drops are cooled down at a constant rate of 1 K min$^{-1}$. Freezing temperatures are determined optically based on the change in brightness when the transparent liquid drops become opaque during freezing. The temperature uncertainty is ±0.3 K.

The suspension droplets were positioned onto the hydrophobic glass slide using an electronic pipette (Brand Transferpette®, accuracy ≤ ±1.0%). To minimize sedimentation the drops were pipetted in rows of 3 x 12 drops and the suspension was stirred with a vortex mixer (VWR) at 1000 rpm in advance of each row placement.

Due to contact with the lower glass slide and dust contaminations the apparatus has an applicable temperature range from 273 K down to about 245 K (25$^{th}$ percentile freezing temperature of "pure" water). Based on the background freezing temperatures experiments with significant overlap were excluded from further analysis. Additionally, single data points were excluded if they did not satisfy the following rules: 1.) number of INPs per litre of examined suspension (diluted) $[L^{-1}] > 10^{(51.24 - 0.186 \cdot (T [K]-1))}$, 2.) number of INPs per litre of examined suspension (diluted) $[L^{-1}] > 10^5$.

Data subsampling was done by temperature binning into 1 K intervals giving the midpoint of each bin. INP concentrations $c_{IN}$ for each bin are shown in terms of median (50$^{th}$ percentile) values. Error bars indicate the 5$^{th}$ and 95$^{th}$ percentile, respectively. Please note that especially for samples as Snomax® where INP concentrations increase strongly with decreasing temperature the large error bars do not stem from a significant spread of the data but rather from the steep increase in $c_{IN}$ within a 1 K temperature bin.

**S.2.6 M-AL (Mainz Acoustic Levitator)**

For a detailed description of the Mainz Acoustic Levitator see Diehl et al. (2014). The M-AL consists of an ultrasonic trap (APOS BA 10, tec5 AG, Germany) in which single water droplets of 2 mm in diameter were levitated; the imaging digital video camera which served for determining the drop size; and an infrared thermometer (KT 19.82 II from Heitronics) which measured the drop surface temperature (ΔT (2σ) = 0.5 K) continuously. The M-AL was placed in the walk-in cold chamber of the Mainz vertical wind tunnel laboratory in which the air temperature was cooled down to –26±2°C, and monitored by a platinum resistance (Pt-100) thermometer. For each measurement, a

single drop was generated using a medical syringe and injected into the M-AL. Prior to injection the sample was hand shaken briefly in order to avoid any sedimentation. When injecting, the drop temperature was approximately +10°C which decayed continuously adapting to the ambient temperature. The onset of drop freezing in the M-AL is characterized by a rapid increase of the drop surface temperature ensuing from the latent heat release as the phase change is initiated. Thus, the freezing temperature could be determined in the experiments from the lowest surface temperature recorded by the infrared thermometer. For each collected impinger sample, 22 to 60 individual drops were measured without diluting the delivered FIN-02 suspensions, and the fraction of frozen drops, $f_{ice}$, was calculated. The only exception was the mystery sample M2-D used for blind studies (Hoose et al., 2017), which was diluted to 1:9 (i.e., 10%). Since the Snomax samples were also measured undiluted, i.e., the INPs were in high concentration, they initiated freezing at high temperatures (just below zero °C). It resulted in a very limited number of observed freezing events, and large freezing temperature uncertainties. Because of these insufficient statistics, the Snomax results from M-AL are not presented here.

FIN-02 samples were kept frozen inside a refrigerator until the freezing experiment was carried out. Samples were thawed at room temperature until no ice remained in the sample. The whole sample volume in the sample tube was used for that day's experiments exclusively.

The frozen fractions were binned into 1 °C temperature intervals giving the midpoint of each bin, and the concentration of active sites per liter water was calculated using Eq. (S1). The error of $c_{IN}$(T) was derived from,

$$\Delta c_{IN} = \sqrt{\left(\frac{3 \cdot \ln(1-f_{ice})}{\frac{\pi}{6}d_t^4}\Delta d_t\right)^2 + \left(\frac{1}{1-f_{ice}} \cdot \frac{1}{\frac{\pi}{6}d_t^3}\Delta f_{ice}\right)^2} \tag{S4}$$

where $\Delta f_{ice}$ represents the error originating from the temperature uncertainty (the error due to the drop volume uncertainty on $\Delta f_{ice}$ was neglected), and was calculated from the number of drops frozen within a temperature interval of 0.5 $\Delta T$ (the factor 0.5 comes from the fact that the drops were continuously cooled down, thus, only one direction of temperature uncertainty was taken into account). $\Delta d_t$ is the uncertainty of the drop size which was determined from the individual drop images, and represents a 2σ error.

The INP per liter air was calculated from Eq. (S3). Since no errors are assumed in the values of the $V_{imp}$ or $V_a$, the error of INP (per liter) in air was derived from,

$$\Delta INP = \frac{\Delta c_{IN} \cdot V_{imp}}{V_a} \tag{S5}$$

where only the uncertainty from $\Delta c_{IN}$ was taken into account.

**S.2.7 KIT-CS (Karlsruhe Institute of Technology Cold Stage)**

The central part of the experimental setup of the KIT-CS includes a Cold Stage (Linkham, Model MDBCS-196), which was used to carry out temperature ramp experiments with defined cooling rates. Cooling is achieved by pumping liquid nitrogen from a reservoir to the sample holder.

A silicon substrate (Plano GmbH, 10x10mm) for supporting droplets was first cleaned with high grade acetone (p.a.), then rinsed several times with NanoPure® water. Finally, the silicon wafer was purged with nitrogen to remove residual water. The cleaned silicon wafer was mounted into a copper basin on top of the sample holder.

A piezo injector (GeSIM, Model A010-006 SPIP) was filled with aqueous suspensions. Before printing, the substrate was cooled to the ambient dew point to reduce the evaporation of droplets. Up to one thousand identical suspension droplets (0.5 nL) were printed onto the silicon wafer, resulting in 100μm drops in spherical cap geometry. After printing, the droplets were covered with silicone oil (VWR, Rhodorsil® 47 V 1000,) to prevent any interaction
5    between supercooled and frozen droplets. Typical ramp experiments started from 0°C to -40°C with the cooling rate of -1K min⁻¹ and followed by heating the sample up to +1°C.

To accurately determine the temperature of the droplets a calibrated thin film platinum resistance sensor (Pt-100) was directly fixed on the surface of silicon substrate by applying the small amount of heat conducting paste. The Pt-100 was calibrated in the temperature range from -40°C to +30°C prior to the experiment.

10    A charge-coupled device (CCD)-camera (EO® progressive) with a wide field objective (DiCon fiberoptics Inc.) was used to visualize the droplets. The substrate was illuminated by a circular light emitting array installed around the objective lens. Two polarizers (one in front of the light emitting diode and one in front of the objective) were used to detect the frozen droplets. A video (AVI) and temperature file were recorded, allowing for identification of individual freezing events with 0.125s temporal resolution and 0.1K temperature accuracy. Subsequent data processing with a
15    LabView® routine allowed for calculation of a fraction frozen curve.

For data processing, the number of INP per liter water was calculated according to Eq. (S1). The initial concentration of impurities was obtained from experiments with NanoPure® water. To estimate the background prior to homogenous freeing limit a 3rd degree polynomial was applied to pure water data. Equation (S2) was used to calculate the number of IN per liter air. The scaling factors were calculated by dividing the volumes of the impingers
20    and the added water during operation through the sample volume. For the impingers, a sampling efficiency of 100% was assumed.

**S.2.8 VODCA (Vienna Optical Droplet Crystallization Analyzer)**

The Vienna Optical Droplet Crystallization Analyzer (VODCA) device has at its core a cryo-microscope cell that consists of a single-stage Peltier element (Quick-cool QC-31-1.4-3.7M) mounted on a copper cooling block on ice
25    water sewage, placed in an airtight cell, that can be flushed with dry nitrogen between measurements (Pummer et al., 2012). A glass window in the cover of the cell allows observation of the sample via a light microscope (Olympus BX51M) and an attached camera (Hengtech MDC320), linked to a computer. The temperature measurement has a standard deviation of 0.5 K. Samples were measured as emulsion with 90 wt% paraffin and 10 wt% lanolin (water-free grade) as oil phase. The emulsion was created directly on a thin glass slide with the help of a pipette tip. We use
30    slightly more oil than liquid phase. Droplets diameters ranged from 20-41 μm. Prepared samples were placed on a glass slide and set onto the Peltier stage, where they were chilled with a cooling rate of approx. 10 K min⁻¹. Each slide was measured at four different positions with all measured droplets being summed up to one freezing curve. This was done twice for each sample. Droplet sizes were divided in three groups regarding their diameter to minimize the error of their non-uniformity (20-26 μm, 26-35 μm, 35-41 μm). Calculations of INP concentrations in air were made
35    following the procedures used by other investigators.

**S.2.9 WISDOM**

The **WeIzmann Supercooled Droplets Observation on Microarray** (**WISDOM**) is an instrument designed to study immersion freezing down to the homogeneous freezing temperature region, which combines a cryo-optic-stage with microfluidics techniques for fast generation of static picoliter to nanoliter droplet arrays (Reicher et al., 2018). The droplets are generated in a flow-focusing junction, and trapped in chamber arrays following their generation. In this study, the microfluidic device was based on a design by Schmitz et al. (2009). Each experiment contained about 500 droplets with diameter of 30-40µm, suspended in an oil phase, and a cooling rate of 1 K min$^{-1}$ was applied. Temperature uncertainty of ±0.3 K was estimated. The suspension was sonicated for 5 min prior to droplet generation process. After droplet production, the microfluidic device, which contained the droplets, was placed in a cooling stage (Linkam, THMS 600), and the experiments were monitored using an optical microscope (Olympus, BX-51, transmitted mode) and a CCD camera. Freezing events were determined automatically based on the optical difference of frozen and unfrozen droplets. The microfluidic devices were fabricated in our laboratory using polydimethylsiloxane (PDMS) and a 1-mm thick microscope slide. For each material processed, same microfluidic device was recycled, each time with a new freshly prepared array.

[Figure]

**Figure S5.** a) Generation process of monodispersed droplets by the microfluidic device. b) Example of an array filled with ~40 µm diameter droplets, which are trapped inside the microfluidic device chambers, before cooling applied. The orange squares present the automatic online identification of the droplets.

**S.2.10 FRIDGE (FRankfurt Ice nucleation Deposition freezinG Experiment)**

The FRIDGE methods address either,

    a) deposition/condensation freezing INP or

    b) immersion freezing INP number concentration

Both approaches work offline: aerosol particles are first collected on substrates, and the samples are then processed in the FRIDGE INP counter (Klein et al., 2010; Schrod et al., 2016). The ice nucleus counter FRIDGE itself is a 500 ml thermostated vessel with a cold table inside that carries the sample. A CCD camera records images of the sample through a window on top of the chamber.

    *a)    Deposition/Condensation mode operation (standard mode: FRIDGE-STD)*

For this measurement aerosol is collected from the atmosphere by electrostatic precipitation of the particles onto the surface of a silicon wafer of 45mm diameter Klein et al., 2010). For sampling of aerosol particles air is pumped through the central tube of a cylindrical sampler that carries 12 electrodes of gold wire. The electrodes are arranged concentrically around the inlet, and are at 12 kV voltage against the grounded substrate at the bottom of the sampler. Aerosol particles are charged by emitted electrons and are deposited downstream on the substrate. Usually 30-100 L

of air sample volume are collected. In an effort to limit the depletion of water vapor during analysis by the presence of too many particles (volume effect) the sample volume was adjusted according to the estimated particle number concentration.

After collection, the substrates are stored in at room temperature in petri dishes until analysis. During FIN-02 Samples were stored at room temperature. Most samples were analyzed on-site within a couple of days. For some cases the storage ranged between 10 to 47 days. For analysis, a wafer is placed on the cold table inside the FRIDGE INP counter. The chamber is evacuated, and the temperature of the cold table is adjusted to the desired temperature. Operational temperatures may range between 0 and -35°C. For practical reasons (sample volume, detection limit, occurrence of blank counts) samples are routinely analyzed at -20°C, -25°C and -30°C, and at up to 1% water supersaturation. When the chamber is inflated with water vapor ice grows on the activated INPs to become macroscopic crystals. The vapor pressure is regulated to the desired value. The water vapor saturation is calculated from the pressure inside the chamber and from the temperature of the substrate. The substrate is photographed by a CCD camera. During the first 100 seconds of crystal growth one image is stored every 10 seconds. The image at 100 seconds is usually taken as the final image for counting. The ice crystals are counted automatically. It is assumed that one crystal represents one INP. Only bright objects (i.e., ice crystals) that grow during the 100 seconds to sizes larger than 30 Pixels (*~600µm) are counted. The operating parameters of the chamber, as well as the image processing and the counting of crystals, are controlled by LabView software.

After analysis, the sampling cell is evacuated, the temperature and vapor pressure may be set to new conditions, and the sample may be processed again. The substrates carry a coordinate system (3 laser-marked crosses) that allows to identify from the images the positions of the ice crystals on the substrate. Using this information, the morphology and composition of individual particles at the sites of crystal growth (i.e., the INPs) can be analyzed subsequently by electron microscopical analysis (with EDX) of the substrate.

*b) immersion freezing mode operation (FRIDGE-IMM)*

The measurement of immersion freezing INPs combines membrane filter sampling of aerosol particles with analysis of droplet freezing temperatures of aqueous filter extracts on the cold stage of FRIDGE. Aerosol is sampled on Teflon membrane filters (Fluoropore PTFE, 47 mm, 0.2µm, Merck Millipore Ltd.). Air sample volumes range between a few to 200 liters, the maximum during FIN-02. The particles are extracted into 5 or 10 ml of DIW by agitating. Around 150 drops of 0.5 µl each are taken with a pipette from the washing solution and placed randomly on a silicon plate on the cold stage of FRIDGE. With the chamber almost closed, but at ambient atmospheric pressure the temperature of the cold stage is lowered by 1°C min$^{-1}$. The number of drops that freeze as function of temperature is recorded by the CCD camera and is counted. This process is repeated several times with fresh droplets. A total number of 1000 droplets at minimum is exposed. The INP number concentration is derived following Eqs. (S1) and (S2).

The uncertainty in the measurement is ± 0.2°C for T, and is estimated at around 40% for INP number concentrations at -20°C for Illite NX, but may become lower with decreasing temperature.

**S.2.11 DFPC-ISAC (Dynamic Filter Processing Chamber – Institute of Atmospheric Sciences and Climate (CNR Bologna))**

The Dynamic Filter Processing Chamber (DFPC) (Santachiara et al., 2010; Belosi et al., 2014) is a replica of the Langer dynamic developing chamber (Langer and Rogers, 1975). Concentrations of INPs are detected by the membrane filter technique. Aerosol particles are sampled onto nitrocellulose black gridded membrane filters (0.45 μm porosity Millipore). At the FIN-02 workshop sampling flow rate was 2 L min⁻¹ and the volume sampled was about 20 L. After collection, the filters are stored in Petri dishes. Before being processed the sampled filter is inserted onto a metal plate, previously covered with a smooth surface of paraffin, in order to assure good thermal contact of the filter with the supporting substrate. Subsequently the paraffin is slightly heated and rapidly cooled in order to fill the filter pores.

[Figure]

**Figure S6.** 1: air inlet; 2: minced ice; 3: slit and air temperature thermocouple; 4: filter; 5: filter temperature thermocouple; 6: Peltier cooling device; 7: thermocouple; 8: air outlet; 9: plexiglass cover; 10: observation slit; 11: aluminium plate.

Figure S6 shows the schematic of the chamber, which is housed in a refrigerator. Filtered air is forced by a pump to flow through the chamber in a closed loop. Air enters the chamber through a perforated plate (1), spreads into the ice bed (2) and becomes saturated with respect to ice, which is cooled by the base plate. The temperature of the air is measured just in front of the nozzle (3) aiming the air at the metal plate supporting the sampled filter (4), placed on a top of the Peltier cooled surface (6). The temperatures of the air and of the Peltier device (7) are measured with resistance temperature sensors (PT 100). By controlling the temperatures of the filter and of the air, saturated with respect to finely minced ice and flowing continuously grazing the filter, it is possible to obtain different supersaturations with respect to ice and water, $SS_{ice}$ and $SS_w$, respectively. Therefore, deposition and condensation freezing ice nucleation modes can be investigated. Measurement uncertainties in consideration of T and $SS_w$ uncertainties are estimated as 30% in all cases. Limit of detection is estimated as 0.025 L⁻¹ on the basis of contaminant background of 0.5 per filter (Belosi et al., 2014) and the sample volumes used in this study.

Supersaturations are calculated theoretically from vapour pressures of ice and water (Buck, 1981) at the considered temperatures. The exposure time of the filter is 20 min, long enough to grow sizeable ice crystals on INPs at the considered relative humidity and temperature. Use of the dynamic chamber circumvents some of the problems arising with the static chamber, e.g. that the moisture supply under static conditions may be rather inadequate at the filter

surface, both in overcoming the effect of hygroscopic particles and in activating all potential INPs. Fig. S7 shows a picture of the DFPC with the top cover removed (minced ice is visible) and a filter with ice crystals.

[Figure]

5      **Figure S7.** DFPC housed in a refrigerator (on the left). Ice crystal growth (on the right).

**Table S1.** Direct processing (online) instrument data used in the manuscript figures. Listed are the date, experiment identifier, aerosol particle sample type, sample number if applicable, instrument, relative humidity, temperature, correction factor (\*: see footnote), INP concentration in air (#: see footnote), positive INP concentration uncertainty defined by confidence interval, negative INP uncertainty, total particle number of reference to INP concentration, surface area concentration, INP active fraction, positive uncertainty in active fraction defined by confidence interval, negative uncertainty in active fraction, active site density, positive uncertainty in active site density, negative uncertainty in active site density, threshold size or optical particle counter channel for defining ice crystals versus aerosol particles ("depol" refers to depolarization detection of ice), and evaporation section temperature (&: see footnote).

| Date | Expt | Aerosol | Instrument | Sample | RHw % | Temp °C | Corr fact* | INP Conc.# m⁻³ | ci INP+ m⁻³ | ci INP- m⁻³ | total particles cm⁻³ | Sfc area µm² cm⁻³ | frac | frac_ci+ | frac_ci- | $n_{s,geo}$ m⁻² | $n_{s,geo}$ ci+ m⁻² | $n_{s,geo}$ ci- m⁻² | Ice threshold | Evap T& |
|---|---|---|---|---|---|---|---|---|---|---|---|---|---|---|---|---|---|---|---|---|
| 11-Mar-15 | APC-01 | IS03 | SPIN-TROPOS | | 101.3 | -34.4 | 1.0 | 9750000.0 | 390000.0 | 390000.0 | 390.0 | 230.0 | 2.50E-02 | 1.00E-03 | 1.00E-03 | 4.24E+10 | 1.07E+10 | 1.07E+10 | 3 µm | lamina T |
| 13-Mar-15 | APC-03 | IS03 | CFDC-CSU | | 105.5 | -20.9 | 1.0 | 70900.0 | 10500.0 | 10500.0 | 2000.0 | 430.0 | 3.55E-05 | 5.25E-06 | 5.25E-06 | 1.74E+08 | 8.58E+07 | 8.58E+07 | 3 µm | cold wall |
| 13-Mar-15 | | | INKA | | 105.5 | -20.3 | 1.0 | 75753.7 | 15150.7 | 15150.7 | 1800.0 | 414.0 | 4.21E-05 | 8.42E-06 | 8.42E-06 | 3.08E+09 | 7.80E+08 | 3.29E+09 | >4 µm | cold wall |
| 16-Mar-15 | APC-08 | IS03 | CFDC-CSU | | 105.5 | -30.3 | 4.0 | 18107847.8 | 680412.3 | 171600.0 | 18636.0 | 5569.0 | 9.72E-04 | 3.65E-05 | 3.65E-05 | 3.08E+09 | 7.80E+08 | 3.29E+09 | Chnl 165 | cold wall |
| | | | | | 105.4 | -25.0 | 12.2 | 234774.7 | 148406.8 | 12227.0 | 18636.0 | 5569.0 | 1.26E-05 | 7.96E-06 | 7.96E-06 | 4.00E+07 | 2.72E+07 | 8.37E+08 | Chnl 165 | cold wall |
| | | | | | 105.5 | -20.0 | 10.8 | 79853.2 | 93881.4 | 7392.0 | 18636.0 | 5569.0 | 4.28E-06 | 5.04E-06 | 5.04E-06 | 1.36E+07 | 1.64E+07 | 1.86E+08 | Chnl 165 | cold wall |
| | | IS03 | SPIN-TROPOS | | 103.7 | -34.4 | 1.8 | 181542633.7 | 2648820.8 | 2648820.8 | 18636.0 | 5569.0 | 1.04E-01 | 1.51E-03 | 1.51E-03 | 4.51E+11 | 6.58E+09 | 6.58E+09 | 3 µm | lamina T |
| | | | | | 103.6 | -27.0 | 3.3 | 4853838.9 | 616796.0 | 616796.0 | 18636.0 | 5569.0 | 4.26E-03 | 5.41E-04 | 5.41E-04 | 1.85E+10 | 2.35E+09 | 2.35E+09 | 3 µm | lamina T |
| | | IS03 | INKA | | 105.1 | -25.0 | 5.0 | 196000.0 | 39200.0 | 39200.0 | 18636.0 | 5569.0 | 1.06E-05 | 2.12E-06 | 2.12E-06 | 4.41E+07 | 8.83E+06 | 8.83E+06 | >4 µm | cold wall |
| | | | | | 104.7 | -30.1 | 12.1 | 20449000.0 | 4089800.0 | 4089800.0 | 18636.0 | 5569.0 | 7.35E-04 | 1.47E-04 | 1.47E-04 | 3.19E+09 | 6.39E+08 | 6.39E+08 | >4 µm | cold wall |
| 16-Mar-15 | AIDA-04 | IS03 | AIDA | | 101.7 | -30.6 | 1.0 | 1569810.0 | 1883770.0 | 1255850.0 | 484.0 | 86.0 | 4.18E-03 | 5.02E-03 | 3.35E-03 | 2.35E+10 | 2.84E+10 | 1.90E+10 | N/A | N/A |
| | | | | | 101.3 | -31.1 | 1.0 | 3459500.0 | 4151400.0 | 2767600.0 | 484.0 | 86.0 | 9.34E-03 | 1.12E-02 | 7.47E-03 | 5.26E+10 | 6.33E+10 | 4.24E+10 | N/A | N/A |
| | | | | | 101.1 | -31.6 | 1.0 | 6587250.0 | 7904700.0 | 5269800.0 | 484.0 | 86.0 | 1.81E-02 | 2.17E-02 | 1.45E-02 | 1.02E+11 | 1.23E+11 | 8.21E+10 | N/A | N/A |
| | | | CFDC-CSU | | 105.5 | -25.0 | 1.0 | 68778.0 | 16326.0 | 16326.0 | 394.0 | 93.0 | 1.75E-04 | 4.14E-05 | 4.14E-05 | 7.40E+08 | 1.90E+08 | 1.90E+08 | 3 µm | cold wall |
| | | | | | 105.6 | -30.2 | 1.0 | 584341.0 | 68601.0 | 68601.0 | 421.0 | 97.2 | 1.39E-03 | 1.63E-04 | 1.63E-04 | 6.01E+09 | 9.27E+08 | 9.27E+08 | 3 µm | cold wall |
| | | | | | 109.4 | -25.3 | 1.0 | 828162.0 | 59964.0 | 59964.0 | 394.0 | 93.0 | 2.10E-03 | 1.52E-04 | 1.52E-04 | 8.90E+09 | 1.10E+09 | 1.10E+09 | 3 µm | cold wall |
| | | | | | 111.5 | -30.0 | 1.0 | 8946137.0 | 193338.0 | 193338.0 | 421.0 | 97.2 | 2.12E-02 | 4.59E-04 | 4.59E-04 | 9.20E+10 | 9.42E+09 | 9.42E+09 | 3 µm | cold wall |
| | | | CIC-PNNL | | 106.0 | -30.0 | 1.0 | 17160000.0 | - | - | 550.0 | 99.0 | 1.50E-02 | | - | 8.33E+10 | 2.08E+10 | 2.08E+10 | 3 µm | lamina T |
| 19-Mar-15 | AIDA-10 | IS03 | AIDA | | 101.1 | -26.3 | 1.0 | 192448.0 | 230938.0 | 153958.0 | 400.0 | 132.0 | 6.17E-04 | 7.40E-04 | 4.93E-04 | 1.87E+09 | 2.25E+09 | 1.51E+09 | N/A | N/A |
| | | | | | 101.2 | -26.5 | 1.0 | 543041.0 | 651649.0 | 434433.0 | 400.0 | 132.0 | 1.76E-03 | 2.12E-03 | 1.41E-03 | 5.34E+09 | 6.44E+09 | 4.31E+09 | N/A | N/A |
| | | | | | 102.5 | -26.7 | 1.0 | 732475.0 | 878970.0 | 585980.0 | 400.0 | 132.0 | 2.41E-03 | 2.89E-03 | 1.93E-03 | 7.30E+09 | 8.79E+09 | 5.89E+09 | N/A | N/A |
| | | | | | 102.2 | -26.9 | 1.0 | 823390.0 | 988068.0 | 658712.0 | 400.0 | 132.0 | 2.74E-03 | 3.29E-03 | 2.19E-03 | 8.31E+09 | 1.00E+10 | 6.70E+09 | N/A | N/A |
| | | | | | 100.8 | -27.0 | 1.0 | 1162000.0 | 1394400.0 | 929600.0 | 400.0 | 132.0 | 3.92E-03 | 4.70E-03 | 3.13E-03 | 1.19E+10 | 1.43E+10 | 9.57E+09 | N/A | N/A |
| | | | CFDC-CSU | | 105.5 | -30.6 | 1.0 | 2413879.0 | 120848.0 | 120848.0 | 378.0 | 134.0 | 6.39E-03 | 3.20E-04 | 3.20E-04 | 1.80E+10 | 2.01E+09 | 2.01E+09 | 3 µm | cold wall |
| | | | PIMCA-PINC | | N/A | -38.5 | 1.0 | 299061333.2 | 33953807.2 | 38404474.0 | 400.0 | 134.0 | 7.48E-01 | 8.49E-02 | 9.60E-02 | 2.23E+12 | 6.13E+11 | 1.72E+11 | depol | N/A |
| | | | | | N/A | -38.2 | 1.0 | 276145333.2 | 36848323.2 | 34481656.8 | 400.0 | 134.0 | 6.90E-01 | 9.21E-02 | 8.62E-02 | 2.06E+12 | 5.84E+11 | 1.63E+11 | depol | N/A |
| | | | | | N/A | -37.8 | 1.0 | 212377000.0 | 48921000.0 | 38201000.0 | 400.0 | 134.0 | 5.31E-01 | 1.22E-01 | 9.55E-02 | 1.58E+12 | 5.39E+11 | 1.66E+11 | depol | N/A |
| | | | | | N/A | -36.5 | 1.0 | 134644000.0 | 68755168.4 | 47004502.0 | 400.0 | 134.0 | 3.37E-01 | 1.72E-01 | 1.18E-01 | 1.00E+12 | 5.71E+11 | 2.45E+11 | depol | N/A |
| | | | | | N/A | -36.4 | 1.0 | 120190000.0 | 62093168.8 | 40477168.8 | 400.0 | 134.0 | 3.00E-01 | 1.55E-01 | 1.01E-01 | 8.97E+11 | 5.15E+11 | 2.16E+11 | depol | N/A |
| | | | | | N/A | -36.3 | 1.0 | 97368000.0 | 54838768.0 | 36028768.0 | 400.0 | 134.0 | 2.43E-01 | 1.37E-01 | 9.01E-02 | 7.27E+11 | 4.48E+11 | 2.00E+11 | depol | N/A |
| | | | | | N/A | -34.0 | 1.0 | 83825333.2 | 67323007.2 | 37000340.8 | 400.0 | 134.0 | 2.10E-01 | 1.68E-01 | 9.25E-02 | 6.26E+11 | 5.26E+11 | 2.67E+11 | depol | N/A |
| 26-Mar-15 | AIDA-22 | IS03 | CFDC-TAMU | | 103.0 | -25.5 | 1.0 | 171540.1129 | 148194.826 | 148194.826 | 1850.0 | 890.0 | 9.27E-05 | 8.01E-05 | 8.01E-05 | 1.93E+08 | 1.67E+08 | 1.67E+08 | 2 µm | ~lamina T@ |
| | | | | | 101.7 | -26.5 | 1.0 | 1865311.781 | 287731.3033 | 287731.3033 | 1600.0 | 870.0 | 1.17E-03 | 1.80E-04 | 1.80E-04 | 2.14E+09 | 3.31E+08 | 3.31E+08 | 2 µm | ~lamina T@ |
| | | | | | 104.0 | -30.4 | 1.0 | 1302200.052 | 287731.3033 | 287731.3033 | 1507.0 | 802.0 | 8.64E-04 | 1.91E-04 | 1.91E-04 | 1.62E+09 | 3.59E+08 | 3.59E+08 | 2 µm | ~lamina T@ |
| | | | FRIDGE-STD | 73 | 101.0 | -20.0 | 1.0 | 4390.0 | 3740.0 | 3740.0 | 1500.0 | 852.0 | 2.93E-06 | 2.49E-06 | 2.49E-06 | 5.15E+06 | 4.39E+06 | 4.39E+06 | N/A | N/A |
| | | | | | 101.0 | -25.0 | 1.0 | 14800.0 | 12600.0 | 12600.0 | 1500.0 | 852.0 | 9.87E-06 | 8.40E-06 | 8.40E-06 | 1.74E+07 | 1.48E+07 | 1.48E+07 | N/A | N/A |
| | | | | | 101.0 | -30.0 | 1.0 | 159000.0 | 136000.0 | 136000.0 | 1500.0 | 852.0 | 1.06E-04 | 9.07E-05 | 9.07E-05 | 1.87E+08 | 1.60E+08 | 1.60E+08 | N/A | N/A |
| | | | FRIDGE-STD | 74 | 101.0 | -20.0 | 1.0 | 1100.0 | 936.0 | 936.0 | 1500.0 | 852.0 | 7.33E-07 | 6.24E-07 | 6.24E-07 | 1.39E+06 | 1.23E+06 | 1.23E+06 | N/A | N/A |

| Date | Sample | Code | Instrument | # | | | | | | | | | | | | | | | | |
|---|---|---|---|---|---|---|---|---|---|---|---|---|---|---|---|---|---|---|---|---|
| | | | | | 101.0 | -25.0 | 1.0 | 82300.0 | 70200.0 | 70200.0 | 1500.0 | 852.0 | 5.49E-05 | 4.68E-05 | 4.68E-05 | 1.04E+08 | 9.23E+07 | 9.23E+07 | N/A | N/A |
| | | | | | 101.0 | -30.0 | 1.0 | 777000.0 | 663000.0 | 663000.0 | 1500.0 | 852.0 | 5.18E-04 | 4.42E-04 | 4.42E-04 | 9.80E+08 | 8.72E+08 | 8.72E+08 | N/A | N/A |
| | | | PINC | | 112.4 | -30.5 | 1.0 | 521000.0 | 50000.0 | 50000 | 1500.0 | 901.0 | 3.47E-05 | 3.33E-05 | 3.33E-05 | 7.32E+08 | 1.96E+08 | 1.96E+08 | 3 μm | warm wall |
| | | | | | 109.8 | -25.5 | 1.0 | 17700.0 | 10000.0 | 10000 | 1500.0 | 892.0 | 1.18E-05 | 6.67E-06 | 6.67E-06 | 2.23E+07 | 1.38E+07 | 1.38E+07 | 3 μm | warm wall |
| | | | DFPC-ISAC | | 101.0 | -20.2 | 1.0 | 600.0 | 180.0 | 180.0 | 1600.0 | 850.0 | 4.00E-07 | 1.20E-07 | 1.20E-07 | 7.53E+05 | 2.94E+05 | 2.94E+05 | N/A | N/A |
| 27-Mar-15 | AIDA-25 | IS03 | PINC | | 101.9 | -35.5 | 1.0 | 7500000.0 | 200000.0 | 200000.0 | 973.0 | 303.0 | 3.75E-02 | 1.00E-03 | 1.00E-03 | 1.20E+11 | 3.03E+10 | 3.03E+10 | 3 μm | warm wall |
| | | | | | | | | | | | | | | | | | | | | |
| 18-Mar-15 | AIDA-08 | FS02 | CFDC-CSU | | 105.5 | -30.0 | 1.0 | 175974148.0 | 1175834.0 | 1175834.0 | 380.0 | 155.0 | 4.78E-01 | 3.19E-03 | 3.19E-03 | 1.17E+12 | 2.93E+11 | 2.93E+11 | 3 μm | cold wall |
| | | | DFPC-ISAC | | 101.0 | -20.2 | 1.0 | 87.0 | 26.1 | 26.1 | 380.0 | 155.0 | 2.49E-04 | 7.46E-05 | 7.46E-05 | 6.09E+08 | 2.38E+08 | 2.38E+08 | N/A | N/A |
| | | | FRIDGE-STD | 21 | 101.0 | -20 | 1.0 | 48300.0 | 47800.0 | 47800.0 | 380.0 | 155.0 | 1.27E-04 | 1.26E-04 | 1.26E-04 | 3.12E+08 | 3.18E+08 | 3.18E+08 | N/A | N/A |
| | | | | | 101.0 | -25 | 1.0 | 677000.0 | 669000.0 | 669000.0 | 380.0 | 155.0 | 1.78E-03 | 1.76E-03 | 1.76E-03 | 4.37E+09 | 4.45E+09 | 4.45E+09 | N/A | N/A |
| | | | PIMCA-PINC | | N/A | -34.8 | 1.0 | 358134800.0 | 9184517.9 | 11168117.9 | 380.0 | 155.0 | 9.42E-01 | 2.42E-02 | 2.94E-02 | 2.31E+12 | 5.81E+11 | 1.46E+11 | depol | N/A |
| | | | | | N/A | -31.4 | 1.0 | 365743666.5 | 4440032.5 | 4448899.4 | 380.0 | 155.0 | 9.62E-01 | 1.17E-02 | 1.17E-02 | 2.36E+12 | 5.91E+11 | 1.48E+11 | depol | N/A |
| | | | | | N/A | -29.0 | 1.0 | 361282466.5 | 7665231.2 | 7829898.1 | 380.0 | 155.0 | 9.51E-01 | 2.02E-02 | 2.06E-02 | 2.33E+12 | 5.85E+11 | 1.47E+11 | depol | N/A |
| | | | | | N/A | -27.0 | 1.0 | 288390866.5 | 23884327.0 | 20018460.4 | 380.0 | 155.0 | 7.59E-01 | 6.29E-02 | 5.27E-02 | 1.86E+12 | 4.90E+11 | 1.27E+11 | depol | N/A |
| | | | | | N/A | -25.9 | 1.0 | 193612533.5 | 32959764.4 | 24299564.4 | 380.0 | 155.0 | 5.10E-01 | 8.67E-02 | 6.39E-02 | 1.25E+12 | 3.78E+11 | 1.06E+11 | depol | N/A |
| | | | | | N/A | -25.0 | 1.0 | 92454000.0 | 46089619.0 | 30184085.5 | 380.0 | 155.0 | 2.43E-01 | 1.21E-01 | 7.94E-02 | 5.96E+11 | 3.33E+11 | 1.37E+11 | depol | N/A |
| | | | | | N/A | -24.0 | 1.0 | 52326000.0 | 47028852.4 | 26026252.4 | 380.0 | 155.0 | 1.38E-01 | 1.24E-01 | 6.85E-02 | 3.38E+11 | 3.15E+11 | 1.75E+11 | depol | N/A |
| | | | | | N/A | -23.2 | 1.0 | 34250666.5 | 39367891.7 | 19691491.7 | 380.0 | 155.0 | 9.01E-02 | 1.04E-01 | 5.18E-02 | 2.21E+11 | 2.60E+11 | 1.63E+11 | depol | N/A |
| 19-Mar-15 | APC-14 | FS02 | CFDC-CSU | | 105.0 | -25.0 | 15.1 | 1927241005.9 | 7604512.7 | 7604512.7 | 7750.2 | 3703.1 | 2.49E-01 | 9.81E-04 | 5.80E-04 | 5.20E+11 | 1.30E+11 | 1.30E+11 | 3 μm | cold wall |
| | | | INKA | | 104.1 | -20.0 | 4.1 | 5378173.7 | 114153.8 | 114153.8 | 7750.2 | 3703.1 | 6.94E-04 | 1.47E-05 | 1.47E-05 | 1.45E+09 | 3.64E+08 | 3.64E+08 | >4 μm | cold wall |
| | | | | | 104.7 | -25.0 | 5.7 | 903242700.7 | 2460189.4 | 2460189.4 | 7750.2 | 3703.1 | 1.17E-01 | 3.17E-04 | 3.17E-04 | 2.44E+11 | 6.10E+10 | 6.10E+10 | >4 μm | cold wall |
| | | | | | 104.7 | -30.1 | 8.6 | 2670340000.0 | 18051988.9 | 18051988.9 | 7750.2 | 3703.1 | 3.45E-01 | 2.33E-03 | 2.33E-03 | 7.21E+11 | 1.80E+11 | 1.80E+11 | >4 μm | cold wall |
| | | | PIMCA-PINC | | N/A | -32.9 | 1.0 | 7575532413.6 | 48408615.5 | 78324372.0 | 7750.2 | 3703.1 | 9.77E-01 | 6.25E-03 | 1.01E-02 | 2.05E+12 | 5.12E+11 | 5.12E+11 | depol | N/A |
| | | | | | N/A | -29.4 | 1.0 | 7558494901.9 | 44331121.1 | 86995950.1 | 7750.2 | 3703.1 | 9.75E-01 | 5.72E-03 | 1.12E-02 | 2.04E+12 | 5.10E+11 | 5.11E+11 | depol | N/A |
| | | | | | N/A | -28.4 | 1.0 | 7299031259.4 | 135528351.7 | 202877555.0 | 7750.2 | 3703.1 | 9.42E-01 | 1.75E-02 | 2.62E-02 | 1.97E+12 | 4.94E+11 | 4.96E+11 | depol | N/A |
| | | | | | N/A | -28.1 | 1.0 | 7097371159.5 | 219962614.0 | 194154461.4 | 7750.2 | 3703.1 | 9.16E-01 | 2.84E-02 | 2.51E-02 | 1.92E+12 | 4.83E+11 | 4.82E+11 | depol | N/A |
| | | | | | N/A | -27.4 | 1.0 | 6882354883.3 | 244974891.3 | 305633094.6 | 7750.2 | 3703.1 | 8.88E-01 | 3.16E-02 | 3.94E-02 | 1.86E+12 | 4.69E+11 | 4.72E+11 | depol | N/A |
| | | | | | N/A | -27.1 | 1.0 | 6434755232.9 | 380252656.0 | 427838859.4 | 7750.2 | 3703.1 | 8.30E-01 | 4.91E-02 | 5.52E-02 | 1.74E+12 | 4.46E+11 | 4.50E+11 | depol | N/A |
| | | | | | N/A | -26.1 | 1.0 | 5234714884.3 | 554639769.7 | 479643711.5 | 7750.2 | 3703.1 | 6.75E-01 | 7.16E-02 | 6.19E-02 | 1.41E+12 | 3.84E+11 | 3.76E+11 | depol | N/A |
| | | | | | N/A | -25.5 | 1.0 | 3817694879.0 | 599085113.2 | 434806789.4 | 7750.2 | 3703.1 | 4.93E-01 | 7.73E-02 | 5.61E-02 | 1.03E+12 | 3.04E+11 | 2.83E+11 | depol | N/A |
| | | | SPIN-TROPOS | | 105.7 | -36.5 | 6.6 | 3013418082.5 | 17131132.6 | 17131132.6 | 7750.2 | 3703.1 | 3.89E-01 | 2.21E-03 | 1.43E-03 | 8.14E+11 | 2.03E+11 | 2.03E+11 | 3 μm | lamina T |
| | | | | | 105.1 | -31.7 | 8.0 | 3235505258.5 | 19002992.0 | 19002992.0 | 7750.2 | 3703.1 | 4.17E-01 | 2.45E-03 | 1.59E-03 | 8.74E+11 | 2.18E+11 | 2.18E+11 | 3 μm | lamina T |
| | | | | | 103.9 | -27.2 | 7.5 | 1965720157.0 | 16199258.7 | 16199258.7 | 7750.2 | 3703.1 | 2.54E-01 | 2.09E-03 | 1.94E-03 | 5.31E+11 | 1.33E+11 | 1.33E+11 | 3 μm | lamina T |
| | | | | | 102.8 | -22.9 | 9.5 | 147317204.7 | 5739492.3 | 5739492.3 | 7750.2 | 3703.1 | 1.90E-02 | 7.41E-04 | 7.55E-04 | 3.98E+10 | 1.01E+10 | 1.01E+10 | 3 μm | lamina T |
| | | | SPIN-MIT | | 105.0 | -30.8 | 4.8 | 331363861.3 | - | - | 7750.2 | 3703.1 | 4.28E-02 | | | 8.95E+10 | 2.24E+10 | 2.24E+10 | depol | lamina T[%] |
| | | | | | 105.0 | -21.3 | 6.2 | 217900313.9 | | | 7750.2 | 3703.1 | 2.81E-02 | | | 5.88E+10 | 1.47E+10 | 1.47E+10 | depol | lamina T[%] |
| | | | | | | | | | | | | | | | | | | | | |
| 20-Mar-15 | AIDA-11 | FS02 | CFDC-CSU | | 105.7 | -20.1 | 1.0 | 67556.0 | 7422.0 | 7422.0 | 389.0 | 151.0 | 1.80E-04 | 9.00E-05 | 9.00E-05 | 4.64E+08 | 3.43E+09 | 3.43E+09 | 3 μm | cold wall |
| | | | CIC-PNNL | | 106.0 | -30.0 | 1.0 | 171000000.0 | - | - | 380.0 | 151.0 | 4.50E-01 | - | - | 1.13E+12 | 2.83E+11 | 2.83E+11 | 3 μm | lamina T |
| | | | FRIDGE-STD | 28 | 101.0 | -20.0 | 1.0 | 411000.0 | 406000.0 | 406000.0 | 380.0 | 155.0 | 1.08E-03 | 1.07E-03 | 1.07E-03 | 2.65E+09 | 2.70E+09 | 2.70E+09 | N/A | N/A |
| | | | | | 99.0 | -25.0 | 1.0 | 7880000.0 | 7800000.0 | 7800000.0 | 380.0 | 155.0 | 2.07E-02 | 2.05E-02 | 2.05E-02 | 5.08E+10 | 5.19E+10 | 5.19E+10 | N/A | N/A |
| | | | | | 99.0 | -30.0 | 1.0 | 25800000.0 | 25500000.0 | 25500000.0 | 380.0 | 155.0 | 6.79E-02 | 6.71E-02 | 6.71E-02 | 1.66E+11 | 1.70E+11 | 1.70E+11 | N/A | N/A |
| | | | SPIN-MIT | | 103.0 | -31.1 | 1.0 | 80000.0 | 79.0 | 79.0 | 405.0 | 252.0 | 1.98E-01 | 9.88E-04 | 9.88E-04 | 3.17E+11 | 7.94E+10 | 7.94E+10 | depol | lamina T[%] |
| | | | | | 106.0 | -21.7 | 1.0 | 15000.0 | 6.8 | 6.8 | 382.0 | 151.0 | 3.93E-02 | 4.53E-04 | 4.53E-04 | 9.93E+10 | 2.49E+10 | 2.49E+10 | depol | lamina T[%] |
| | | | SPIN-TROPOS | | 104.1 | -27.2 | 1.0 | 120590.0 | 448.7 | 448.7 | 389.0 | 151.0 | 3.10E-01 | 3.72E-03 | 3.72E-03 | 7.99E+11 | 2.00E+11 | 2.00E+11 | 3 μm | lamina T |
| | | | | | 103.3 | -24.4 | 1.0 | 37500.0 | 94.8 | 94.8 | 375.0 | 145.0 | 1.00E-01 | 2.53E-03 | 2.53E-03 | 2.59E+11 | 6.50E+10 | 6.50E+10 | 3 μm | lamina T |
| 23-Mar-15 | AIDA-14 | FS02 | AIDA | | 101.6 | -20.3 | 1.0 | 1332420.0 | 1598900.0 | 1065940.0 | 300.0 | 114.0 | 5.03E-03 | 6.04E-03 | 4.03E-03 | 1.32E+10 | 1.60E+10 | 1.07E+10 | N/A | N/A |
| | | | | | 100.8 | -20.9 | 1.0 | 6922620.0 | 8307140.0 | 5538100.0 | 300.0 | 114.0 | 2.66E-02 | 3.19E-02 | 2.13E-02 | 7.00E+10 | 8.43E+10 | 5.64E+10 | N/A | N/A |

| Date | Campaign | Sample | Instrument | | | | | | | | | | | | | | | | | |
|---|---|---|---|---|---|---|---|---|---|---|---|---|---|---|---|---|---|---|---|---|
| | | | CFDC-CSU | | 105.6 | -20.2 | 1.0 | 9917338.0 | 150739.0 | 150739.0 | 300.0 | 114.0 | 3.41E-02 | 5.19E-04 | 5.19E-04 | 8.98E+10 | 2.40E+10 | 2.40E+10 | 3 µm | cold wall |
| | | | | | 105.5 | -25.6 | 1.0 | 105206098.0 | 751115.0 | 751115.0 | 268.0 | 102.0 | 3.69E-01 | 2.63E-03 | 2.63E-03 | 9.69E+11 | 2.42E+11 | 2.42E+11 | 3 µm | cold wall |
| | | | | | 105.5 | -30.2 | 1.0 | 121683497.0 | 1277222.0 | 1277222.0 | 270.0 | 107.0 | 4.51E-01 | 4.73E-03 | 4.73E-03 | 1.14E+12 | 2.84E+11 | 2.84E+11 | 3 µm | cold wall |
| | | | INKA | | 104.0 | -20.0 | 1.0 | 139424.0 | 27884.9 | 27884.9 | 290.0 | 115.0 | 4.81E-04 | 9.62E-05 | 9.62E-05 | 1.21E+09 | 3.88E+08 | 3.88E+08 | >4 µm | cold wall |
| | | | | | 103.3 | -30.0 | 1.0 | 70256300.0 | 14051300.0 | 14051300.0 | 290.0 | 102.0 | 2.42E-01 | 4.85E-02 | 4.85E-02 | 6.89E+11 | 2.21E+11 | 2.21E+11 | >4 µm | cold wall |
| | | | PINC | | 110.9 | -30.5 | 1.0 | 98000.0 | 700.0 | 700.0 | 300.0 | 114.0 | 3.92E-01 | 2.80E-03 | 2.80E-03 | 1.03E+12 | 2.58E+11 | 2.58E+11 | 3 µm | warm wall |
| | | | | | 104.7 | -25.3 | 1.0 | 2840.0 | 140.0 | 140.0 | 268.0 | 102.0 | 1.14E-02 | 5.60E-04 | 5.60E-04 | 2.98E+10 | 7.61E+09 | 7.61E+09 | 3 µm | warm wall |
| | | | | | 106.6 | -20.3 | 1.0 | 14.7 | 20.0 | 20.0 | 270.0 | 107.0 | 5.88E-05 | 8.00E-05 | 8.00E-05 | 1.48E+08 | 2.05E+08 | 2.05E+08 | 3 µm | warm wall |
| | | | | | | | | | | | | | | | | | | | | |
| 17-Mar-15 | APC-16 | SM04 | CFDC-CSU | | 105.2 | -9.3 | 12.9 | 169686600.0 | 3096.0 | 3096.0 | 27802.0 | 4028.0 | 6.10E-03 | 1.11E-04 | 1.11E-04 | 4.21E+10 | 1.06E+10 | 1.06E+10 | 2 µm | cold wall |
| | | | | | 101.8 | -19.9 | 5.0 | 186550000.0 | 2205.0 | 2205.0 | 27802.0 | 4028.0 | 6.71E-03 | 7.93E-05 | 7.93E-05 | 4.63E+10 | 1.16E+10 | 1.16E+10 | 3 µm | cold wall |
| | | | CFDC-TAMU | | 109.3 | -33.0 | 2.8 | 242718750.0 | 94660312.0 | 94660312.0 | 27802.0 | 4028.0 | 8.73E-03 | 3.40E-03 | 3.40E-03 | 6.03E+10 | 2.79E+10 | 2.79E+10 | 2 µm | ~lamina T[Φ] |
| | | | | | 101.0 | -21.0 | 5.3 | 847058824.0 | 330352941.0 | 330352941.0 | 27802.0 | 4028.0 | 3.05E-02 | 1.19E-02 | 1.19E-02 | 2.10E+11 | 9.74E+10 | 9.74E+10 | 2 µm | ~lamina T[Φ] |
| | | | INKA | | 104.3 | -10.1 | 3.8 | 151875000.0 | 30375000.0 | 30375000.0 | 27802.0 | 4028.0 | 4.98E-03 | 1.13E-03 | 1.13E-03 | 3.44E+10 | 1.16E+10 | 1.16E+10 | >4 µm | cold wall |
| | | | | | 104.4 | -10.1 | 4.0 | 156246400.0 | 31249320.0 | 31249320.0 | 27802.0 | 4028.0 | 3.16E-03 | 1.14E-04 | 1.14E-04 | 2.18E+10 | 5.51E+09 | 5.51E+09 | >4 µm | cold wall |
| | | | | | 104.9 | -15.0 | 5.7 | 149797834.4 | 29959509.6 | 29959509.6 | 27802.0 | 4028.0 | 5.39E-03 | 1.08E-03 | 1.11E-04 | 3.72E+10 | 1.19E+10 | 9.33E+09 | >4 µm | cold wall |
| | | | | | 109.8 | -15.0 | 5.9 | 530290657.9 | 106057894.7 | 106057894.7 | 27802.0 | 4028.0 | 1.91E-02 | 3.81E-03 | 1.11E-04 | 1.32E+11 | 4.21E+10 | 3.29E+10 | >4 µm | cold wall |
| | | | | | 104.2 | -20.1 | 8.5 | 175567924.5 | 35113584.9 | 35113584.9 | 27802.0 | 4028.0 | 6.31E-03 | 1.26E-03 | 1.11E-04 | 4.36E+10 | 1.40E+10 | 1.09E+10 | >4 µm | cold wall |
| | | | | | 104.2 | -20.1 | 8.6 | 171373714.3 | 34274742.9 | 34274742.9 | 27802.0 | 4028.0 | 6.16E-03 | 1.23E-03 | 1.11E-04 | 4.25E+10 | 1.36E+10 | 1.07E+10 | >4 µm | cold wall |
| | | | | | 110.2 | -20.0 | 10.5 | 202632907.0 | 40526476.7 | 40526476.7 | 27802.0 | 4028.0 | 7.29E-03 | 1.46E-03 | 1.11E-04 | 5.03E+10 | 1.61E+10 | 1.26E+10 | >4 µm | cold wall |
| | | | SPIN-MIT | | 105.0 | -20.0 | 3.4 | 82127547.2 | - | - | 27802.0 | 4028.0 | 2.95E-03 | | | 2.04E+10 | 5.10E+09 | 5.10E+09 | depol | lamina T[※] |
| | | | | | 105.0 | -15.0 | 4.1 | 180294545.5 | - | - | 27802.0 | 4028.0 | 6.48E-03 | | | 4.48E+10 | 1.12E+10 | 1.12E+10 | depol | lamina T[※] |
| | | | SPIN-TROPOS | | 101.3 | -18.6 | 3.5 | 138461538.5 | 31475769.2 | 31475769.2 | 27802.0 | 4028.0 | 4.98E-03 | 1.13E-03 | 1.13E-03 | 3.44E+10 | 1.16E+10 | 1.16E+10 | 3 µm | lamina T |
| | | | | | 100.9 | -14.0 | 4.4 | 87804878.0 | 3165365.9 | 3165365.9 | 27802.0 | 4028.0 | 3.16E-03 | 1.14E-04 | 1.14E-04 | 2.18E+10 | 5.51E+09 | 5.51E+09 | 3 µm | lamina T |
| 17-Mar-15 | AIDA-06 | SM04 | AIDA | | 101.6 | -24.2 | 1.0 | 2505730.0 | 3006880.0 | 2004580.0 | 241.0 | 23.3 | 1.11E-02 | 1.33E-02 | 8.85E-03 | 1.14E+11 | 1.38E+11 | 9.23E+10 | N/A | N/A |
| | | | | | 101.6 | -24.6 | 1.0 | 2316010.0 | 2779210.0 | 1852810.0 | 241.0 | 23.3 | 1.04E-02 | 1.24E-02 | 8.30E-03 | 1.07E+11 | 1.29E+11 | 8.65E+10 | N/A | N/A |
| | | | | | 101.8 | -25.2 | 1.0 | 4938950.0 | 5926740.0 | 3951160.0 | 241.0 | 23.3 | 2.26E-02 | 2.71E-02 | 1.80E-02 | 2.33E+11 | 2.81E+11 | 1.88E+11 | N/A | N/A |
| | | | CFDC-CSU | | 105.5 | -10.0 | 1.0 | 552972.0 | 59028.0 | 59028.0 | 259.0 | 24.0 | 2.14E-03 | 2.28E-04 | 2.28E-04 | 2.30E+10 | 3.37E+09 | 3.37E+09 | 3 µm | cold wall |
| | | | | | 105.5 | -15.0 | 1.0 | 799823.0 | 73751.0 | 73751.0 | 241.0 | 23.0 | 3.24E-03 | 2.99E-04 | 2.99E-04 | 3.48E+10 | 4.73E+09 | 4.73E+09 | 3 µm | cold wall |
| | | | | | 108.4 | -9.9 | 1.0 | 733245.0 | 32036.0 | 32036.0 | 259.0 | 24.0 | 2.83E-03 | 1.24E-04 | 1.24E-04 | 3.06E+10 | 3.33E+09 | 3.33E+09 | 3 µm | cold wall |
| | | | | | 124.4 | -15.4 | 1.0 | 1140013.0 | 115665.0 | 115665.0 | 241.0 | 23.0 | 4.62E-03 | 4.68E-04 | 4.68E-04 | 4.96E+10 | 7.06E+09 | 7.06E+09 | 3 µm | cold wall |
| | | | FRIDGE-STD | 6 | 101.0 | -10.0 | 1.0 | 648000.0 | 297000.0 | 297000.0 | 260.0 | 25.0 | 2.59E-03 | 1.19E-03 | 1.19E-03 | 2.70E+10 | 1.41E+10 | 1.41E+10 | N/A | N/A |
| | | | | | 101.0 | -15.0 | 1.0 | 1150000.0 | 530000.0 | 530000.0 | 260.0 | 25.0 | 4.60E-03 | 2.12E-03 | 2.12E-03 | 4.78E+10 | 2.51E+10 | 2.51E+10 | N/A | N/A |
| | | | | | 101.0 | -20.0 | 1.0 | 1350000.0 | 621000.0 | 621000.0 | 260.0 | 25.0 | 5.40E-03 | 2.48E-03 | 2.48E-03 | 5.62E+10 | 2.94E+10 | 2.94E+10 | N/A | N/A |
| | | | SPIN-TROPOS | | 102.0 | -18.3 | 1.0 | 1912.5 | 215.0 | 215.0 | 255.0 | 26.0 | 7.50E-03 | 8.43E-04 | 8.43E-04 | 7.36E+10 | 2.02E+10 | 2.02E+10 | 3 µm | lamina T |
| | | | | | 101.2 | -13.8 | 1.0 | 725.2 | 139.1 | 139.1 | 259.0 | 24.0 | 2.80E-03 | 5.37E-04 | 5.37E-04 | 3.02E+10 | 9.52E+09 | 9.52E+09 | 3 µm | lamina T |
| 21-Mar-15 | AIDA-13 | SM04 | AIDA | | 101.1 | -9.1 | 1.0 | 2215550.0 | 2658660.0 | 1772440.0 | 229.0 | 23.0 | 9.81E-03 | 1.18E-02 | 7.85E-03 | 1.18E+11 | 7.88E+10 | 7.88E+10 | N/A | N/A |
| | | | CFDC-CSU | | 105.5 | -9.5 | 1.0 | 1135824.0 | 44125.0 | 44125.0 | 229.0 | 23.0 | 4.53E-03 | 1.76E-04 | 1.76E-04 | 4.94E+10 | 5.30E+09 | 5.30E+09 | 3 µm | cold wall |
| | | | | | 105.5 | -14.9 | 1.0 | 1128809.0 | 73599.0 | 73599.0 | 229.0 | 23.0 | 4.50E-03 | 2.93E-04 | 2.93E-04 | 4.91E+10 | 5.86E+09 | 5.86E+09 | 3 µm | cold wall |
| | | | | | 106.0 | -9.6 | 1.0 | 733245.0 | 32036.0 | 32036.0 | 229.0 | 23.0 | 4.57E-03 | 1.73E-04 | 1.73E-04 | 4.99E+10 | 5.34E+09 | 5.34E+09 | 3 µm | cold wall |
| | | | | | 109.4 | -14.5 | 1.0 | 1148100.0 | 43328.0 | 115665.0 | 229.0 | 23.0 | 4.82E-03 | 3.11E-04 | 3.11E-04 | 5.26E+10 | 6.26E+09 | 6.26E+09 | 3 µm | cold wall |
| | | | CFDC-TAMU | | 102.4 | -15.0 | 1.0 | 984751.9 | 406643.9 | 406643.9 | 75.0 | 8.0 | 4.10E-03 | 1.69E-03 | 1.69E-03 | 3.85E+10 | 1.86E+10 | 1.86E+10 | 2 µm | ~lamina T[Φ] |
| | | | | | 101.2 | -20.0 | 1.0 | 839963.3 | 685955.4 | 685955.4 | 230.0 | 23.0 | 3.50E-03 | 2.86E-03 | 2.86E-03 | 3.50E+10 | 2.99E+10 | 2.99E+10 | 2 µm | ~lamina T[Φ] |
| | | | INKA | | 104.6 | -10.0 | 1.0 | 1210537.0 | 77950.0 | 216243.0 | 279.0 | 24.9 | 4.51E-03 | 9.01E-04 | 9.01E-04 | 5.05E+10 | 1.62E+10 | 1.62E+10 | >4 µm | cold wall |
| | | | | | 104.6 | -15.0 | 1.0 | 866660.0 | 173332.0 | 173332.0 | 250.0 | 23.0 | 3.61E-03 | 7.22E-04 | 7.22E-04 | 3.93E+10 | 1.26E+10 | 1.26E+10 | >4 µm | cold wall |
| 27-Mar-15 | AIDA-27 | SM04 | AIDA | | 99.5 | -12.6 | 1.0 | 1953930.0 | 2344720.0 | 1563140.0 | 236.0 | 28.3 | 9.01E-03 | 1.08E-02 | 7.21E-03 | 9.05E+10 | 6.06E+10 | 6.06E+10 | N/A | N/A |
| | | | | | 101.3 | -13.6 | 1.0 | 2453210.0 | 2943850.0 | 1962570.0 | 236.0 | 28.3 | 1.15E-02 | 1.38E-02 | 9.22E-03 | 1.16E+11 | 7.75E+10 | 7.75E+10 | N/A | N/A |
| | | | | | | | | | | | | | | | | | | | | |
| 16-Mar-15 | AIDA-05 | SDAr01 | CFDC-CSU | | 105.5 | -25.0 | 1.0 | 4403989.0 | 215519.0 | 215519.0 | 370.0 | 185.0 | 1.26E-02 | 6.15E-04 | 6.15E-04 | 2.51E+10 | 6.36E+09 | 6.36E+09 | 3 µm | cold wall |

| | | | | | | | | | | | | | | | | | | | | |
|---|---|---|---|---|---|---|---|---|---|---|---|---|---|---|---|---|---|---|---|---|
| | | | | | 105.4 | -30.1 | 1.0 | 9803125.0 | 135193.0 | 135193.0 | 370.0 | 185.0 | 2.93E-02 | 4.04E-04 | 4.04E-04 | 5.86E+10 | 1.47E+10 | 1.47E+10 | 3 µm | cold wall |
| | | | | | 105.5 | -30.3 | 1.0 | 21252577.0 | 448727.0 | 448727.0 | 370.0 | 185.0 | 6.61E-02 | 1.40E-03 | 1.40E-03 | 1.32E+11 | 3.31E+10 | 3.31E+10 | 3 µm | cold wall |
| | | | | | 109.6 | -25.2 | 1.0 | 14823835.0 | 366291.0 | 366291.0 | 370.0 | 185.0 | 4.23E-02 | 1.04E-03 | 1.04E-03 | 8.45E+10 | 2.11E+10 | 2.11E+10 | 3 µm | cold wall |
| | | | | | 110.5 | -29.0 | 1.0 | 35103544.0 | 542065.0 | 542065.0 | 370.0 | 185.0 | 1.05E-01 | 1.61E-03 | 1.61E-03 | 2.09E+11 | 5.24E+10 | 5.24E+10 | 3 µm | cold wall |
| | | | | | 110.5 | -30.5 | 1.0 | 48786312.0 | 599960.0 | 599960.0 | 370.0 | 185.0 | 1.52E-01 | 1.87E-03 | 1.87E-03 | 3.03E+11 | 7.59E+10 | 7.59E+10 | 3 µm | cold wall |
| | | PIMCA-PINC | | | N/A | -36.1 | 1.0 | 223032300.0 | 50843550.0 | 32954050.0 | 370.0 | 185.0 | 6.03E-01 | 1.37E-01 | 8.91E-02 | 1.21E+12 | 4.08E+11 | 1.18E+11 | depol | N/A |
| | | | | | N/A | -34.9 | 1.0 | 174525300.0 | 73737127.2 | 44798194.0 | 370.0 | 185.0 | 4.72E-01 | 1.99E-01 | 1.21E-01 | 9.43E+11 | 4.63E+11 | 1.66E+11 | depol | N/A |
| | | | | | N/A | -33.1 | 1.0 | 108551833.2 | 60830879.0 | 29242745.4 | 370.0 | 185.0 | 2.93E-01 | 1.64E-01 | 7.90E-02 | 5.87E+11 | 3.60E+11 | 1.32E+11 | depol | N/A |
| | | | | | N/A | -30.5 | 1.0 | 67196933.2 | 61870305.9 | 21883172.7 | 370.0 | 185.0 | 1.82E-01 | 1.67E-01 | 5.91E-02 | 3.63E+11 | 3.47E+11 | 1.42E+11 | depol | N/A |
| 17-Mar-15 | APC-10 | SDAr01 | CFDC-CSU | | 105.4 | -19.9 | 5.2 | 151304.3 | 83478.3 | 83478.3 | 3439.0 | 3038.0 | 4.40E-05 | 2.43E-05 | 2.43E-05 | 4.98E+07 | 3.02E+07 | 3.02E+07 | Chnl 165 | cold wall |
| | | | | | 105.5 | -25.0 | 5.5 | 12992727.3 | 681818.2 | 681818.2 | 3439.0 | 3038.0 | 3.78E-03 | 1.98E-04 | 1.98E-04 | 4.28E+09 | 1.09E+09 | 1.09E+09 | Chnl 165 | cold wall |
| | | | | | 105.5 | -30.2 | 6.3 | 235515789.5 | 3448421.1 | 3448421.1 | 3439.0 | 3038.0 | 6.85E-02 | 1.00E-03 | 1.00E-03 | 7.75E+10 | 1.94E+10 | 1.94E+10 | Chnl 165 | cold wall |
| | | INKA | | | 104.2 | -20.0 | 1.9 | 474285.7 | 95238.1 | 95238.1 | 3439.0 | 3038.0 | 1.38E-04 | 2.77E-05 | 2.43E-05 | 1.56E+08 | 5.01E+07 | 4.77E+07 | >4 µm | cold wall |
| | | | | | 104.9 | -25.0 | 3.0 | 16350000.0 | 3270000.0 | 3270000.0 | 3439.0 | 3038.0 | 4.75E-03 | 9.51E-04 | 1.98E-04 | 5.38E+09 | 1.72E+09 | 1.36E+09 | >4 µm | cold wall |
| | | | | | 104.8 | -30.0 | 4.6 | 156073846.2 | 31213846.2 | 31213846.2 | 3439.0 | 3038.0 | 4.54E-02 | 9.08E-03 | 1.00E-03 | 5.14E+10 | 1.64E+10 | 1.29E+10 | >4 µm | cold wall |
| | | CIC-PNNL | | | 106.0 | -25.0 | 2.4 | 26860800.0 | - | - | 3439.0 | 3038.0 | 7.81E-03 | | | 8.84E+09 | 2.21E+09 | 2.21E+09 | 3 µm | lamina T |
| | | | | | 106.0 | -30.0 | 3.4 | 328097142.9 | - | - | 3439.0 | 3038.0 | 9.54E-02 | | | 1.08E+11 | 2.70E+10 | 2.70E+10 | 3 µm | lamina T |
| | | SPIN-TROPOS | | | 106.2 | -36.5 | 3.5 | 402185792.3 | 5434754.1 | 5434754.1 | 3439.0 | 3038.0 | 1.17E-01 | 1.58E-03 | 2.43E-05 | 1.32E+11 | 3.31E+10 | 3.31E+10 | 3 µm | lamina T |
| | | | | | 105.6 | -31.7 | 4.3 | 209066666.7 | 4462933.3 | 4462933.3 | 3439.0 | 3038.0 | 6.08E-02 | 1.30E-03 | 2.43E-05 | 6.88E+10 | 1.73E+10 | 1.72E+10 | 3 µm | lamina T |
| | | | | | 104.3 | -27.2 | 5.3 | 53333333.3 | 2602666.7 | 2602666.7 | 3439.0 | 3038.0 | 1.55E-02 | 7.57E-04 | 2.43E-05 | 1.76E+10 | 4.47E+09 | 4.39E+09 | 3 µm | lamina T |
| | | | | | 102.9 | -23.0 | 5.6 | 5861052.6 | 887017.5 | 887017.5 | 3439.0 | 3038.0 | 1.70E-03 | 2.58E-04 | 2.43E-05 | 1.93E+09 | 5.64E+08 | 4.83E+08 | 3 µm | lamina T |
| | | PIMCA-PINC | | | N/A | -37.4 | 1.0 | 2866688000.0 | 137243683.2 | 136539683.2 | 3439.0 | 3038.0 | 8.34E-01 | 3.99E-02 | 3.97E-02 | 9.44E+11 | 2.40E+11 | 2.40E+11 | depol | N/A |
| | | | | | N/A | -36.4 | 1.0 | 2634869334.4 | 243020144.0 | 213569475.2 | 3439.0 | 3038.0 | 7.66E-01 | 7.07E-02 | 6.21E-02 | 8.67E+11 | 2.31E+11 | 2.28E+11 | depol | N/A |
| | | | | | N/A | -35.5 | 1.0 | 2523424000.0 | 278050764.8 | 253282764.8 | 3439.0 | 3038.0 | 7.34E-01 | 8.08E-02 | 7.36E-02 | 8.31E+11 | 2.27E+11 | 2.24E+11 | depol | N/A |
| | | | | | N/A | -33.9 | 1.0 | 2056736000.0 | 407975027.2 | 304839027.2 | 3439.0 | 3038.0 | 5.98E-01 | 1.19E-01 | 8.86E-02 | 6.77E+11 | 2.16E+11 | 1.97E+11 | depol | N/A |
| | | | | | N/A | -32.2 | 1.0 | 1574480000.0 | 456975712.0 | 305167712.0 | 3439.0 | 3038.0 | 4.58E-01 | 1.33E-01 | 8.87E-02 | 5.18E+11 | 1.99E+11 | 1.64E+11 | depol | N/A |
| | | | | | N/A | -31.2 | 1.0 | 1151029334.4 | 537707110.4 | 306197776.0 | 3439.0 | 3038.0 | 3.35E-01 | 1.56E-01 | 8.90E-02 | 3.79E+11 | 2.01E+11 | 1.38E+11 | depol | N/A |
| | | | | | N/A | -29.3 | 1.0 | 775477334.4 | 513819776.0 | 226929107.2 | 3439.0 | 3038.0 | 2.25E-01 | 1.49E-01 | 6.60E-02 | 2.55E+11 | 1.81E+11 | 9.82E+10 | depol | N/A |
| 19-Mar-15 | AIDA-09 | SDAr01 | AIDA | | 101.4 | -24.1 | 1.0 | 609000.0 | 731000.0 | 487000.0 | 310.0 | 172.0 | 2.12E-03 | 1.70E-03 | 2.54E-03 | 3.82E+09 | 4.60E+09 | 3.08E+09 | N/A | N/A |
| | | | | | 101.1 | -24.8 | 1.0 | 1730000.0 | 2070000.0 | 1380000.0 | 310.0 | 172.0 | 6.10E-03 | 4.88E-03 | 7.32E-03 | 1.10E+10 | 1.32E+10 | 8.86E+09 | N/A | N/A |
| | | | | | 100.9 | -25.3 | 1.0 | 5180000.0 | 6210000.0 | 4140000.0 | 310.0 | 172.0 | 1.86E-02 | 1.49E-02 | 2.23E-02 | 3.36E+10 | 4.04E+10 | 2.71E+10 | N/A | N/A |
| | | DFPC-ISAC | | | 101.0 | -20.2 | 1.0 | 11300.0 | 3390.0 | 3390.0 | 310.0 | 172.0 | 5.65E-05 | 1.70E-05 | 1.70E-05 | 1.02E+08 | 3.98E+07 | 3.98E+07 | N/A | N/A |
| | | FRIDGE-STD | | 23 | 101.0 | -20.0 | 1.0 | 6590.0 | 7060.0 | 7060.0 | 310.0 | 172.0 | 2.21E-05 | 2.37E-05 | 2.37E-05 | 3.99E+07 | 4.38E+07 | 4.38E+07 | N/A | N/A |
| | | | | | 101.0 | -25.0 | 1.0 | 292000.0 | 313000.0 | 313000.0 | 310.0 | 172.0 | 9.80E-04 | 1.05E-03 | 1.05E-03 | 1.77E+09 | 1.94E+09 | 1.94E+09 | N/A | N/A |
| | | | | | 101.0 | -30.0 | 1.0 | 3920000.0 | 4200000.0 | 4200000.0 | 310.0 | 172.0 | 1.32E-02 | 1.41E-02 | 1.41E-02 | 2.37E+10 | 2.61E+10 | 2.61E+10 | N/A | N/A |
| T% | | SPIN-MIT | | | 108.0 | -30.0 | 1.0 | 4840000.0 | 96234.2 | 96234.2 | 310.0 | 172.0 | 1.53E-02 | 3.10E-04 | 3.10E-04 | 2.79E+10 | 7.00E+09 | 7.00E+09 | depol | lamina T% |
| T% | | | | | 104.0 | -25.0 | 1.0 | 5500000.0 | 108825.5 | 108825.5 | 310.0 | 172.0 | 1.84E-02 | 3.51E-04 | 3.51E-04 | 3.21E+10 | 8.05E+09 | 8.05E+09 | depol | lamina T% |
| | | SPIN-TROPOS | | | 104.4 | -27.3 | 1.0 | 10850000.0 | 500000.4 | 500.4 | 310.0 | 172.0 | 3.50E-02 | 1.63E-03 | 1.63E-03 | 6.31E+10 | 1.60E+10 | 1.60E+10 | 3 µm | warm wall |
| 26-Mar-15 | AIDA-24 | SDAr01 | PINC | | 105.1 | -30.5 | 1.0 | 2040000.0 | 101957.0 | 101957.0 | 177.0 | 105.0 | 8.33E-03 | 5.76E-04 | 1.00E-06 | 1.40E+10 | 3.51E+09 | 3.51E+09 | 3 µm | warm wall |
| | | | | | | | | | | | | | | | | | | | | |
| 18-Mar-15 | APC-12 | SDT01 | CFDC-CSU | | 105.3 | -19.9 | 6.0 | 980255.2 | 214538.4 | 214538.4 | 3001.0 | 2010.0 | 3.27E-04 | 7.15E-05 | 7.15E-05 | 4.88E+08 | 1.62E+08 | 1.62E+08 | Chnl 165 | cold wall |
| | | | | | 105.6 | -30.6 | 2.8 | 107772733.3 | 621116.7 | 621116.7 | 3001.0 | 2010.0 | 3.59E-02 | 2.07E-04 | 2.07E-04 | 5.36E+10 | 1.34E+10 | 1.34E+10 | Chnl 165 | cold wall |
| | | INKA | | | 103.8 | -20.0 | 2.2 | 307368.5 | 61517.6 | 61517.6 | 3001.0 | 2010.0 | 1.02E-04 | 2.05E-05 | 2.05E-05 | 1.53E+08 | 4.90E+07 | 4.90E+07 | >4 µm | cold wall |
| | | | | | 104.8 | -25.0 | 3.5 | 7555123.3 | 1511372.1 | 1511372.1 | 3001.0 | 2010.0 | 2.52E-03 | 5.04E-04 | 5.04E-04 | 3.76E+09 | 1.20E+09 | 1.20E+09 | >4 µm | cold wall |
| | | | | | 104.8 | -30.0 | 5.0 | 52200360.0 | 10438080.0 | 10438080.0 | 3001.0 | 2010.0 | 1.74E-02 | 3.48E-03 | 3.48E-03 | 2.60E+10 | 8.31E+09 | 8.31E+09 | >4 µm | cold wall |
| | | CIC-PNNL | | | 106.0 | -25.0 | 2.4 | 54370689.7 | - | - | 3001.0 | 2010.0 | 1.81E-02 | 0.00E+00 | - | - | 6.76E+09 | 6.76E+09 | 3 µm | lamina T |
| | | | | | 106.0 | -30.0 | 5.6 | 137950400.0 | - | - | 3001.0 | 2010.0 | 4.60E-02 | 0.00E+00 | - | - | 1.72E+10 | 1.72E+10 | 3 µm | lamina T |
| | | SPIN-TROPOS | | | 105.0 | -36.4 | 2.6 | 407329565.2 | 4763478.3 | 4763478.3 | 3001.0 | 2010.0 | 1.36E-01 | 1.59E-03 | 1.59E-03 | 2.03E+11 | 5.07E+10 | 5.07E+10 | 3 µm | lamina T |
| | | | | | 105.0 | -31.7 | 3.5 | 157894736.8 | 3600000.0 | 3600000.0 | 3001.0 | 2010.0 | 5.26E-02 | 1.20E-03 | 1.20E-03 | 7.86E+10 | 1.97E+10 | 1.97E+10 | 3 µm | lamina T |

| | | | | | | | | | | | | | | | | | | | | |
|---|---|---|---|---|---|---|---|---|---|---|---|---|---|---|---|---|---|---|---|---|
| | | | | | 103.1 | -26.7 | 4.4 | 35912408.8 | 1927007.3 | 1927007.3 | 3001.0 | 2010.0 | 1.20E-02 | 6.42E-04 | 6.42E-04 | 1.79E+10 | 4.57E+09 | 4.57E+09 | 3 μm | lamina T |
| | | | PIMCA-PINC | | N/A | -36.2 | 1.0 | 2380899999.0 | 177632439.0 | 181722438.0 | 3001.0 | 2010.0 | 7.93E-01 | 5.92E-02 | 6.06E-02 | 1.51E-02 | 3.95E-03 | 3.96E-03 | depol | N/A |
| | | | | | N/A | -35.2 | 1.0 | 1967649999.0 | 232633491.0 | 195113490.0 | 3001.0 | 2010.0 | 6.56E-01 | 6.76E-02 | 6.50E-02 | 1.63E-02 | 4.40E-03 | 4.37E-03 | depol | N/A |
| | | | | | N/A | -33.7 | 1.0 | 1732509999.0 | 306133458.0 | 262643457.0 | 3001.0 | 2010.0 | 5.77E-01 | 8.90E-02 | 8.75E-02 | 2.19E-02 | 6.43E-03 | 6.40E-03 | depol | N/A |
| | | | | | N/A | -32.4 | 1.0 | 1299159999.0 | 236006148.0 | 165046149.0 | 3001.0 | 2010.0 | 4.33E-01 | 6.86E-02 | 5.50E-02 | 1.37E-02 | 4.07E-03 | 3.86E-03 | depol | N/A |
| | | | | | N/A | -28.9 | 1.0 | 803850000.0 | 244168851.0 | 152658852.0 | 3001.0 | 2010.0 | 2.68E-01 | 7.10E-02 | 5.09E-02 | 1.27E-02 | 4.63E-03 | 3.99E-03 | depol | N/A |
| 18-Mar-15 | AIDA-07 | SDT01 | AIDA | | 102.3 | -23.8 | 1.0 | 259237.0 | 311084.0 | 207390.0 | 270.0 | 131.0 | 1.08E-03 | 8.62E-04 | 1.29E-03 | 2.22E+09 | 2.67E+09 | 1.79E+09 | N/A | N/A |
| | | | | | 102.3 | -24.5 | 1.0 | 537809.0 | 645371.0 | 430247.0 | 270.0 | 131.0 | 2.27E-03 | 1.82E-03 | 2.73E-03 | 4.68E+09 | 5.64E+09 | 3.78E+09 | N/A | N/A |
| | | | | | 102.2 | -25.0 | 1.0 | 1093630.0 | 1312360.0 | 874904.0 | 270.0 | 131.0 | 4.69E-03 | 3.75E-03 | 5.62E-03 | 9.66E+09 | 1.16E+10 | 7.79E+09 | N/A | N/A |
| | | | | | 101.6 | -25.4 | 1.0 | 1567560.0 | 1881070.0 | 1254050.0 | 270.0 | 131.0 | 6.81E-03 | 5.44E-03 | 8.17E-03 | 1.40E+10 | 1.69E+10 | 1.13E+10 | N/A | N/A |
| | | | | | 100.7 | -25.7 | 1.0 | 2011470.0 | 2413760.0 | 1609180.0 | 270.0 | 131.0 | 8.85E-03 | 7.08E-03 | 1.06E-02 | 1.82E+10 | 2.20E+10 | 1.47E+10 | N/A | N/A |
| | | | CFDC-CSU | | 105.5 | -25.6 | 1.0 | 923063.0 | 33533.0 | 33533.0 | 270.0 | 131.0 | 3.76E-03 | 1.37E-04 | 1.37E-04 | 7.76E+09 | 7.74E+09 | 1.94E+09 | 3 μm | cold wall |
| | | | | | 105.6 | -30.3 | 1.0 | 5842154.0 | 203342.0 | 203342.0 | 270.0 | 131.0 | 2.43E-02 | 8.47E-04 | 8.47E-04 | 5.08E+10 | 5.01E+10 | 1.25E+10 | 3 μm | cold wall |
| | | | | | 112.0 | -25.2 | 1.0 | 5273442.0 | 113137.0 | 113137.0 | 270.0 | 131.0 | 2.16E-02 | 4.64E-04 | 4.64E-04 | 4.50E+10 | 4.46E+10 | 1.13E+10 | 3 μm | cold wall |
| | | | | | 106.5 | -30.3 | 1.0 | 6523451.0 | 234653.0 | 234653.0 | 270.0 | 131.0 | 2.72E-02 | 9.78E-04 | 9.78E-04 | 5.68E+10 | 5.60E+10 | 1.40E+10 | 3 μm | cold wall |
| | | | FRIDGE-STD | 16 | 101.0 | -20.0 | 1.0 | 1330.0 | 1030.0 | 1030.0 | 270.0 | 131.0 | 1.02E-05 | 7.86E-06 | 7.86E-06 | 2.09E+07 | 1.70E+07 | 1.70E+07 | N/A | N/A |
| | | | | | 101.0 | -25.0 | 1.0 | 146000.0 | 115000.0 | 115000.0 | 270.0 | 131.0 | 1.11E-03 | 8.78E-04 | 8.78E-04 | 2.30E+09 | 1.90E+09 | 1.90E+09 | N/A | N/A |
| | | | | | 101.0 | -30.0 | 1.0 | 733000.0 | 576000.0 | 576000.0 | 270.0 | 131.0 | 5.60E-03 | 4.40E-03 | 4.40E-03 | 1.15E+10 | 9.51E+09 | 9.51E+09 | N/A | N/A |
| | | | SPIN-TROPOS | | 104.1 | -27.1 | 1.0 | 1781000.0 | 194000.0 | 194000.0 | 270.0 | 131.0 | 6.50E-03 | 7.98E-04 | 7.98E-04 | 1.36E+10 | 3.79E+09 | 3.79E+09 | 3 μm | lamina T |
| 20-Mar-15 | AIDA-12 | SDT01 | CFDC-CSU | | 105.6 | -20.6 | 1.0 | 4400.0 | 4575.0 | 4399.0 | 285.0 | 118.0 | 1.69E-05 | 1.76E-05 | 1.68E-05 | 4.09E+07 | 1.02E+07 | 1.02E+07 | 3 μm | cold wall |
| | | | | | 110.8 | -20.4 | 1.0 | 22400.0 | 18153.0 | 18153.0 | 285.0 | 118.0 | 8.60E-05 | 6.97E-05 | 6.97E-05 | 2.08E+08 | 5.19E+07 | 5.19E+07 | 3 μm | cold wall |
| | | | DFPC-ISAC | | 101.0 | -20.2 | 1.0 | 25000.0 | 7500.0 | 7500.0 | 290.0 | 116.0 | 1.00E-04 | 3.00E-05 | 3.00E-05 | 2.50E+08 | 9.76E+07 | 9.76E+07 | N/A | N/A |
| | | | FRIDGE-STD | 31 | 101.0 | -20.0 | 1.0 | 46100.0 | 36200.0 | 36200.0 | 291.0 | 119.0 | 1.58E-04 | 1.24E-04 | 1.24E-04 | 3.87E+08 | 3.19E+08 | 3.19E+08 | N/A | N/A |
| | | | | | 101.0 | -25.0 | 1.0 | 698000.0 | 548000.0 | 548000.0 | 291.0 | 119.0 | 2.40E-03 | 1.88E-03 | 1.88E-03 | 5.87E+09 | 4.83E+09 | 4.83E+09 | N/A | N/A |
| | | | | | 101.0 | -30.0 | 1.0 | 4380000.0 | 3440000.0 | 3440000.0 | 291.0 | 119.0 | 1.51E-02 | 1.18E-02 | 1.18E-02 | 3.68E+10 | 3.03E+10 | 3.03E+10 | N/A | N/A |
| | | | SPIN-TROPOS | | 105.8 | -36.3 | 1.0 | 7275000.0 | 401209.0 | 401209.0 | 291.0 | 120.0 | 2.50E-02 | 1.48E-03 | 1.48E-03 | 6.06E+10 | 1.56E+10 | 1.56E+10 | 3 μm | lamina T |
| | | | PIMCA-PINC | | N/A | -33.5 | 1.0 | 59357899.9 | 37332147.2 | 18606697.0 | 285.0 | 118.0 | 2.08E-01 | 1.31E-01 | 6.53E-02 | 5.03E+11 | 3.40E+11 | 1.37E+11 | depol | N/A |
| | | | | | N/A | -31.3 | 1.0 | 45349200.0 | 44806043.0 | 18488193.1 | 285.0 | 118.0 | 1.59E-01 | 1.57E-01 | 6.49E-02 | 3.84E+11 | 3.92E+11 | 1.87E+11 | depol | N/A |
| | | | | | N/A | -28.2 | 1.0 | 37908799.9 | 47585575.1 | 18572575.1 | 285.0 | 118.0 | 1.33E-01 | 1.67E-01 | 6.52E-02 | 3.21E+11 | 4.11E+11 | 2.26E+11 | depol | N/A |

*: correction factor refers to the particle number concentration ratio in Eq. (3) of the manuscript, the ratio between the integrated average particle concentration during offline sampling to the total particle concentration at the time of online sampling.

**: INP concentration is the actual INP concentration at the time of sampling when the correction factor is 1, but is equal to $n_{INP,online,corr}$ (sample time) when the correction factor exceeds 1.**

&: evaporation section temperature refers to the temperature of the online flow chamber walls within the evaporation region of the instruments. Cold wall means that the ice surfaces on both walls are adjusted to the cold wall temperature to induce evaporation. Warm wall is just the opposite. Lamina T means that the walls are adjusted to the aerosol lamina temperature to induce evaporation without a change in temperature.

@: For the CFDC-TAMU, wall temperatures are maintained in the evaporation section, but the warm wall is covered by a hydrophobic material to actively stimulate evaporation, rather than wall temperature control alone. It is not known the extent to which this limits heat transfer from the warm wall, and thereby leads to cooling in the evaporation region.

%: The SPIN-MIT evaporation region temperature was 0 to 5°C warmer than the lamina temperature.

**Table S2.** Post-processed (offline) instrument data used in the manuscript figures. Listed are the date, experiment identifier, aerosol particle sample type, instrument, sample identifier (shared, or separate filter and number if applicable), temperature, INP concentration in air, positive INP concentration uncertainty defined by confidence interval, negative INP uncertainty, total particle number of reference to INP concentration, surface area concentration, INP active fraction, positive uncertainty in active fraction defined by confidence interval, negative uncertainty in active fraction, active site density, positive uncertainty in active site density, negative uncertainty in active site density, droplet or aliquot volumes used, and and notes regarding the data point ("binned" if multiple experiments are combined, and "selected points" if not all temperature points are shown.

| Date | Expt | aerosol | Instrument | Sample | Temp | INP Conc | ci INP+ | ci INP- | total particles | Sfc area | frac | frac_ci+ | frac_ci- | ns,geo | ns,geo ci+ | ns,geo ci- | Drop/aliquot volume | Notes |
|---|---|---|---|---|---|---|---|---|---|---|---|---|---|---|---|---|---|---|
| | | | | | °C | m⁻³ | m⁻³ | m⁻³ | cm⁻³ | μm² cm⁻³ | | | | m⁻² | m⁻² | m⁻² | μL | |
| 16-Mar-15 | APC-07 | IS03 | BINARY | Shared impinger | -21.0 | 1630.0 | 0.0 | 0.0 | 18636.2 | 5568.6 | 8.75E-08 | 0.00E+00 | 0.00E+00 | 2.93E+05 | 7.32E+04 | 7.32E+04 | 0.6 | binned data |
| | | | | | -22.0 | 4710.0 | 17400.0 | 2500.0 | 18636.2 | 5568.6 | 2.53E-07 | 9.34E-07 | 1.34E-07 | 8.46E+05 | 3.13E+06 | 4.96E+05 | 0.6 | binned data |
| | | | | | -23.0 | 19900.0 | 48600.0 | 12300.0 | 18636.2 | 5568.6 | 1.07E-06 | 2.61E-06 | 6.60E-07 | 3.57E+06 | 8.77E+06 | 2.38E+06 | 0.6 | binned data |
| | | | | | -24.0 | 84000.0 | 205000.0 | 53300.0 | 18636.2 | 5568.6 | 4.51E-06 | 1.10E-05 | 2.86E-06 | 1.51E+07 | 3.70E+07 | 1.03E+07 | 0.6 | binned data |
| | | | | | -25.0 | 260000.0 | 350000.0 | 119000.0 | 18636.2 | 5568.6 | 1.40E-05 | 1.88E-05 | 6.39E-06 | 4.67E+07 | 6.39E+07 | 2.44E+07 | 0.6 | binned data |
| | | | | | -26.0 | 542000.0 | 416000.0 | 336000.0 | 18636.2 | 5568.6 | 2.91E-05 | 2.23E-05 | 1.80E-05 | 9.73E+07 | 7.86E+07 | 6.51E+07 | 0.6 | binned data |
| | | | | | -27.0 | 1520000.0 | 1080000.0 | 1050000.0 | 18636.2 | 5568.6 | 8.16E-05 | 5.80E-05 | 5.63E-05 | 2.73E+08 | 2.06E+08 | 2.01E+08 | 0.6 | binned data |
| | | | | | -28.0 | 3510000.0 | 3170000.0 | 1290000.0 | 18636.2 | 5568.6 | 1.88E-04 | 1.70E-04 | 6.92E-05 | 6.30E+08 | 5.91E+08 | 2.80E+08 | 0.6 | binned data |
| | | | | | -29.0 | 6070000.0 | 650000.0 | 650000.0 | 18636.2 | 5568.6 | 3.26E-04 | 3.49E-05 | 3.49E-05 | 1.09E+09 | 2.96E+08 | 2.96E+08 | 0.6 | binned data |
| | | | IS | Filter | -8.9 | 2.8 | 12.1 | 2.3 | 18636.2 | 5568.6 | 1.48E-10 | 4.95E+02 | 6.50E-10 | 1.22E-10 | 2.18E+03 | 4.27E+02 | 50 | selected points |
| | | | | | -9.0 | 5.6 | 13.9 | 4.1 | 18636.2 | 5568.6 | 3.01E-10 | 1.01E+03 | 7.47E-10 | 2.19E-10 | 2.51E+03 | 7.76E+02 | 50 | selected points |
| | | | | | -9.5 | 5.6 | 13.9 | 4.1 | 18636.2 | 5568.6 | 3.01E-10 | 1.01E+03 | 7.47E-10 | 2.19E-10 | 2.51E+03 | 7.76E+02 | 50 | selected points |
| | | | | | -10.0 | 18.0 | 19.8 | 9.9 | 18636.2 | 5568.6 | 9.67E-10 | 3.24E+03 | 1.06E-09 | 5.34E-10 | 3.64E+03 | 1.96E+03 | 50 | selected points |
| | | | | | -11.0 | 18.0 | 19.8 | 9.9 | 18636.2 | 5568.6 | 9.67E-10 | 3.24E+03 | 1.06E-09 | 5.34E-10 | 3.64E+03 | 1.96E+03 | 50 | selected points |
| | | | | | -12.0 | 36.6 | 26.6 | 16.7 | 18636.2 | 5568.6 | 1.96E-09 | 6.57E+03 | 1.43E-09 | 8.99E-10 | 5.05E+03 | 3.43E+03 | 50 | selected points |
| | | | | | -13.3 | 45.3 | 29.5 | 19.7 | 18636.2 | 5568.6 | 2.43E-09 | 8.13E+03 | 1.58E-09 | 1.06E-09 | 5.68E+03 | 4.08E+03 | 50 | selected points |
| | | | | | 14.0 | 45.3 | 29.5 | 19.7 | 18636.2 | 5568.6 | 2.43E-09 | 8.13E+03 | 1.58E-09 | 1.06E-09 | 5.68E+03 | 4.08E+03 | 50 | selected points |
| | | | | | -15.0 | 85.1 | 42.7 | 32.8 | 18636.2 | 5568.6 | 4.57E-09 | 1.53E+04 | 2.29E-09 | 1.76E-09 | 8.57E+03 | 7.03E+03 | 50 | selected points |
| | | | | | -16.0 | 170.9 | 310.5 | 113.7 | 18636.2 | 5568.6 | 9.17E-09 | 3.07E+04 | 1.67E-08 | 6.10E-09 | 5.63E+04 | 2.18E+04 | 50 | selected points |
| | | | | | -17.0 | 295.0 | 368.3 | 171.5 | 18636.2 | 5568.6 | 1.58E-08 | 5.30E+04 | 1.98E-08 | 9.20E-09 | 6.75E+04 | 3.35E+04 | 50 | selected points |
| | | | | | -18.0 | 499.4 | 449.5 | 252.6 | 18636.2 | 5568.6 | 2.68E-08 | 8.97E+04 | 2.41E-08 | 1.36E-08 | 8.38E+04 | 5.06E+04 | 50 | selected points |
| | | | | | -19.0 | 816.0 | 560.5 | 363.7 | 18636.2 | 5568.6 | 4.38E-08 | 1.47E+05 | 3.01E-08 | 1.95E-08 | 1.07E+05 | 7.49E+04 | 50 | selected points |
| | | | | | -20.0 | 1853.9 | 905.0 | 708.2 | 18636.2 | 5568.6 | 9.95E-08 | 3.33E+05 | 4.86E-08 | 3.80E-08 | 1.83E+05 | 1.52E+05 | 50 | selected points |
| | | | | | -20.5 | 4109.6 | 1844.5 | 1647.7 | 18636.2 | 5568.6 | 2.21E-07 | 7.38E+05 | 9.90E-08 | 8.84E-08 | 3.79E+05 | 3.49E+05 | 50 | selected points |
| | | | | | -21.0 | 5899.3 | 7366.7 | 3430.1 | 18636.2 | 5568.6 | 3.17E-07 | 1.06E+06 | 3.95E-07 | 1.84E-07 | 1.35E+06 | 6.70E+05 | 50 | selected points |
| | | | | | -21.5 | 11466.7 | 9529.0 | 5592.4 | 18636.2 | 5568.6 | 6.15E-07 | 2.06E+06 | 5.11E-07 | 3.00E-07 | 1.79E+06 | 1.13E+06 | 50 | selected points |
| | | | | | -22.0 | 28704.1 | 15292.0 | 11355.4 | 18636.2 | 5568.6 | 1.54E-06 | 5.15E+06 | 8.21E-07 | 6.09E-07 | 3.03E+06 | 2.41E+06 | 50 | selected points |
| | | | | | -22.5 | 64454.8 | 28560.3 | 24623.7 | 18636.2 | 5568.6 | 3.46E-06 | 1.16E+07 | 1.53E-06 | 1.32E-06 | 5.89E+06 | 5.28E+06 | 50 | selected points |
| | | | | | -23.0 | 171430.6 | 169050.9 | 90319.4 | 18636.2 | 5568.6 | 9.20E-06 | 3.08E+07 | 9.07E-06 | 4.85E-06 | 3.13E+07 | 1.80E+07 | 50 | selected points |
| | | | | | -23.5 | 229334.5 | 190579.1 | 111847.6 | 18636.2 | 5568.6 | 1.23E-05 | 4.12E+07 | 1.02E-05 | 6.00E-06 | 3.57E+07 | 2.26E+07 | 50 | selected points |
| | | | | | -24.0 | 292509.4 | 212709.8 | 133978.3 | 18636.2 | 5568.6 | 1.57E-05 | 5.25E+07 | 1.14E-05 | 7.19E-06 | 4.04E+07 | 2.74E+07 | 50 | selected points |
| | | | | | -24.5 | 526170.6 | 290088.8 | 211357.3 | 18636.2 | 5568.6 | 2.82E-05 | 9.45E+07 | 1.56E-05 | 1.13E-05 | 5.72E+07 | 4.47E+07 | 50 | selected points |
| | | | | | -25.0 | 574082.3 | 305839.2 | 227107.8 | 18636.2 | 5568.6 | 3.08E-05 | 1.03E+08 | 1.64E-05 | 1.22E-05 | 6.07E+07 | 4.82E+07 | 50 | selected points |
| | | | | | -25.5 | 719508.3 | 360434.7 | 279306.3 | 18636.2 | 5568.6 | 3.86E-05 | 1.29E+08 | 1.93E-05 | 1.50E-05 | 7.23E+07 | 5.97E+07 | 50 | selected points |
| | | | | | -26.0 | 1094122.5 | 515677.1 | 429288.1 | 18636.2 | 5568.6 | 5.87E-05 | 1.96E+08 | 2.77E-05 | 2.30E-05 | 1.05E+08 | 9.14E+07 | 50 | selected points |
| | | | | | -26.6 | 1444056.6 | 733940.5 | 630831.2 | 18636.2 | 5568.6 | 7.75E-05 | 2.59E+08 | 3.94E-05 | 3.38E-05 | 1.47E+08 | 1.31E+08 | 50 | selected points |

| | | | | | | | | | | | | | | | | | | |
|---|---|---|---|---|---|---|---|---|---|---|---|---|---|---|---|---|---|---|
| | | | KIT-CS | Shared impinger | -26.8 | 938469.4 | 0.0 | 0.0 | 18636.2 | 5568.6 | 5.04E-05 | 0.00E+00 | 0.00E+00 | 1.69E+08 | 4.21E+07 | 4.21E+07 | 4.2 | |
| | | | | | -27.7 | 2303457.9 | 2039358.3 | 2039358.3 | 18636.2 | 5568.6 | 1.24E-04 | 1.09E-04 | 1.09E-04 | 4.14E+08 | 3.81E+08 | 3.81E+08 | 4.2 | |
| | | | | | -28.7 | 7287660.4 | 4318310.9 | 4318310.9 | 18636.2 | 5568.6 | 3.91E-04 | 2.32E-04 | 2.32E-04 | 1.31E+09 | 8.42E+08 | 8.42E+08 | 4.2 | |
| | | | | | -29.7 | 16616677.8 | 7473340.4 | 7473340.4 | 18636.2 | 5568.6 | 8.92E-04 | 4.01E-04 | 4.01E-04 | 2.98E+09 | 1.54E+09 | 1.54E+09 | 4.2 | |
| | | | | | -30.6 | 32222547.3 | 10854765.2 | 10854765.2 | 18636.2 | 5568.6 | 1.73E-03 | 5.82E-04 | 5.82E-04 | 5.79E+09 | 2.43E+09 | 2.43E+09 | 4.2 | |
| | | | | | -31.6 | 55556777.0 | 11227616.0 | 11227616.0 | 18636.2 | 5568.6 | 2.98E-03 | 6.02E-04 | 6.02E-04 | 9.98E+09 | 3.21E+09 | 3.21E+09 | 4.2 | |
| | | | | | -32.7 | 90103941.6 | 5543139.6 | 5543139.6 | 18636.2 | 5568.6 | 4.83E-03 | 2.97E-04 | 2.97E-04 | 1.62E+10 | 4.17E+09 | 4.17E+09 | 4.2 | |
| | | | M-AL | Shared impinger | -22.0 | 1124.2 | 346.7 | 346.7 | 18636.2 | 5568.6 | 6.03E-08 | 1.86E-08 | 1.86E-08 | 2.02E+05 | 8.01E+04 | 8.01E+04 | 4.2 | |
| | | | | | -21.0 | 860.8 | 262.0 | 262.0 | 18636.2 | 5568.6 | 4.62E-08 | 1.41E-08 | 1.41E-08 | 1.55E+05 | 6.09E+04 | 6.09E+04 | 4.2 | |
| | | | | | -20.0 | 737.8 | 227.5 | 227.5 | 18636.2 | 5568.6 | 3.96E-08 | 1.22E-08 | 1.22E-08 | 1.32E+05 | 5.26E+04 | 5.26E+04 | 4.2 | |
| | | | | | -19.0 | 562.7 | 179.2 | 179.2 | 18636.2 | 5568.6 | 3.02E-08 | 9.62E-09 | 9.62E-09 | 1.01E+05 | 4.09E+04 | 4.09E+04 | 4.2 | |
| | | | | | -18.0 | 344.9 | 111.4 | 111.4 | 18636.2 | 5568.6 | 1.85E-08 | 5.98E-09 | 5.98E-09 | 6.19E+04 | 2.53E+04 | 2.53E+04 | 4.2 | |
| | | | | | -17.0 | 191.9 | 63.1 | 63.1 | 18636.2 | 5568.6 | 1.03E-08 | 3.39E-09 | 3.39E-09 | 3.45E+04 | 1.42E+04 | 1.42E+04 | 4.2 | |
| | | | | | -16.0 | 94.3 | 28.5 | 28.5 | 18636.2 | 5568.6 | 5.06E-09 | 1.53E-09 | 1.53E-09 | 1.69E+04 | 6.64E+03 | 6.64E+03 | 4.2 | |
| | | | | | -15.0 | 94.3 | 28.5 | 28.5 | 18636.2 | 5568.6 | 5.06E-09 | 1.53E-09 | 1.53E-09 | 1.69E+04 | 6.64E+03 | 6.64E+03 | 4.2 | |
| | | | | | -14.0 | 94.3 | 37.3 | 37.3 | 18636.2 | 5568.6 | 5.06E-09 | 2.00E-09 | 2.00E-09 | 1.69E+04 | 7.92E+03 | 7.92E+03 | 4.2 | |
| | | | NCSU-CS | Shared impinger | -17.0 | 233.5 | 0.0 | 0.0 | 18636.2 | 5568.6 | 1.25E-08 | 0.00E+00 | 0.00E+00 | 4.19E+04 | 1.05E+04 | 1.05E+04 | 1 | binned data |
| | | | | | -18.0 | 297.1 | 0.0 | 0.0 | 18636.2 | 5568.6 | 1.59E-08 | 0.00E+00 | 0.00E+00 | 5.34E+04 | 1.33E+04 | 1.33E+04 | 1 | binned data |
| | | | | | -19.0 | 360.8 | 0.0 | 0.0 | 18636.2 | 5568.6 | 1.94E-08 | 0.00E+00 | 0.00E+00 | 6.48E+04 | 1.62E+04 | 1.62E+04 | 1 | binned data |
| | | | | | -20.0 | 439.3 | 0.0 | 0.0 | 18636.2 | 5568.6 | 2.36E-08 | 0.00E+00 | 0.00E+00 | 7.89E+04 | 1.97E+04 | 1.97E+04 | 1 | binned data |
| | | | | | -21.0 | 49167.9 | 95188.9 | 49167.0 | 18636.2 | 5568.6 | 2.64E-06 | 5.11E-06 | 2.64E-06 | 8.83E+06 | 1.72E+07 | 9.10E+06 | 1 | binned data |
| | | | | | -22.0 | 83620.9 | 97537.7 | 83620.0 | 18636.2 | 5568.6 | 4.49E-06 | 5.23E-06 | 4.49E-06 | 1.50E+07 | 1.79E+07 | 1.55E+07 | 1 | binned data |
| | | | | | -23.0 | 232405.8 | 229532.0 | 229532.0 | 18636.2 | 5568.6 | 1.25E-05 | 1.23E-05 | 1.23E-05 | 4.17E+07 | 4.25E+07 | 4.25E+07 | 1 | binned data |
| | | | | | -24.0 | 525003.8 | 501473.1 | 501473.1 | 18636.2 | 5568.6 | 2.82E-05 | 2.69E-05 | 2.69E-05 | 9.43E+07 | 9.31E+07 | 9.31E+07 | 1 | binned data |
| | | | | | -25.0 | 1315450.4 | 1284452.0 | 1284452.0 | 18636.2 | 5568.6 | 7.06E-05 | 6.89E-05 | 6.89E-05 | 2.36E+08 | 2.38E+08 | 2.38E+08 | 1 | binned data |
| | | | NIPI | Shared impinger | -24.0 | 35300.0 | 47900.0 | 20400.0 | 18636.2 | 5568.6 | 1.89E-06 | 2.57E-06 | 1.09E-06 | 6.34E+06 | 8.75E+06 | 3.99E+06 | 1 | binned data |
| | | | | | -23.0 | 15400.0 | 33700.0 | 10580.0 | 18636.2 | 5568.6 | 8.26E-07 | 1.81E-06 | 5.68E-07 | 2.77E+06 | 6.09E+06 | 2.02E+06 | 1 | binned data |
| | | | | | -22.0 | 5600.0 | 15000.0 | 4080.0 | 18636.2 | 5568.6 | 3.00E-07 | 8.05E-07 | 2.19E-07 | 1.01E+06 | 2.71E+06 | 7.75E+05 | 1 | binned data |
| | | | | | -21.0 | 2100.0 | 8000.0 | 1665.0 | 18636.2 | 5568.6 | 1.13E-07 | 4.29E-07 | 8.93E-08 | 3.77E+05 | 1.44E+06 | 3.14E+05 | 1 | binned data |
| | | | | | -20.0 | 701.0 | 3979.0 | 596.0 | 18636.2 | 5568.6 | 3.76E-08 | 2.14E-07 | 3.20E-08 | 1.26E+05 | 7.15E+05 | 1.12E+05 | 1 | binned data |
| | | | | | -19.0 | 342.0 | 838.0 | 242.6 | 18636.2 | 5568.6 | 1.84E-08 | 4.50E-08 | 1.30E-08 | 6.14E+04 | 1.51E+05 | 4.62E+04 | 1 | binned data |
| | | | | | -18.0 | 119.0 | 393.0 | 91.5 | 18636.2 | 5568.6 | 6.39E-09 | 2.11E-08 | 4.91E-09 | 2.14E+04 | 7.08E+04 | 1.73E+04 | 1 | binned data |
| | | | | | -17.0 | 89.4 | 89.6 | 44.7 | 18636.2 | 5568.6 | 4.80E-09 | 4.81E-09 | 2.40E-09 | 1.61E+04 | 1.66E+04 | 8.97E+03 | 1 | binned data |
| | | | | | -12.0 | 205.0 | 205.0 | 103.0 | 18636.2 | 5568.6 | 1.10E-08 | 1.10E-08 | 5.53E-09 | 3.68E+04 | 3.79E+04 | 2.07E+04 | 1 | binned data |
| | | | VODCA | Shared impinger | -34.0 | 410000000.0 | 39513400.3 | 39513400.3 | 18636.2 | 5568.6 | 2.20E-02 | 2.12E-03 | 2.12E-03 | 7.36E+10 | 1.97E+10 | 1.97E+10 | 0.000004 - 0.000036 | selected points |
| | | | | | -33.0 | 205000000.0 | 144781645.5 | 144781645.5 | 18636.2 | 5568.6 | 1.10E-02 | 7.77E-03 | 7.77E-03 | 3.68E+10 | 2.76E+10 | 2.76E+10 | 0.000004 - 0.000036 | selected points |
| | | | | | -32.0 | 110000000.0 | 45533430.3 | 45533430.3 | 18636.2 | 5568.6 | 5.90E-03 | 2.44E-03 | 2.44E-03 | 1.98E+10 | 9.55E+09 | 9.55E+09 | 0.000004 - 0.000036 | selected points |
| | | | | | -31.0 | 48000000.0 | 5213291.1 | 5213291.1 | 18636.2 | 5568.6 | 2.58E-03 | 2.80E-04 | 2.80E-04 | 8.62E+09 | 2.35E+09 | 2.35E+09 | 0.000004 - 0.000036 | selected points |
| 26-Mar-15 | AIDA-22 | | FRIDGE-IMM | Filter 17 | -13.2 | 1090.0 | 1700.0 | 793.0 | 1688 | 892 | 6.46E-07 | 1.01E-06 | 4.70E-07 | 1.22E+06 | 1.93E+06 | 9.40E+05 | 0.5 | selected points |
| | | | | | -14.1 | 1910.0 | 2030.0 | 1141.0 | 1688 | 892 | 1.13E-06 | 1.20E-06 | 6.76E-07 | 2.14E+06 | 2.34E+06 | 1.39E+06 | 0.5 | selected points |
| | | | | | -16.0 | 3010.0 | 2380.0 | 1510.0 | 1688 | 892 | 1.78E-06 | 1.41E-06 | 8.95E-07 | 3.37E+06 | 2.80E+06 | 1.89E+06 | 0.5 | selected points |
| | | | | | -18.0 | 5780.0 | 3050.0 | 2200.0 | 1688 | 892 | 3.42E-06 | 1.81E-06 | 1.30E-06 | 6.48E+06 | 3.78E+06 | 2.95E+06 | 0.5 | selected points |
| | | | | | -20.0 | 14000.0 | 4400.0 | 3600.0 | 1688 | 892 | 8.29E-06 | 2.61E-06 | 2.13E-06 | 1.57E+07 | 6.30E+06 | 5.63E+06 | 0.5 | selected points |
| | | | | | -22.0 | 38500.0 | 7200.0 | 6300.0 | 1688 | 892 | 2.28E-05 | 4.27E-06 | 3.73E-06 | 4.32E+07 | 1.35E+07 | 1.29E+07 | 0.5 | selected points |
| | | | | | -24.0 | 149000.0 | 15000.0 | 14000.0 | 1688 | 892 | 8.83E-05 | 8.89E-06 | 8.29E-06 | 1.67E+08 | 4.50E+07 | 4.46E+07 | 0.5 | selected points |
| | | | | | -26.0 | 826000.0 | 80000.0 | 74000.0 | 1688 | 892 | 6.99E-04 | 9.48E-05 | 8.89E-05 | 1.32E+09 | 3.76E+08 | 3.71E+08 | 0.5 | selected points |
| | | | | | -27.4 | 1600000.0 | 430000.0 | 300000.0 | 1688 | 892 | 9.48E-04 | 2.55E-04 | 1.78E-04 | 1.79E+09 | 6.58E+08 | 5.61E+08 | 0.5 | selected points |
| 17-Mar-15 | APC-09 | SDAr01 | BINARY | Shared impinger | -10.2 | 1720.0 | 0.0 | 0.0 | 3439.3 | 3038.1 | 5.00E-07 | 0.00E+00 | 0.00E+00 | 5.66E+05 | 1.42E+05 | 1.42E+05 | 0.6 | binned data |
| | | | | | -11.2 | 1720.0 | 0.0 | 0.0 | 3439.3 | 3038.1 | 5.00E-07 | 0.00E+00 | 0.00E+00 | 5.66E+05 | 1.42E+05 | 1.42E+05 | 0.6 | binned data |

[remaining 88,738 characters of this post omitted]